# TRUST: A DECENTRALIZED FRAMEWORK FOR AUDITING LARGE LANGUAGE MODEL REASONING

Large Language Models (LLMs) can produce complex reasoning chains, offering a window into their decision-making processes. However, verifying the quality (e.g., faithfulness and harmlessness) of these intermediate steps is a critical, unsolved challenge. Current auditing methods are often centralized, opaque, and struggle to scale, creating significant risks for the deployment of proprietary models in high-stakes domains. This paper addresses four key challenges in reasoning verification: (1) *Robustness*: Centralized systems are single points of failure, vulnerable to attacks and systemic bias. (2) *Scalability*: The length and complexity of reasoning traces create a severe bottleneck for human auditors. (3) *Opacity*: Internal auditing processes are typically hidden from end-users, undermining public trust. (4) *Privacy*: Model providers risk intellectual property theft or unauthorized model distillation when exposing complete reasoning traces. To overcome these barriers, we introduce TRUST, a decentralized framework for auditing LLM reasoning. TRUST makes the following contributions: (1) It establishes a decentralized consensus mechanism among a diverse set of auditors, provably guaranteeing audit correctness with up to 30% malicious participants and mitigating single-source bias. (2) It introduces a scalable decomposition method that transforms reasoning traces into hierarchical directed acyclic graphs, enabling atomic reasoning steps to be audited in parallel by a distributed network. (3) All verification decisions are recorded on a transparent blockchain ledger, creating a permanent and publicly auditable record. (4) The framework is privacy-preserving by distributing only partial segments of the reasoning trace to auditors, thus protecting the full proprietary logic from distillation. We provide theoretical guarantees for the security and economic incentives of the TRUST framework. Experiments across multiple LLMs (e.g., GPT-OSS, DeepSeek-r1, Qwen) and reasoning tasks (e.g., mathematical, medical, science, and humanities) demonstrate that TRUST is highly effective at identifying reasoning flaws and is significantly more resilient to corrupted auditors than centralized baselines. Our work pioneers the field of decentralized AI auditing, offering a practical pathway for the safe and secure deployment of AI systems.

## 1 INTRODUCTION

The capabilities of large language models (LLMs) have expanded from text generation to complex, multi-step reasoning, leading to the development of Large Reasoning Models (LRMs) that produce explicit reasoning traces (Wei et al., 2022). While offering a view of a model's logical flow, this explicit reasoning also exposes potential flaws, including logical errors, a lack of faithfulness to the model's true internal state (Turpin et al., 2023), and safety vulnerabilities. The verification of these intermediate steps is a critical prerequisite for the safe and reliable deployment of LRMs in high-stakes domains such as medicine (Singhal et al., 2023), law (Chalkidis et al., 2021), and finance (Wang et al., 2023). The urgency of this task is underscored by emerging regulatory frameworks like the EU AI Act (COM, 2021) and the NIST AI RMF (AI, 2023), which mandate rigorous documentation and monitoring (OECD, 2019). However, even as recent works advance auditing for semi-structured reasoning (Leng et al., 2025) or propose new faithfulness metrics (Lanham et al., 2023), prevailing auditing methods remain misaligned with this paradigm. Their centralized, opaque, and unscalable nature creates unacceptable risks, as they either rely on a single trusted entity, cannot process the volume and complexity of reasoning traces, or force a dangerous trade-off between public transparency and the protection of proprietary models.

The inadequacy of current auditing systems stems from four interconnected challenges. A primary issue is a lack of **Robustness** since systems relying on a single auditor, whether a human expert or another LLM, constitute a "single point of failure" and are vulnerable to targeted attacks like prompt

injection (Zou et al., 2023; Perez and Ribeiro, 2022) and susceptible to systemic biases (Bender et al., 2021; Liang et al., 2022). Compounding this issue is a severe **Scalability** bottleneck. The volume and combinatorial complexity of reasoning traces from modern LRMs, especially those employing branching search (Lightman et al., 2023; Yao et al., 2023a), make comprehensive manual verification practically and economically infeasible, a fact evidenced by the massive human effort required for existing process supervision datasets (Bai et al., 2022; Lightman et al., 2023). Furthermore, the **Opacity** of internal auditing processes at proprietary model providers erodes public trust and prevents independent verification of safety claims, conflicting with established principles of transparent reporting (Bommasani et al., 2023; Mitchell et al., 2019). In parallel, addressing opacity creates a critical tension with **Privacy**, since exposing complete reasoning traces for public audit risks the theft of valuable intellectual property through model distillation (Carlini et al., 2021) and increases the surface area for extracting sensitive training data (Nasr et al., 2023).

Addressing these simultaneous challenges of robustness, scalability, opacity, and privacy demands a new approach to the auditing paradigm. Our work is guided by the following research questions:

> **RQ 1.** *How can we design an auditing system that is robust to malicious participants and systemic bias without relying on a central trusted authority*

> **RQ 2.** *How can this system scale to audit complex reasoning traces while preserving the intellectual property of the model provider and ensuring public transparency*

Answering these questions naturally leads to a framework that integrates decentralized consensus, privacy-preserving protocols, and a novel representation for reasoning itself.

We introduce TRUST, a decentralized framework for auditing LLM reasoning (see Figure 1). To achieve **robustness**, TRUST establishes a consensus mechanism among a diverse, multi-tier set of auditors, drawing on principles from Byzantine Fault Tolerant systems (Castro et al., 1999; Lamport et al., 2019) to provably guarantee audit correctness even with a significant fraction of malicious participants. For **scalability**, the framework introduces a novel decomposition method that transforms reasoning traces into *Hierarchical Directed Acyclic Graphs (HDAGs)*, a structured representation that permits parallel verification of atomic reasoning steps by a distributed network. To jointly address **opacity** and **privacy**, all verification decisions are recorded on a

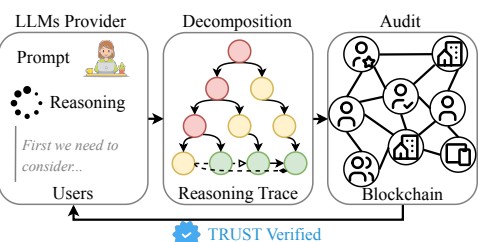

Figure 1: Reasoning traces (left) are decomposed into hierarchical segments (middle) and verified by a heterogeneous, multi-tier auditor network (right). Audit outcomes are recorded on-chain, while raw trace is stored off-chain for privacy.

transparent blockchain ledger for public auditability, while the protocol preserves confidentiality by distributing only partial, disconnected trace segments to individual auditors, protecting proprietary logic from reconstruction.

The design of TRUST is supported by rigorous theoretical guarantees for security and economic viability. We prove a *Safety-Profitability Guarantee* theorem, which formally ensures that under configurable statistical and economic conditions, the system achieves a target audit safety level while making honest participation profitable and malicious behavior result in a net loss. Our empirical validation spans multiple state-of-the-art LLMs (e.g., GPT-OSS (OpenAI, 2025), DeepSeek-r1 (DeepSeek-AI et al., 2025), Qwen (Yang et al., 2025)) and diverse reasoning tasks, and incorporates human-in-the-loop experiments with expert auditors to validate the multi-tier design. The results demonstrate that TRUST is highly effective at identifying reasoning flaws and is significantly more resilient to coordinated attacks than centralized baselines.

In summary, our main contributions are:

- We introduce TRUST, the first decentralized auditing system for reasoning traces that achieve privacy-preserving verification without exposing the proprietary model.

- We develop a systematic approach to decompose Chain-of-Thought reasoning into Hierarchy Directed Acyclic Graphs (HDAGs) that enable modular verification coupled with a multi-tier verification, routing simple problems to automated validators and complex problems to human experts.

- We develop theoretically grounded incentive mechanisms which ensure that honest auditors profit while malicious actor incur losses, providing the foundation necessary for sustainable real-world deployment at scale.

- We conduct comprehensive experiments on diverse datasets (e.g., MMLU-Pro, GSM8K) and models (e.g., GPT-OSS, DeepSeek-r1), including human-in-the-loop studies, to demonstrate the effectiveness and robustness of TRUST against centralized baselines.

## 2 RELATED WORKS

**Reasoning Model Verification.** Chain-of-Thought (CoT) prompting has revolutionized LLM reasoning by exposing intermediate steps (Wei et al., 2022), evolving into sophisticated tree-based search methods (Yao et al., 2023a) and Large Reasoning Models that treat reasoning as a primary objective (Jaech et al., 2024; Guo et al., 2025). However, these advances lack systematic verification mechanisms for generated reasoning traces, particularly for privacy-preserving and decentralized auditing.

**Auditing and Evaluation.** Current auditing approaches range from centralized "LLM-as-a-judge" methods (Zheng et al., 2023) to Process Reward Models that provide step-by-step supervision (Lightman et al., 2023). This line of work has been further refined by methods focusing on specific aspects of verification. For instance, Leng et al. (2025) proposes a rigorous auditing method for semi-structured reasoners, focusing on formal verification within a structured environment, while Lanham et al. (2023) concentrates on metrics to measure the faithfulness of chain-of-thought reasoning. While valuable, these approaches typically presume a centralized verifier and do not address scalable auditing via decentralized consensus. Recent work also addresses service-level integrity through cryptographic verification (Sun et al., 2025) and detection of model substitution (Cai et al., 2025). While addressing inference integrity, these approaches lack unified frameworks for scalable semantic auditing with decentralized consensus.

**Decentralized Verification.** Foundational work in Byzantine Fault Tolerant consensus (Castro et al., 1999) and Zero-Knowledge Proofs for ML (Chen et al., 2024; Sun et al., 2024) provides primitives for verifiable computation. The emerging field of Zero-Knowledge Machine Learning (ZKML) specifically aims to ensure the verifiability of ML models without disclosing sensitive data (Peng et al., 2025). However, existing approaches focus on computational correctness rather than semantic quality verification through human-in-the-loop consensus processes.

Our work synthesizes these directions by introducing the first framework for decentralized, privacy-preserving semantic auditing of reasoning traces at scale. Due to space limitation, we provide more comprehensive related works in Section B.

## 3 DECENTRALIZED AUDITING FOR LARGE REASONING MODELS

As illustrated in Figure 1, TRUST can integrate either human or LLMs to audit *faithfulness*, *harmlessness*, and *logical consistency* of chain-of-thought (CoT) reasoning. By operating on intermediate traces rather than final outputs alone, TRUST enables earlier and more comprehensive detection of reasoning flaws. TRUST features the following key innovations:

- **Batch & Segmentation.** Reasoning traces from multiple providers are *batched* to anonymize source identity and mitigate provider-specific bias. Traces are then *segmented* into minimal, auditable units and stored as content-addressed objects in decentralized storage. Segmentation protects proprietary logic: each auditor only sees the segment(s) they are assigned, preventing full-trace reconstruction.

- **Auditing & Consensus.** Heterogeneous auditors (computational checkers, LLMs, and humans) independently evaluate assigned segments. Votes are submitted via a cryptographic *commit–reveal* protocol: in the commit phase, auditors submit hashed votes; in the reveal phase, they disclose votes for verification against commitments. Segment-level quorums validate local steps; a trace-level aggregator combines weighted segment outcomes to reach the final decision.

- **Blockchain & Decentralized Storage.** A blockchain layer provides immutable audit trails and trustless consensus using a Proof-of-Stake (PoS)-style mechanism adapted for AI auditing. Smart contracts orchestrate session lifecycle, auditor assignment (by stake and expertise), commit–reveal voting, and performance-based rewards/slashing. Reasoning content is stored off-chain on IPFS; the blockchain records metadata, vote commitments, and final outcomes.

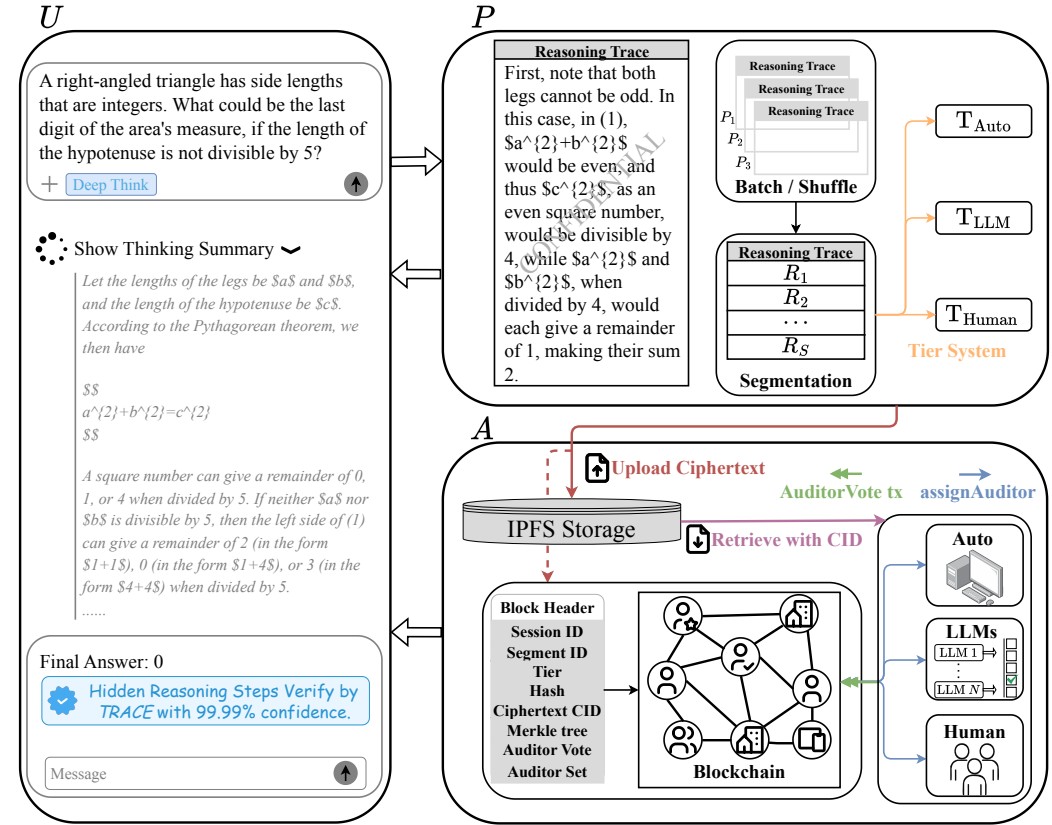

Figure 2: Overview of TRUST. The TRUST framework decomposes Chain-of-Thought reasoning traces into hierarchical directed acyclic graphs (DAGs) for modular verification across a three-tier auditor system. The process begins with a reasoning query (left panel) that generates intermediate reasoning steps, which are then decomposed into graph components and distributed across automated computers, LLM-based, and human auditors. TRUST utilizes IPFS for decentralized storage of reasoning traces and blockchain technology for immutable audit records, vote aggregation, and consensus mechanisms. Auditors verify reasoning segments independently, while cryptographic protocols ensure the privacy preservation of proprietary model internals. The final verification result provides confidence guarantees for reasoning trace faithfulness and correctness for the end user.

We formalize the three key participant parties in the TRUST ecosystem—**Provider**, **Auditor**, and **User**. TRUST supports both Business-to-Business (B2B) and Business-to-Consumer (B2C) settings, where the reasoning trace provider may be either a proprietary LLM vendor or an individual customer. We provide the illustration of the parties involved in TRUST in Figure 2.

- **Provider** ($P$): A proprietary LRM vendor or customer submitting reasoning traces for audit.
- **Auditor** ($A$): An anonymous seat (computer, LLM, or human) that verifies assigned segments.
- **User** ($U$): An end-user who consumes audited outputs and provenance via APIs or dashboards.

The practical deployment of TRUST hinges on its economic viability. Our framework is designed to be agnostic to the underlying blockchain technology, enabling integration with emerging high-throughput, low-cost Layer-2 solutions to mitigate transaction latency and costs (i.e., gas fees). In our model, these transaction costs are considered an operational expense for the Provider requesting the audit, ensuring the system remains sustainable and economically practical for the auditors who form the backbone of the network.

Given a reasoning trace with $S$ segments (including CoT and tool calls), TRUST maps the trace to a *Hierarchical Directed Acyclic Graph (HDAG)* with five abstraction levels: *Goal*, *Strategy*, *Tactic*, *Step*, and *Operation*. This representation is *problem-agnostic* (math, science, programming, general reasoning, etc.) and enables scalable, parallel verification because most nodes are independently auditable. Each node carries metadata (ID, summary, complexity, auditor type, and dependencies), and edges encode relationships (*decomposes_to*, *depends_on*, *enables*, *validates*, *contradicts*, etc.). Formally,

each segment $s \in \{1, \ldots, S\}$ is assigned to a primary auditor type $\in \{\mathbf{C}\text{omputer}, \mathbf{LLM}, \mathbf{H}\text{uman}\}$:

$$\underbrace{\text{Segment 1}}_{\mathbf{C}\text{omputer}}, \quad \underbrace{\text{Segment 2}}_{\mathbf{LLM}}, \quad \underbrace{\text{Segment 3}}_{\mathbf{H}\text{uman}}, \ldots, \quad \underbrace{\text{Segment } S}_{\text{type}\in\{\mathbf{C},\mathbf{L},\mathbf{H}\}}.$$

**Hierarchical Directed Acyclic Graphs (HDAGs).** Prior work on CoT decomposition, such as DLCoT, introduced automatic frameworks for breaking down long reasoning traces into structured segments, primarily to generate high-quality data for model distillation (Luo et al., 2025). These works observe that CoTs can follow linear, tree, or more general network structures. DLCoT, for instance, applies macro-structure parsing to divide CoTs into four parts—*Problem Restatement*, *Approach Exploration*, *Verification*, and *Summary*—before further segmenting the approach and verification stages into stepwise units. Other lines of research (Kothapalli et al., 2025) focus on extracting causal structures from token-level processing functions.

In contrast, we propose a general, problem-agnostic approach: decomposing CoTs into *Hierarchical Directed Acyclic Graphs* (HDAGs). Our hierarchy consists of five abstraction levels: *Goal*, *Strategy*, *Tactic*, *Step*, and *Operation*. This abstraction provides two key advantages. First, it is broadly applicable across domains—mathematics, science, engineering, and open-domain reasoning. Second, it enables scalable verification, since most nodes are independently auditable and can be naturally mapped to different auditor types (e.g., computer programs, LLMs, or human experts). An illustration is provided in Figure 3, where each node is annotated by difficulty and type (basic reasoning step, tool usage, or fact/premise). This hierarchical decomposition mirrors neural circuits in the frontal cortex, which process reasoning through multi-level evidence integration (Sarafyazd and Jazayeri, 2019). Just as the brain organizes reasoning hierarchically rather than linearly, our HDAG design enables different reasoning components to be audited at the appropriate granularity. Edges capture logical relationships between nodes, including *dependencies* (depends on, enables), *structural links* (decomposes, refines), *validation* (validates, exemplifies), and *conflicts* (contradicts).

Concretely, TRUST constructs HDAGs in five steps:

**Step 1: Identify Abstraction Levels.** The raw problem statement, reasoning trace (with tool usage), and final output are parsed into semantic hierarchy levels.

**Step 2: Segment Within Each Level.** Each level is further divided into granular units with associated metadata (IDs, complexity, summaries). Difficulty annotations guide later auditor assignment.

**Step 3: Extract Relationships.** Logical dependencies between segments are mapped into relations (*decomposes_to*, *depends_on*, *enables*, *validates*, etc.).

**Step 4: Assign Auditor Types.** Segments are routed to auditor types from {Human, Computer, LLM}, based on complexity and modality.

**Step 5: Refine and Construct HDAG.** Segments and relationships are synthesized into a final auditable HDAG, with quality assurance checks.

Further implementation details and examples are provided in Sections C.1, C.3 and C.4.

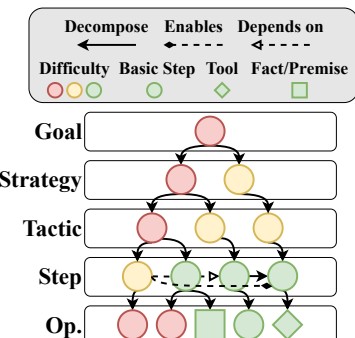

Figure 3: Example of an HDAG. Node color indicates the level of difficulty assigned to different types of auditors. Node shape denotes the type of step and edges represent relationships between nodes.

**Auditing & Consensus.** Reasoning traces vary in both complexity and type of reasoning step. In TRUST, each segment is routed to the most appropriate auditor type—*Human*, *LLM*, or *Computer*—to ensure accurate and efficient verification. Decentralized participants either contribute computational resources for deterministic checks (e.g., arithmetic, tool calls) or provide expertise in evaluating semantic coherence and faithfulness. This heterogeneous auditor pool improves both accuracy (by matching segments to suitable verifiers) and robustness (by reducing vulnerability to malicious or biased auditors). We analyze the consensus process at three levels: *seat*, *segment*, and *trace*.

1. **Seat layer.** Within a segment $s$, each of the $k_{t(s)}$ auditor seats votes independently. Computer seats are assumed noiseless, while LLM and human seats have nonzero error rates $\epsilon_t$. Human seats may additionally be adversarial with probability $\rho_{\mathrm{H}}$.

2. **Segment layer.** For the segment $s$, define the segment pass indicator $B_s = \mathbf{1}\big[\#\{\text{correct votes}\} \geq q_{t(s)}\big]$, where $q_t = \lceil \tau\, k_t \rceil$ is the quorum threshold for type $t$. The exact pass probability for a segment of type $t$ with parameters $(k_t, \epsilon_t, \rho_t)$ (with $\rho_C = \rho_L = 0$) is

$$p_t = \Pr[B_s = 1] = \sum_{m=0}^{k_t} \binom{k_t}{m} \rho_t^m (1-\rho_t)^{k_t-m} \sum_{c=q_t}^{k_t-m} \binom{k_t-m}{c} (1-\epsilon_t)^c\, \epsilon_t^{k_t-m-c}, \quad (3.1)$$

where $m$ malicious seats vote incorrectly, and among the $k_t - m$ honest seats, $c$ cast correct votes.

3. **Trace layer.** To aggregate across all $S$ segments, we assign weights $w_{t(s)}$ and define $W = \sum_{s=1}^{S} w_{t(s)}\, B_s$, $\quad W_\beta = \beta \sum_s w_{t(s)}$. We then bound the failure probability $\Pr[W < W_\beta]$ using Hoeffding and Chernoff inequalities:

$$\Pr[W < W_\beta] \leq \underbrace{\exp\!\Big[-2\,(\mu_{\text{vote}} - W_\beta)^2/\sigma_{\max}^2\Big]}_{\text{Hoeffding}} \wedge \underbrace{\min_{\lambda>0} \exp\!\Big(\lambda W_\beta + \sum_{s=1}^{S} \ln\!\big(p_s\, e^{-\lambda w_s} + (1-p_s)\big)\Big)}_{\text{Chernoff}}.$$

$$(3.2)$$

Figure 4 compares these bounds with the exact solution under representative parameters ($\epsilon_C = 0$, $\epsilon_L = 0.05$, $\epsilon_H = 0.30$, $\rho_H = 0.1$). The full derivation on seat, segment, and trace levels results are provided in Section D.

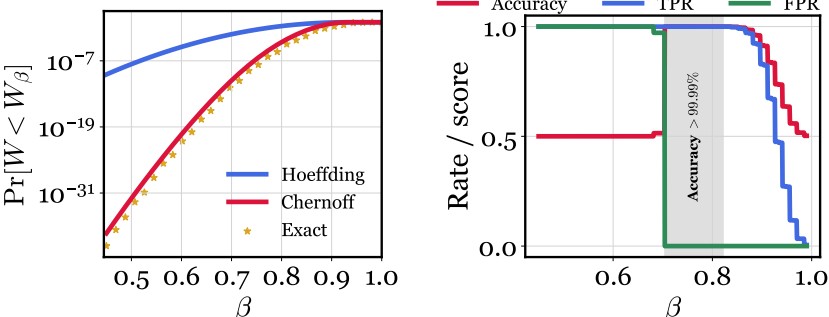

Figure 4: The parameters are $\epsilon_C = 0$, $\epsilon_L = 0.05$, $\epsilon_H = 0.30$, and $\rho_H = 0.1$. (Left) Comparison of probability of failure of Hoeffding and Chernoff bounds and exact solution in (3.2) as a function of trace-level quorum threshold $\beta$. (Right) The true positive rate (TPR), false positive rate (FPR), and accuracy with different values of trace-level quorum threshold. The grey shaded area indicates the width of the trace-level quorum that achieves greater accuracy than $99.99\%$.

**Privacy by Design.** The privacy-preserving nature of TRUST is an intrinsic property of its structural design. By decomposing the reasoning trace into an HDAG, we compartmentalize the verification process. Each auditor is assigned only one or more atomic segments of the trace, without access to the complete context or the final conclusion. This "need-to-know" basis ensures that the full proprietary logic of the reasoning process is never exposed to any single party, thus preventing intellectual property theft or model distillation. The on-chain records are limited to cryptographic commitments of these segments and their verification outcomes, serving as immutable proof of work while keeping the reasoning content itself off-chain on the InterPlanetary File System (IPFS) and private.

### 3.1 ECONOMICS ANALYSIS

In this section, we provide the economic analysis of the TRUST framework on reputation, slashing, reward, statistical, and economic guarantees.

**Reputation-Weighted Slashing and Rewards.** Each human auditor seat $i$ maintains a reputation score $r_i(t) \in [0,1]$, updated after every segment as $r_i(t+1) = (1-\gamma)\, r_i(t) + \gamma\, \mathbf{1}[\text{vote correct}]$, where $\gamma \in (0,1]$ controls adaptation speed. Incorrect votes trigger a slashing probability $p_{\text{slash}}(r) = p_{\min} + (p_{\max} - p_{\min})(1-r)$, with $0 < p_{\min} < p_{\max} \leq 1$, penalizing low-reputation seats more heavily. The per-segment payoff $X_i \in \{-P, 0, R\}$ is defined as: $R$ for a correct vote, $0$ for an incorrect vote without slashing, and $-P$ for a slashed incorrect vote. For an honest seat with error rate $\epsilon_H$, the expected payoff is $\mu_H(r) := \mathbb{E}[X_i] = (1-\epsilon_H)R - \epsilon_H\, P\, p_{\text{slash}}(r)$.

Take parameters $R = 6$, $P = 8$, $p_{\min} = 0.2$, $p_{\max} = 0.5$, $\delta = 0.2$, $\lambda = 60$, and $\epsilon_H = 0.30$ for example, an honest seat achieves an expected per-segment payoff of $\mu_{\min} = 0.7 \times 6 - 0.3 \times 8 \times 0.5 = 3.0$, with variance $\sigma_H^2 = 25.8$ and worst-case increment $b = 6$. A malicious seat, by contrast, suffers an expected loss of $\mathbb{E}[X_{\mathrm{mal}}] = -0.5 \times 8 = -4.0$, with variance 16 and worst-case increment $b = 8$. Over a 24-hour window ($T = 24$) with 1440 seg-

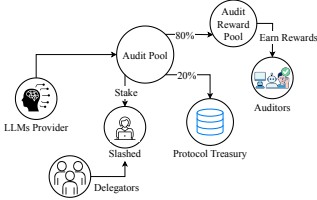

Figure 5: Tokenomics of TRUST.

ments, tail bounds from Theorem D.1 show that the probability of an honest auditor ending with nonpositive payoff is at most $\exp\big(-\big(60 \times 24 \times 3^2\big) / \big(2 \times 25.8 + (2/3) \times 6 \times 3\big)\big) \approx e^{-204} < 10^{-80}$, while the probability of a malicious auditor breaking even or better is at most $\exp\big(-\big(60 \times 24 \times (0.2 \cdot 8)^2\big) / \big(2 \times 25.8 + (2/3) \times 6 \times 1.6\big)\big) \approx e^{-63.6} < 10^{-27}$.

**Tokenomics.** The tokenomics of TRUST (Illustrated in Section 3.1) ensures incentive-compatible decentralized auditing. Provider fees are split into a *Protocol Treasury* (20%) for long-term sustainability and an *Audit Pool* (80%) for immediate auditor rewards. Auditors and delegators stake tokens to participate; honest auditors aligning with consensus receive rewards, while dishonest ones are slashed, creating strong deterrents against manipulation.

Specifically, Delegators act as capital providers within the ecosystem, staking their assets with auditors who have a strong track record of reliable verification. In return, they earn a percentage of the auditor's rewards. This symbiotic relationship allows skilled auditors to increase their stake-backed influence and auditing capacity, while enabling token holders to productively deploy their capital, thereby enhancing the overall security and robustness of the network. The system's resilience against collusion, including potential dishonest collaboration between delegators and auditors, is maintained through our decentralized consensus protocol and dynamic trust score. Any deviation from the consensus outcome results in financial penalties (slashing) for the auditor, and consequently, their delegators, thus creating a strong economic disincentive for such behaviors.

## 4 EXPERIMENTAL VERIFICATION

In this section, we provide verification on TRUST on the annotated CoT dataset, open-source model-generated CoTs for de-bias, and safety and privacy results.

### 4.1 CORRECTNESS AND FAITHFULNESS.

We use 200 samples from the MMLU-Pro-CoT-Train dataset (Lab, 2024), which provides ground truth annotations for individual reasoning steps and final answers. This allows us to systematically evaluate the correctness and faithfulness of audits at both the step and trace levels. We compare TRUST against centralized approaches, including (i) single-LLM auditors (DeepSeek-R1-8B, Qwen2.5-7B, Mistral-7B, GPT-OSS-20B, LLaMA-3B) and (ii) ensemble-based voting schemes (majority, supermajority, weighted, unanimous). To stress-test robustness, we simulate auditor corruption by systematically flipping a proportion of segment-level votes, with corruption rates ranging from 5% to 20%.

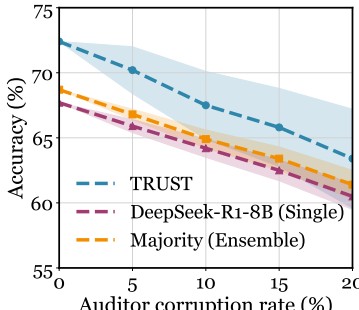

Figure 6 and Table 1 summarize the results. At baseline (no corruption), TRUST achieves the highest accuracy (72.4%), outperforming both single auditors (e.g., DeepSeek-R1-8B at 67.7%) and ensemble methods (e.g., majority voting at 68.7%). As corruption increases, all methods degrade, but TRUST degrades more gracefully: accuracy remains above 63% even at 20% corruption, while centralized ensembles drop below 61% and single auditors fall closer to 60%. The performance gap widens with higher corruption rates, highlighting TRUST's resilience to adversarial or biased auditors.

Figure 6: Correctness of Single, Ensemble (Centralized) with decentralized TRUST framework.

Table 1: Performance comparison of TRUST (decentralized) vs. centralized approaches across corruption rates. Best is **bold**, second-best is underlined.

| Category | Method | Baseline | 5% Corr. | 10% Corr. | 15% Corr. | 20% Corr. |
|---|---|---|---|---|---|---|
| **Decentralized** | TRUST | **72.4**$_{\pm 0.0}$ | **70.2**$_{\pm 1.8}$ | **67.5**$_{\pm 2.6}$ | **65.8**$_{\pm 3.0}$ | **63.4**$_{\pm 3.8}$ |
| | *Ensemble Models* | | | | | |
| | Majority Voting | $\underline{68.7}_{\pm 0.0}$ | $\underline{66.8}_{\pm 0.4}$ | 64.9$_{\pm 0.7}$ | 63.4$_{\pm 0.9}$ | 61.4$_{\pm 1.1}$ |
| | Supermajority | $\underline{68.7}_{\pm 0.0}$ | 66.8$_{\pm 0.5}$ | $\underline{65.0}_{\pm 0.7}$ | $\underline{63.2}_{\pm 0.9}$ | $\underline{61.2}_{\pm 0.9}$ |
| | Weighted Voting | 68.1$_{\pm 0.0}$ | 66.4$_{\pm 0.6}$ | 64.5$_{\pm 0.7}$ | 62.7$_{\pm 1.1}$ | 60.9$_{\pm 0.9}$ |
| **Centralized** | Unanimous | 45.6$_{\pm 0.0}$ | 46.1$_{\pm 0.6}$ | 46.5$_{\pm 0.9}$ | 46.8$_{\pm 1.1}$ | 47.4$_{\pm 1.0}$ |
| | *Single Models* | | | | | |
| | DeepSeek-R1-8B | 67.7$_{\pm 0.0}$ | 65.9$_{\pm 0.5}$ | 64.2$_{\pm 0.7}$ | 62.5$_{\pm 0.8}$ | 60.5$_{\pm 1.0}$ |
| | Qwen2.5-7B | 67.4$_{\pm 0.0}$ | 65.7$_{\pm 0.6}$ | 64.1$_{\pm 0.7}$ | 62.1$_{\pm 0.9}$ | 60.5$_{\pm 1.0}$ |
| | Mistral-7B | 66.8$_{\pm 0.0}$ | 65.2$_{\pm 0.6}$ | 63.6$_{\pm 0.8}$ | 61.8$_{\pm 1.1}$ | 60.1$_{\pm 1.1}$ |
| | DeepSeek-R1-1.5B | 64.1$_{\pm 0.0}$ | 62.9$_{\pm 0.5}$ | 61.2$_{\pm 0.8}$ | 59.7$_{\pm 1.1}$ | 58.5$_{\pm 0.8}$ |
| | GPT-OSS-20B | 63.8$_{\pm 0.0}$ | 62.5$_{\pm 0.6}$ | 60.9$_{\pm 0.7}$ | 59.7$_{\pm 1.1}$ | 58.4$_{\pm 1.0}$ |
| | LLaMA-3B | 52.1$_{\pm 0.0}$ | 51.9$_{\pm 0.6}$ | 51.7$_{\pm 0.7}$ | 51.4$_{\pm 1.0}$ | 51.3$_{\pm 1.1}$ |

**Output-Based Auditing is Insufficient.** As shown in Table 2, output-only evaluation drops accuracy by 18%–47% across all models compared to full-CoT auditing. This confirms that correct final answers often mask flawed reasoning that only semantic auditing can detect.

Table 2: Baseline accuracy under Full-CoT vs. Output-Based evaluation.

| | TRUST | R1-8B | Qwen-7B | Mistral-7B | R1-1.5B | OSS-20B | LLaMA-3B |
|---|---|---|---|---|---|---|---|
| Full-CoT | **72.4** | 67.7 | 67.4 | 66.8 | 64.1 | 63.8 | 52.1 |
| Output-Based | — | 36.0 | 34.0 | 20.0 | 22.0 | 46.0 | 34.0 |

## 4.2 SAFETY AND PROFITABILITY

A central design goal of TRUST is to guarantee both *statistical safety*—ensuring that the probability of a failed audit remains vanishingly small—and *economic sustainability*—ensuring that honest auditors are consistently rewarded while malicious ones suffer provable long-term losses. Figure 7 illustrates these dynamics empirically. On the left of Figure 7, reputation scores naturally separate: honest auditors are reinforced with high reputation, while malicious and random guessers quickly lose credibility. On the right, profit trajectories diverge: honest participants earn steadily increasing rewards, while guessers and malicious seats accumulate losses due to repeated slashing. These empirical trends are formally supported by the *Safety–Profitability Guarantee* (Theorem D.1), which proves that, under appropriate statistical and economic parameters, honest auditors almost surely remain profitable while malicious participants incur provable long-term losses. The detailed derivation of these guarantees is provided in Section D.

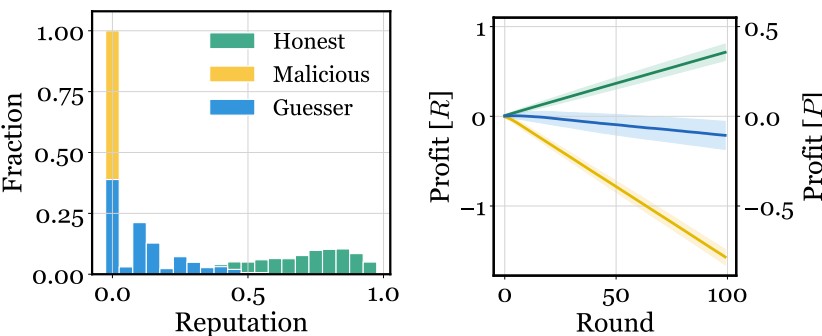

Figure 7: The parameters $\epsilon_C = 0$, $\epsilon_L = 0.05$, $\epsilon_H = 0.30$, and $\rho_H = 0.1$. (Left) Repuation scores. (Right) Profit curves.

**Theorem 4.1** (Safety–Profitability (Informal version of Theorem D.1)). *Fix horizon $T$, target failure $\epsilon_{\text{target}}$, and $\delta \in (0, 1)$.*

- **Statistical dial:** ensure $\mu_{\text{vote}} - W_\beta \geq \sqrt{\frac{1}{2}\sigma_{\text{vote}}^2 \ln \frac{\lambda T}{\epsilon_{\text{target}}}}$.

- **Economic dial:** set $(R, P, p_{\min}, p_{\max})$ with $\mu_{\min} := (1 - \epsilon_H)R - \epsilon_H P p_{\max} > 0, \quad p_{\min} \geq \frac{\delta}{1-\alpha}$.

Then:

(a) **Safety:** $\Pr[\text{fail in } [0, T]] \leq \epsilon_{\text{target}}$.

(b) **Honest profit:** $\Pr[U_{\text{hon}}(T) \leq 0] \leq \exp\left(-c_1 T \mu_{\min}^2\right)$.

(c) **Malicious loss:** $\Pr[U_{\text{mal}}(T) \geq 0] \leq \exp\left(-c_2 T (\delta P)^2\right), \quad \mathbb{E}[U_{\text{mal}}(T)] \leq -\lambda T \delta P$.

### 4.3 BIAS MITIGATION.

Auditing systems are vulnerable to bias, where auditors may favor reasoning traces produced by their own model family or penalize outputs from competing models. This creates two common failure modes: (i) *self-favoritism*, where a model systematically approves its own reasoning, and (ii) *self-criticism*, where a model disproportionately rejects its own outputs.

TRUST is designed to mitigate such bias through three architectural features: (1) *Segment-level decomposition*, which breaks reasoning traces into atomic units; (2) *Multi-tier consensus*, combining human, LLM, and automated auditors; and (3) *Anonymous evaluation*, hiding the source model of each segment.

We construct a benchmark of 200 questions across four domains. For each, CoT traces from DeepSeek-R1-1.5B and GPT-OSS-20B are evaluated under three regimes: single auditors, ensemble auditors, and TRUST. In Table 3, single auditors vary in accuracy (15.2–60.9%) but show bias (avg. +5.5). Ensembles remove bias but perform poorly (16.9–30.5%). TRUST breaks this tradeoff, achieving higher accuracy (34.1%) without bias.

Table 3: Comparison of auditing methods on reasoning trace verification. Accuracy in %. Bias score = (self-approval)−(other-approval). Positive bias = favoritism, Negative bias = criticism.

| Method | Acc. (%) | Bias |
|---|---|---|
| **TRUST (Decentralized)** | **34.1** | **–** |
| **Ensemble** | | |
| Supermajority | 30.5 | – |
| Majority | 26.8 | – |
| Weighted | 16.9 | – |
| **Single LLMs** | | |
| GPT-OSS-20B | 60.9 | +44.4 |
| Qwen-7B | 38.4 | – |
| DeepSeek R1-1.5B | 21.9 | −11.3 |
| Llama-3B | 16.2 | – |
| Mistral-7B | 15.9 | – |
| DeepSeek R1-8B | 15.2 | – |
| **Average** | **28.1** | **+5.5** |

### 4.4 HUMAN EXPERIMENT ON THREE-TIER AUDITING WITH TRUST VARIANTS

To evaluate TRUST in a multi-tier auditing setting, we recruited 30 computer science students as human auditors (Example provided in Section C.2). We constructed a benchmark of 30 math problems from the `openai/gsm8k` dataset (Cobbe et al., 2021b), generating CoT traces with DeepSeek-R1-8B and GPT-OSS-20B. Auditors participated within the TRUST framework alongside LLM and automated auditors, and we compared performance against both single-LLM auditors and a centralized human-only audit.

Table 4 reports F1 and Brier scores. Single LLM auditors perform poorly, reflecting both limited correctness when providing a whole trace to audit. Centralized human auditors achieve moderately stronger results (F1 = 0.85, Brier = 0.21) and require approximately 1.4 minutes per trace on average (medium < 1 minute). While they clearly outperform single LLMs auditors, they remain time-consuming and are susceptible to systematic biases and malicious behavior. In contrast, TRUST achieves a much higher F1 of 0.89 and a substantially lower Brier score of 0.074, demonstrating both accurate and well-calibrated auditing. These results highlight the effectiveness of combining human, LLM, and automated auditors under TRUST, outperforming both centralized human audits and single-model baselines. For TRUST variants, random segment break CoT randomly between 2-6 sentences; fixed-length short segment breaks into two sentences per segment; fixed-length long segment breaks into 5-7 sentences per segment. All three variants are using GPT-OSS-20B.

### 4.5 LATENCY

In Figure 8, we present a latency analysis, decomposing TRUST's overhead into constituent components and comparing it against centralized baselines.

Table 4: Human Experiment on Three-Tier Auditing: Performance comparison across single LLM auditors, centralized human auditing, and TRUST variants. 30 Computer Science students audited 30 GSM8K math problems with CoT traces from DeepSeek-R1-8B and GPT-OSS-20B. Metrics are F1 score (higher better) and Brier score (lower better). Single LLMs perform poorly, centralized human audit achieves F1=0.85, while TRUST w/ HDAG substantially outperforms all baselines (F1=0.89, Brier=0.074) through multi-tier consensus. TRUST variants using random or fixed-length segmentation show degraded performance (F1=0.40), validating the importance of hierarchical decomposition.

| Metric | Single LLM Auditors | | | | | | | TRUST Variants | | | |
| --- | --- | --- | --- | --- | --- | --- | --- | --- | --- | --- | --- |
| | DeepSeek R1-8B | Qwen2.5 7B | Mistral 7B | DeepSeek R1-1.5B | GPT-OSS 20B | LLaMA 3B | Human Audit | TRUST w/ HDAG | TRUST w/ Random Seg | TRUST w/ Fix Seg (Short) | TRUST w/ Fix Seg (Long) |
| F1 | 0.50 | 0.50 | 0.50 | 0.40 | 0.50 | 0.30 | 0.85 | **0.89** | 0.40 | 0.40 | 0.40 |
| Brier Score | 0.500 | 0.486 | 0.500 | 0.500 | 0.544 | 0.890 | 0.21 | **0.074** | 0.49 | 0.49 | 0.49 |

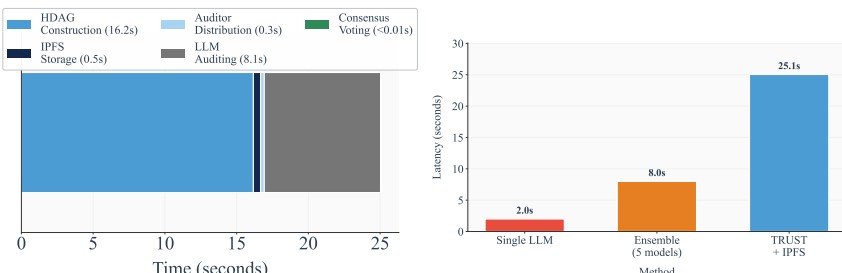

Figure 8: End-to-end latency analysis of TRUST framework. (Left) Breakdown of latency components in the TRUST pipeline for averaging 100 math problem). (Right) Comparison of total audit latency. The additional latency provides robustness guarantees through distributed segment auditing.

## 5 CONCLUSION AND FUTURE WORKS

We introduce TRUST, the first decentralized framework for auditing the reasoning traces of Large Reasoning Models that simultaneously addresses robustness, scalability, opacity, and privacy challenges. TRUST offers an end-to-end pipeline that integrates three key components: a Hierarchical Directed Acyclic Graph (HDAG) decomposition method that breaks Chain-of-Thought reasoning into five abstraction levels; a multi-tier consensus mechanism that routes verification tasks to automated checkers, LLMs, and human experts based on complexity; and a blockchain-based infrastructure with cryptographic privacy preservation that ensures transparent audit trails while protecting proprietary model internals. It supports verification across diverse reasoning domains and enhances transparency through decentralized consensus and immutable audit records.

Our experiments demonstrate TRUST's effectiveness across correctness, bias mitigation, and human-in-the-loop evaluation using multiple datasets and state-of-the-art models. TRUST consistently outperforms centralized ensemble methods and single auditors while maintaining graceful degradation under adversarial conditions. These results highlight its robustness against corruption and effectiveness in eliminating systematic bias while preserving accuracy.

In addition, our theoretical framework provides formal guarantees that honest auditors profit while malicious actors incur losses, creating sustainable economic incentives for real-world deployment. Overall, TRUST pioneers decentralized AI auditing as a practical pathway toward safe and accountable deployment of reasoning-capable AI systems in high-stakes domains. It makes transparent oversight of proprietary AI systems accessible without compromising intellectual property rights.

**Future work** will involve developing more sophisticated graph decomposition methods to capture richer reasoning dependencies, integrating adaptive auditor assignment strategies that leverage task-specific expertise, and extending the framework to support dynamic, interactive reasoning settings. Furthermore, we plan to conduct a longitudinal analysis of the on-chain data generated by TRUST to study the long-term economic dynamics and emergent behaviors of the auditor network. Evaluating the framework's performance across different blockchain infrastructures would also provide deeper insights into its practical scalability and cost-effectiveness. We also plan to explore cross-model reasoning consistency verification for multi-agent scenarios and investigate integration with federated learning frameworks to accelerate trustworthy AI deployment.

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

# Appendix

## A  THE USE OF LARGE LANGAUGE MODELS (LLMs)

To enhance clarity and readability, this paper utilized Large Language Models exclusively as language polishing tools. Their role was confined to general proofreading and writing assistance—functions analogous to those provided by traditional grammar checkers and dictionaries. These tools did not contribute to the generation of new scientific content or ideas, and their usage is consistent with standard practices for manuscript preparation.

# B    MORE RELATED WORKS

## B.1    CHAIN-OF-THOUGHT REASONING

Exposing intermediate reasoning steps, popularized by Chain-of-Thought (CoT) prompting (Wei et al., 2022), has become a cornerstone for enhancing the capabilities of Large Language Models (LLMs) (Kojima et al., 2022; Wang et al., 2022; Nye et al., 2021; Zhou et al., 2022; Zelikman et al., 2022). However, this paradigm has rapidly evolved, and the resulting complexity and length of reasoning traces present a critical, unsolved challenge for verification (Lightman et al., 2023; Jacovi et al., 2024; Ling et al., 2023; Chen et al., 2025). Initial research focused on eliciting reasoning and moving beyond simple linear chains to more structured representations (Creswell et al., 2022; Zhou et al., 2022; Besta et al., 2024; Chen et al., 2022). For instance, Tree-of-Thoughts (ToT) generalized CoT to a tree search, enabling explicit exploration and backtracking within the reasoning process (Yao et al., 2023a). Concurrently, a parallel line of work enabled models to offload complex calculations and external queries to tools, separating logical deduction from information retrieval (Schick et al., 2023; Yao et al., 2023b; Gao et al., 2023; Nakano et al., 2021). More recent efforts have focused on Large Reasoning Models (LRMs) that treat reasoning as a primary objective, allocating substantial computational resources to the process (Jaech et al., 2024; Guo et al., 2025; NVIDIA et al., 2025). These models are trained with large-scale reinforcement learning from process-level feedback and employ verifiers to guide multi-path search during inference (Huang and Chang, 2022). Despite these advances in generating sophisticated reasoning, the mechanisms for *auditing* these processes in a scalable, privacy-preserving, and decentralized manner have remained largely unexplored (Peng et al., 2025). In this paper, we address this gap by introducing a framework that decomposes complex reasoning traces into auditable graphs (HDAGs) and leverages a decentralized network for verification, making it suitable for the proprietary and intricate nature of modern LRMs.

## B.2    AUDITING AND EVALUATION OF LLM REASONING

As the complexity of reasoning in LLMs increases, methods for auditing its quality, faithfulness, and safety have become a critical area of research (Liang et al., 2022; Lanham et al., 2023). An initial and widely adopted approach relies on a centralized "LLM-as-a-judge," which, while scalable, is known to suffer from inherent biases and constitutes a single point of failure (Zheng et al., 2023). A significant advancement was the shift from auditing final outcomes to verifying the reasoning *process* itself, primarily through Process Reward Models (PRMs) that provide step-by-step supervision (Lightman et al., 2023; Uesato et al., 2022; Khalifa et al., 2025). The limitations of this approach were subsequently highlighted by the discovery of unfaithful reasoning, where models generate plausible-looking steps that do not reflect their true decision-making process, underscoring the need for audits to defend against strategic deception (Turpin et al., 2023). More recently, research has expanded to address service-level integrity in opaque commercial settings, including work on detecting model substitution and using cryptographic methods to verify the integrity of the inference process itself (Shi et al., 2024; Cai et al., 2025; Pasquini et al., 2025; South et al., 2024). For example, CoIn introduces a method to audit "invisible" reasoning tokens by using hash trees over embeddings, ensuring providers do not inflate billing without revealing proprietary content (Sun et al., 2025). While these works address vital concerns like inference integrity or billing, they do not offer a unified framework for scalable *semantic* auditing that combines decentralized consensus to mitigate bias with economic incentives to ensure network security. This work synthesizes these needs by proposing a framework that performs process-level semantic auditing on partial traces, uses a decentralized multi-tier auditor network to ensure robustness, and is secured by theoretically-grounded economic guarantees.

## B.3    DECENTRALIZED CONSENSUS AND PRIVACY-PRESERVING VERIFICATION

Building systems that ensure integrity and privacy without a central trusted party has a long-standing foundation in decentralized consensus, cryptography, and confidential computing (Castro et al., 1999; Sabt et al., 2015; Costan and Devadas, 2016). These technologies provide the essential primitives for building auditable systems for proprietary models, where model internals and user data must remain confidential. Foundational work established Byzantine Fault Tolerant (BFT) consensus for robust agreement and public ledgers for tamper-evident records (Castro et al., 1999; Wood et al., 2014; Yin et al., 2019), while Trusted Execution Environments (TEEs) provided hardware-based confidential compute (Sabt et al., 2015). A prominent line of recent research focuses on applying these

primitives directly to language models, particularly using Zero-Knowledge Proofs (ZKPs) to create verifiable attestations of model inference without revealing the model's weights (Chen et al., 2024; Liu et al., 2021; Ghodsi et al., 2017). Specifically, zkLLM (Sun et al., 2024) introduced specialized cryptographic techniques to efficiently prove the correctness of attention mechanisms, demonstrating that even complex transformer computations can be cryptographically verified. Another approach uses TEEs to run auditable safety benchmarks on proprietary models, publishing hardware-signed attestations to a public log to prove that a specific model version passed certain tests (Schnabl et al., 2025). While this work is crucial for verifying *that* a specific computation was performed correctly by a specific model, it does not address the orthogonal problem of verifying the *semantic quality, faithfulness, and safety* of the reasoning trace through a scalable, human-in-the-loop consensus process. Our work provides a novel synthesis by using a public ledger and consensus not just for recording outcomes, but for orchestrating a decentralized, multi-tier network of auditors and managing economic incentives, creating the first system designed for end-to-end semantic auditing of model reasoning.

## C  EXPERIMENTAL DETAILS

### C.1  TRUST IMPLEMENTATION DETAILS

This section provide the LLMs instruction of our auto data processing framework and the example output from gpt-oss-120b.

**Example CoT**  Below is a example CoT trace.

```
### Input Format:
## Problem Statement:
A construction company was building a tunnel. When $\frac{1}{3}$ of the
↪ tunnel was completed at the original speed, they started usin
g new equipment, which increased the construction speed by $20 \%$ and
↪ reduced the working hours to $80 \%$ of the original. As a resu
lt, it took a total of 185 days to complete the tunnel. If they had not
↪ used the new equipment and continued at the original speed, it
 would have taken $\qquad$ days to complete the tunnel.

## Thinking Process:
We have a tunnel. Original speed = some rate (units per day). They
↪ built 1/3 of tunnel at original speed. Then they switched to new
equipment: speed increased by 20%, i.e., new speed = 1.2 * original
↪ speed. Working hours reduced to 80% of original, meaning per day
they work less time, so effective daily output? If originally they
↪ worked some hours per day, now they work only 80% of that, but
speed is per hour? Let's interpret: Original speed in terms of distance
↪ per hour? Or maybe speed per day? Usually speed is distance
per hour. If they increase speed by 20% (i.e., per hour they do 1.2
↪ times) but they work only 80% of the hours per day, then daily
production = speed_per_hour * hours_per_day. If original daily
↪ production = v * h (where v is original speed per hour, h original
hours per day). New daily production = (1.2 v) * (0.8 h) = 0.96 v h =
↪ 0.96 * original daily production. So overall daily rate
actually decreased by 4% compared to original! That seems odd: they
↪ increased speed but reduced hours, net 0.96 factor. But maybe
they meant speed (i.e., daily output) increased by 20% and hours
↪ reduced to 80%, i.e., effective speed per day is 1.2*0.8=0.96 of
original daily work, which is slower. But they say "As a result, it
↪ took a total of 185 days to complete the tunnel." That implies
slower after switching? The initial third took some days, then
↪ remaining two-thirds took more days than original would have. Let's
compute.

Let total tunnel length = 1 (normalized). Original daily output = D
↪ (units per day). So original time to finish whole tunnel = 1/D
days.

They did first 1/3 at original speed: time t1 = (1/3)/D = 1/(3D).

After switch, new daily output = 0.96 D (as above). Remaining work =
↪ 2/3. Time t2 = (2/3) / (0.96 D) = (2/3) * (1/(0.96 D)) = (2)/(3
* 0.96 D) = (2)/(2.88 D) = (0.694444...)/D? Let's compute precisely:
↪ 2/(2.88) = 0.694444... So t2 = 0.694444... / D.

Total days = t1 + t2 = 1/(3D) + 0.694444.../D = (1/3 + 0.694444...)/D =
↪ (0.333333... + 0.694444...)/D = 1.027777... / D.

Given total days = 185. So (1.027777...)/D = 185 => D = 1.027777... /
↪ 185 = approx 0.005555... Wait compute: 1.0277777778 /185 =
0.005555556... Indeed 0.0055556 per day. That seems like original daily
↪ output = 0.0055556 of tunnel per day, i.e., whole tunnel
would take 1/D = 180 days. Let's check: 1/D = 180. Exactly? Since D =
↪ 1/180 = 0.00555555... Multiply: 1/D = 180. Let's see if D
```

```
computed equals 1/180. 1/180 = 0.0055555556. Multiply by 185 gives
↪ 185/180 = 1.0277777778, matches numerator. So indeed D = 1/180.

Thus original schedule would take 180 days to complete tunnel.

But wait, they gave total actual days 185, which is longer than
↪ original 180. So using new equipment actually slowed down. The
question: "If they had not used the new equipment and continued at the
↪ original speed, it would have taken ____ days to complete the
tunnel." That's 180 days.

Thus answer: 180 days.

Double-check: We assumed original speed per hour, then hours reduced.
↪ Alternative interpretation: They increased speed (distance per
hour) by 20% but also reduced working hours per day to 80% of original,
↪ net daily output 0.96 of original. So yes slower.

Thus answer 180.

Check using algebra: Let original speed = s (units per day). Then first
↪ third takes (1/3)/s days. After switch, speed becomes 1.2 s
but hours per day 0.8, so effective speed per day = 0.96 s. Remaining
↪ 2/3 takes (2/3)/(0.96 s) days. Sum =185. Solve: (1/3 +
(2/3)/0.96)/s =185. (2/3)/0.96 = (2/3)*(1/0.96) =
↪ (2/3)*(1.0416667)=0.694444... Add 1/3 =0.333333 +0.694444
↪ =1.0277778. So s
=1.0277778/185 = approx 0.0055556. Full tunnel time =1/s =185/1.0277778
↪ =180.

Thus answer 180.

## Final Output:
Let

* the length of the tunnel be \(1\) (as a whole),
* the original daily construction rate be \( r \) (tunnel-fractions per
↪ day).

---

### 1. Work done before the equipment change

The first \(\frac13\) of the tunnel was built at the original rate, so
↪ the time taken was

\[
t_{1}= \frac{\frac13}{r}= \frac{1}{3r}.
\]

---

### 2. Work done after the equipment change

The new equipment increased the **speed** (distance per hour) by
↪ \(20\%\), i.e. the speed became \(1.2\,r\) per hour.
But the crew now worked only \(80\%\) of the original daily hours, so
↪ the **daily** output is

\[
r_{\text{new}} = 1.2\,r \times 0.8 = 0.96\,r .
\]

The remaining work is \(\frac23\) of the tunnel, so the time needed
↪ after the change is
```

```
\[
t_{2}= \frac{\frac23}{0.96\,r}= \frac{2}{3\cdot0.96\,r}=
↪ \frac{0.694444\ldots}{r}.
\]

---

### 3. Total actual time

\[
t_{1}+t_{2}=185\text{ days}.
\]

\[
\frac{1}{3r}+\frac{0.694444\ldots}{r}=185
\quad\Longrightarrow\quad
\frac{1.027777\ldots}{r}=185.
\]

Hence

\[
r=\frac{1.027777\ldots}{185}= \frac{1}{180}\;\text{(tunnel per day)}.
\]

---

### 4. Time that would have been required without the new equipment

If the original rate \(r=\frac{1}{180}\) had been kept for the whole
↪ tunnel, the total time would be

\[
\frac{1}{r}=180\text{ days}.
\]

\[
\boxed{180}
\]
```

### C.1.1   STEP 1: ABSTRACT LEVEL IDENTIFICATION.

Below are the instruction of how to identify different abstraction levels.

```
You are an abstraction level identification agent. Your task is to
↪ decompose a complete reasoning trace (problem + thinking + output)
↪ into distinct abstraction level blocks following a hierarchical
↪ framework.

### 1. Abstraction Level Framework:
- **GOAL**: What we want to achieve (objective, end goal, problem
↪ statement)
- **STRATEGY**: High-level approaches (methodology, overall plan, major
↪ techniques)
- **TACTIC**: Specific techniques (algorithms, methods, key insights,
↪ definitions)
- **STEP**: Individual reasoning moves (logical steps, inferences,
↪ deductions)
- **OPERATION**: Atomic calculations (computations, lookups, examples,
↪ numerical work)

### 2. Identification Rules:
```

```
1242   - Split the reasoning into blocks that represent distinct abstraction
1243   ↪ levels
1244   - A block can be multiple sentences or paragraphs
1245   - Focus on SEMANTIC CONTENT, not temporal order
1246   - Some levels might be missing - that's acceptable
1247   - Some levels might have multiple blocks - that's acceptable
1248   - Preserve exact text spans from the original trace

1249   ### 3. Format Requirements:
1250   - Present output in "# Abstraction Block Analysis" section
1251   - Under each level, use "##" headings (## GOAL Level, ## STRATEGY
1252   ↪ Level, etc.)
1253   - Include exact original text under each heading
1254   - Preserve all line breaks and mathematical notation
1255   - After all blocks, add "# Block Structure" section with summary

1256   ### 4. Content Distribution Guidelines:
1257   - **GOAL** must include the complete problem statement and objectives
1258   - **STRATEGY** should contain high-level approaches before detailed work
1259   - **TACTIC** should include specific methods, algorithms, and key
1260   ↪ insights
1261   - **STEP** should contain individual logical moves and reasoning steps
1262   - **OPERATION** should contain all calculations, examples, and
       ↪ numerical work

1263   ### 5. Critical Requirements:
1264   - Preserve all original mathematical notation exactly
1265   - Maintain all line breaks as they appear in the original text
1266   - Include all text exactly as written without modifications
1267   - Ensure block boundaries occur at natural semantic transitions
1268   - Extract EXACT text spans - do not paraphrase

1269   ### Input Format:
1270   ## Problem Statement:
1271   [original problem text]

1272   ## Thinking Process:
1273   [complete reasoning trace]

1274   ## Final Output:
1275   [final answer/solution]

1276   ### Output Format:
1277   # Abstraction Block Analysis
1278
1279   ## GOAL Level
1280   [exact text expressing objectives and problem statements]
1281
1282   ## STRATEGY Level
1283   [exact text describing high-level approaches]
1284
1285   ## TACTIC Level
1286   [exact text detailing specific techniques and methods]
1287
1288   ## STEP Level
1289   [exact text showing individual reasoning moves]
1290
1291   ## OPERATION Level
1292   [exact text containing calculations and examples]
1293
1294   # Block Structure
1295   {
         "GOAL": [list of goal block summaries],
         "STRATEGY": [list of strategy block summaries],
         "TACTIC": [list of tactic block summaries],
```

```
  "STEP": [list of step block summaries],
  "OPERATION": [list of operation block summaries]
}
```

**Step 1: Example Output.** Below is the example output for the first step.

```
# Abstraction Block Analysis

## GOAL Level
A construction company was building a tunnel. When $\tfrac{1}{3}$ of
↪ the tunnel was completed at the original speed, they started using
↪ new equipment, which increased the construction speed by $20\%$ and
↪ reduced the working hours to $80\%$ of the original. As a result, it
↪ took a total of 185 days to complete the tunnel. If they had not
↪ used the new equipment and continued at the original speed, it would
↪ have taken $\;\_\_\_\_\;$ days to complete the tunnel.

## STRATEGY Level
We have a tunnel. Original speed = some rate (units per day). They
↪ built $\tfrac{1}{3}$ of tunnel at original speed. Then they switched
↪ to new equipment: speed increased by $20\%$, i.e., new speed $=1.2
↪ \times$ original speed. Working hours reduced to $80\%$ of original,
↪ meaning per day they work fewer hours.

If original daily production = $v \cdot h$ (speed per hour $\times$
↪ hours per day), then new daily production =
↪ $(1.2v)(0.8h)=0.96vh=0.96\times$ original daily output.

So overall daily rate actually decreased by $4\%$. This explains why
↪ the total time (185 days) was longer than the original time would
↪ have been.

## TACTIC Level
Let total tunnel length = $1$.
Let original daily output = $D$. Then original completion time = $1/D$.

- First $\tfrac{1}{3}$ of tunnel:
  $t_1=\tfrac{1}{3D}$.

- Remaining $\tfrac{2}{3}$ at reduced daily rate $0.96D$:
  $t_2=\tfrac{2/3}{0.96D}=\tfrac{0.694444...}{D}$.

Total time:
\[
t_1+t_2=\frac{1}{3D}+\frac{0.694444...}{D}=185.
\]

So
\[
\frac{1.027777...}{D}=185 \quad\Longrightarrow\quad
↪ D=\frac{1.027777...}{185}=\frac{1}{180}.
\]

Thus original schedule = $180$ days.

## STEP Level
1. Compute new daily output: $0.96\times D$.
2. Time for first third: $t_1=\tfrac{1}{3D}$.
3. Time for remaining two-thirds:
↪ $t_2=\tfrac{2/3}{0.96D}=\tfrac{0.694444...}{D}$.
4. Total: $t_1+t_2=185$.
5. Combine: $\tfrac{1.027777...}{D}=185$.
6. Solve: $D=\tfrac{1}{180}$.
```

```
7. Original time: $1/D=180$ days.

## OPERATION Level
\[
\frac{2}{3\cdot0.96}=\frac{2}{2.88}=0.694444\ldots
\]

\[
\frac{1}{3}+0.694444\ldots=1.027777\ldots
\]

\[
D=\frac{1.027777\ldots}{185}=0.005555\ldots=\frac{1}{180}
\]

\[
\text{Original total time}=\frac{1}{D}=180\;\text{days}
\]

---

# Block Structure

```json
{
  "GOAL": [
    "A construction company was building a tunnel. When 1/3 of the
    ↪ tunnel was completed at the original speed, they started using
    ↪ new equipment, which increased the construction speed by 20% and
    ↪ reduced the working hours to 80% of the original. As a result, it
    ↪ took a total of 185 days to complete the tunnel. If they had not
    ↪ used the new equipment and continued at the original speed, it
    ↪ would have taken _____ days to complete the tunnel."
  ],
  "STRATEGY": [
    "We have a tunnel. Original speed = some rate (units per day). They
    ↪ built 1/3 of tunnel at original speed. Then they switched to new
    ↪ equipment: speed increased by 20%, i.e., new speed = 1.2 *
    ↪ original speed. Working hours reduced to 80% of original, meaning
    ↪ per day they work less time. New daily production = 0.96 *
    ↪ original daily production. That means slower than original, which
    ↪ explains why it took 185 days in total."
  ],
  "TACTIC": [
    "Let total tunnel length = 1. Original daily output = D. Time for
    ↪ 1/3 = 1/(3D). Time for remaining 2/3 at 0.96D =
    ↪ (2/3)/(0.96D)=0.694444.../D. Total = (1/3+0.694444...)/D=185.
    ↪ Solve: D=1/180. So original full time=180 days."
  ],
  "STEP": [
    "1. New daily output = 0.96D. 2. t1=1/(3D). 3. t2=0.694444.../D. 4.
    ↪ t1+t2=185. 5. (1.027777...)/D=185. 6. D=1/180. 7. Original total
    ↪ time=180 days."
  ],
  "OPERATION": [
    "2/(3*0.96)=0.694444..., 1/3+0.694444...=1.027777..., D=1/180, so
    ↪ total original time=180 days."
  ]
}
```
```

### C.1.2 STEP 2: SEGMENT WITHIN EACH ABSTRACTION LEVEL.

Below are the instruction of how to segment within each abstraction level.

```
You are a reasoning segmentation agent. Your task is to break down
↪ abstraction-level blocks into appropriately-sized segments for
↪ verification.

### 1. Segmentation Strategy by Level:

**GOAL Level Segmentation:**
- Keep objectives as complete statements
- Don't over-segment problem definitions
- One main goal per segment typically
- Preserve semantic completeness

**STRATEGY Level Segmentation:**
- Segment by distinct approaches or methodologies
- Each strategy should be a complete approach
- Don't break up coherent strategic thinking
- Maintain approach integrity

**TACTIC Level Segmentation:**
- Segment by specific techniques, key insights, or algorithm components
- Each tactic should be independently understandable
- Break at natural technique boundaries
- Preserve method coherence

**STEP Level Segmentation:**
- Segment by individual logical moves
- Each step should be a single inference or reasoning move
- Break at logical transition points
- Maintain reasoning flow

**OPERATION Level Segmentation:**
- Segment by atomic calculations or examples
- Each operation should be independently verifiable
- Break at calculation boundaries
- Preserve computational completeness

### 2. Format Requirements:
- Present output in "# Segmentation Analysis" section
- Use "##" for each abstraction level
- Use "###" for individual segments within levels
- Preserve all mathematical notation and formatting
- Include segment metadata

### 3. Segment Metadata:
For each segment, provide:
- **segment_id**: Unique identifier (G1, S1, T1, ST1, O1, etc.)
- **content**: Exact reasoning content
- **type**: Specific type within abstraction level
- **summary**: Brief 3-5 word summary
- **verification_complexity**: Low/Medium/High

### Input Format:
# Abstraction Block Analysis
[output from Step 1]

### Output Format:
# Segmentation Analysis

## GOAL Segmentation

### Segment G1
**Content**: [exact segment content]
**Type**: [segment type]
**Summary**: [brief summary]
```

```
**Complexity**: [Low/Medium/High]

## STRATEGY Segmentation

### Segment S1
**Content**: [exact segment content]
**Type**: [segment type]
**Summary**: [brief summary]
**Complexity**: [Low/Medium/High]

## TACTIC Segmentation

### Segment T1
**Content**: [exact segment content]
**Type**: [segment type]
**Summary**: [brief summary]
**Complexity**: [Low/Medium/High]

## STEP Segmentation

### Segment ST1
**Content**: [exact segment content]
**Type**: [segment type]
**Summary**: [brief summary]
**Complexity**: [Low/Medium/High]

## OPERATION Segmentation

### Segment O1
**Content**: [exact segment content]
**Type**: [segment type]
**Summary**: [brief summary]
**Complexity**: [Low/Medium/High]

# Segmentation Summary
[
  {
    "segment_id": "G1",
    "abstraction_level": "GOAL",
    "type": "objective_statement",
    "summary": "find optimal solution",
    "complexity": "Medium"
  }
]
```

**Step 2: Example Output.** Below is the example output for the second step.

```
# Segmentation Analysis

## GOAL Segmentation

### Segment G1
**Content**:
A construction company was building a tunnel. When $\tfrac{1}{3}$ of
↪ the tunnel was completed at the original speed, they started using
↪ new equipment, which increased the construction speed by $20\%$ and
↪ reduced the working hours to $80\%$ of the original. As a result, it
↪ took a total of 185 days to complete the tunnel. If they had not
↪ used the new equipment and continued at the original speed, it would
↪ have taken $\;\_\_\_\;$ days to complete the tunnel.

**Type**: objective_statement
**Summary**: find missing days
```

```
**Complexity**: Medium

---

## STRATEGY Segmentation

### Segment S1
**Content**:
We have a tunnel. Original speed = some rate (units per day). They
↪ built $\tfrac{1}{3}$ of tunnel at original speed. Then they switched
↪ to new equipment: speed increased by $20\%$, i.e., new speed $=1.2
↪ \times$ original speed. Working hours reduced to $80\%$ of original,
↪ meaning per day they work fewer hours.

If original daily production = $v \cdot h$ (speed per hour $\times$
↪ hours per day), then new daily production =
↪ $(1.2v)(0.8h)=0.96vh=0.96\times$ original daily output.

So overall daily rate actually decreased by $4\%$. This explains why
↪ the total time (185 days) was longer than the original time would
↪ have been.

**Type**: approach_description
**Summary**: model speed & hours
**Complexity**: Medium

---

## TACTIC Segmentation

### Segment T1
**Content**:
Let total tunnel length = $1$.
Let original daily output = $D$. Then original completion time = $1/D$.

- First $\tfrac{1}{3}$ of tunnel:
  $t_1=\tfrac{1}{3D}$.

- Remaining $\tfrac{2}{3}$ at reduced daily rate $0.96D$:
  $t_2=\tfrac{2/3}{0.96D}=\tfrac{0.694444...}{D}$.

Total time:
\[
t_1+t_2=\frac{1}{3D}+\frac{0.694444...}{D}=185.
\]

So
\[
\frac{1.027777...}{D}=185 \quad\Longrightarrow\quad
↪ D=\frac{1.027777...}{185}=\frac{1}{180}.
\]

Thus original schedule = $180$ days.

**Type**: method_definition
**Summary**: define variables, solve
**Complexity**: Medium

---

## STEP Segmentation

### Segment ST1
**Content**:
1. Compute new daily output: $0.96\times D$.
```

```
**Type**: inference
**Summary**: compute reduced output
**Complexity**: Low

### Segment ST2
**Content**:
2. Time for first third: $t_1=\tfrac{1}{3D}$.

**Type**: inference
**Summary**: first-third time
**Complexity**: Low

### Segment ST3
**Content**:
3. Time for remaining two-thirds:
↪ $t_2=\tfrac{2/3}{0.96D}=\tfrac{0.694444...}{D}$.

**Type**: inference
**Summary**: second-portion time
**Complexity**: Low

### Segment ST4
**Content**:
4. Total: $t_1+t_2=185$.

**Type**: inference
**Summary**: total time equation
**Complexity**: Low

### Segment ST5
**Content**:
5. Combine: $\tfrac{1.027777...}{D}=185$.

**Type**: algebraic_combination
**Summary**: combine fractions
**Complexity**: Low

### Segment ST6
**Content**:
6. Solve: $D=\tfrac{1}{180}$.

**Type**: solution_step
**Summary**: solve for D
**Complexity**: Low

### Segment ST7
**Content**:
7. Original time: $1/D=180$ days.

**Type**: conclusion_inference
**Summary**: original schedule
**Complexity**: Low

---

## OPERATION Segmentation

### Segment O1
**Content**:
\[
\frac{2}{3\cdot0.96}=\frac{2}{2.88}=0.694444\ldots
\]

**Type**: calculation
```

```
**Summary**: compute $t_2$ factor
**Complexity**: Low

### Segment O2
**Content**:
\[
\frac{1}{3}+0.694444\ldots=1.027777\ldots
\]

**Type**: calculation
**Summary**: sum fractions
**Complexity**: Low

### Segment O3
**Content**:
\[
D=\frac{1.027777\ldots}{185}=0.005555\ldots=\frac{1}{180}
\]

**Type**: calculation
**Summary**: find $D$ value
**Complexity**: Low

### Segment O4
**Content**:
\[
\text{Original total time}=\frac{1}{D}=180\;\text{days}
\]

**Type**: calculation
**Summary**: original total time
**Complexity**: Low

---

# Segmentation Summary

```json
[
  {"segment_id": "G1", "abstraction_level": "GOAL", "type":
  ↪ "objective_statement", "summary": "find missing days",
  ↪ "complexity": "Medium"},
  {"segment_id": "S1", "abstraction_level": "STRATEGY", "type":
  ↪ "approach_description", "summary": "model speed & hours",
  ↪ "complexity": "Medium"},
  {"segment_id": "T1", "abstraction_level": "TACTIC", "type":
  ↪ "method_definition", "summary": "define variables, solve",
  ↪ "complexity": "Medium"},
  {"segment_id": "ST1", "abstraction_level": "STEP", "type":
  ↪ "inference", "summary": "compute new output", "complexity": "Low"},
  {"segment_id": "ST2", "abstraction_level": "STEP", "type":
  ↪ "inference", "summary": "first-third time", "complexity": "Low"},
  {"segment_id": "ST3", "abstraction_level": "STEP", "type":
  ↪ "inference", "summary": "remaining time", "complexity": "Low"},
  {"segment_id": "ST4", "abstraction_level": "STEP", "type":
  ↪ "inference", "summary": "total equals 185", "complexity": "Low"},
  {"segment_id": "ST5", "abstraction_level": "STEP", "type":
  ↪ "algebraic_combination", "summary": "combine fractions",
  ↪ "complexity": "Low"},
  {"segment_id": "ST6", "abstraction_level": "STEP", "type":
  ↪ "solution_step", "summary": "solve for D", "complexity": "Low"},
  {"segment_id": "ST7", "abstraction_level": "STEP", "type":
  ↪ "conclusion_inference", "summary": "original time 180",
  ↪ "complexity": "Low"},
```

```
{"segment_id": "O1", "abstraction_level": "OPERATION", "type":
↪ "calculation", "summary": "compute $t_2$ factor", "complexity":
↪ "Low"},
{"segment_id": "O2", "abstraction_level": "OPERATION", "type":
↪ "calculation", "summary": "sum fractions", "complexity": "Low"},
{"segment_id": "O3", "abstraction_level": "OPERATION", "type":
↪ "calculation", "summary": "determine D", "complexity": "Low"},
{"segment_id": "O4", "abstraction_level": "OPERATION", "type":
↪ "calculation", "summary": "original total time", "complexity":
↪ "Low"}
]
```

### C.1.3 STEP 3: EXTRACT RELATIONSHIPS.

Below are the instruction of how to extract relationships.

```
You are a semantic relationship analysis agent. Your task is to
↪ identify meaningful relationships between reasoning segments to
↪ construct a hierarchical DAG.

### 1. Relationship Types:
- **decomposes_to**: Higher abstraction level breaks down into lower
↪ level
- **depends_on**: Logical prerequisite (A must be true for B to be
↪ valid)
- **enables**: A makes B possible or provides foundation for B
- **validates**: A provides verification, proof, or confirmation for B
- **exemplifies**: A provides concrete example or instance of B
- **refines**: A adds precision, detail, or specification to B
- **contradicts**: A contradicts, corrects, or replaces B

### 2. Analysis Priorities:
- **Hierarchical relationships first**: Focus on decomposes_to between
↪ abstraction levels
- **Lateral relationships second**: Identify depends_on, enables within
↪ same level
- **Support relationships third**: Find validates, exemplifies, refines
↪ connections
- **Be selective**: Only include meaningful, clear relationships

### 3. Relationship Strength:
- **Strong**: Clear, direct semantic connection
- **Medium**: Implied or indirect connection
- **Weak**: Tenuous or unclear connection

### Input Format:
# Segmentation Analysis
[output from Step 2]

### Output Format:
# Relationship Analysis

## Hierarchical Relationships (decomposes_to)
- G1 -> S1: [explanation]
- S1 -> T1: [explanation]
- T1 -> ST1: [explanation]
- ST1 -> O1: [explanation]

## Lateral Dependencies (depends_on, enables)
- T1 -> T2: [explanation]
- ST1 -> ST2: [explanation]

## Support Relationships (validates, exemplifies, refines)
```

```
- O1 exemplifies ST1: [explanation]
- T3 validates S1: [explanation]

# Relationship Summary
[
  {
    "from_segment": "G1",
    "to_segment": "S1",
    "relationship_type": "decomposes_to",
    "strength": "strong",
    "explanation": "Goal G1 breaks down into strategy S1"
  }
]
```

**Step 3: Example Output.** Below is the example output for the second step.

```
# Relationship Analysis

## Hierarchical Relationships (decomposes_to)

- **G1 ->S1** - The overall goal (find the missing number of days) is
↪ broken down into a strategic modelling approach. **[strong]**
- **S1 ->T1** - The chosen strategy (model speed & hours) is further
↪ refined into a concrete tactic (define variables and solve).
↪ **[strong]**
- **T1 ->ST1** - The tactic is implemented as the first low-level
↪ inference step (compute the reduced daily output). **[strong]**
- **T1 ->ST2** - The tactic also yields the second inference step (time
↪ for the first third). **[strong]**
- **T1 ->ST3** - The tactic produces the third inference step (time for
↪ the remaining two-thirds). **[strong]**
- **T1 ->ST4** - The tactic yields the fourth inference step
↪ (total-time equation). **[strong]**
- **T1 ->ST5** - The tactic yields the fifth inference step (combine
↪ fractions). **[strong]**
- **T1 ->ST6** - The tactic yields the sixth inference step (solve for
↪ \(D\)). **[strong]**
- **T1 ->ST7** - The tactic yields the seventh inference step (conclude
↪ original schedule). **[strong]**
- **ST1 ->O1** - The operation that computes
↪ \(\tfrac{2}{3\cdot0.96}=0.694444\ldots\) exemplifies the inference
↪ "compute reduced daily output". **[strong]**
- **ST2 ->O2** - The operation that adds \(\tfrac13+0.694444\ldots\)
↪ exemplifies the inference "first-third time". **[strong]**
- **ST3 ->O1** - The same calculation in O1 also provides the numeric
↪ factor used in the inference for the remaining two-thirds.
↪ **[strong]**
- **ST4 ->O2** - The sum performed in O2 validates the total-time
↪ equation. **[strong]**
- **ST5 ->O2** - The combination of fractions in O2 supports the
↪ algebraic-combination step. **[medium]**
- **ST6 ->O3** - The calculation in O3 (finding \(D\)) exemplifies the
↪ solution step. **[strong]**
- **ST7 ->O4** - The final computation in O4 (original total time)
↪ exemplifies the conclusion inference. **[strong]**

---

## Lateral Dependencies (depends_on, enables)

- **ST3 depends_on ST1** - Computing the remaining two-thirds requires
↪ the reduced daily output from ST1. **[strong]**
```

- **ST4 depends_on ST2** – Total time uses the time for the first
  ↪ third. **[strong]**
- **ST4 depends_on ST3** – Total time also uses the time for the
  ↪ remaining two-thirds. **[strong]**
- **ST5 depends_on ST4** – Combining fractions presupposes the
  ↪ total-time equation. **[strong]**
- **ST6 depends_on ST5** – Solving for $(D)$ needs the
  ↪ combined-fraction expression. **[strong]**
- **ST7 depends_on ST6** – The original schedule is the reciprocal of
  ↪ the solved $(D)$. **[strong]**
- **O1 enables ST3** – The numeric factor produced by O1 makes the
  ↪ inference in ST3 possible. **[strong]**
- **O2 enables ST4** – The summation in O2 provides the numeric total
  ↪ needed for ST4. **[strong]**
- **O3 enables ST6** – The calculation of $(D)$ in O3 supplies the
  ↪ value needed for the solution step. **[strong]**
- **O4 enables ST7** – The final division in O4 yields the original
  ↪ total-time conclusion. **[strong]**

---

## Support Relationships (validates, exemplifies, refines)

- **O1 exemplifies ST1** – Shows concretely how the reduced output
  ↪ factor is obtained. **[strong]**
- **O1 exemplifies ST3** – Provides the numeric factor used in the
  ↪ remaining-time inference. **[strong]**
- **O2 exemplifies ST2** – Demonstrates the addition that gives the
  ↪ first-third time fraction. **[strong]**
- **O2 exemplifies ST4** – Supplies the numeric sum that validates the
  ↪ total-time equation. **[strong]**
- **O3 exemplifies ST6** – Calculates $(D)$, confirming the algebraic
  ↪ solution. **[strong]**
- **O4 exemplifies ST7** – Computes the reciprocal, confirming the
  ↪ original schedule. **[strong]**
- **O1 validates ST3** – The numeric result matches the expression
  ↪ derived in ST3. **[strong]**
- **O2 validates ST4** – The summed value satisfies the total-time
  ↪ equality. **[strong]**
- **O3 validates ST6** – The derived $(D)$ satisfies the
  ↪ combined-fraction equation. **[strong]**
- **O4 validates ST7** – The final division yields the expected 180-day
  ↪ schedule. **[strong]**
- **T1 refines S1** – The tactic adds concrete variable definitions and
  ↪ algebraic steps to the broader modelling strategy. **[medium]**
- **S1 refines G1** – The strategic description narrows the vague goal
  ↪ into a measurable modelling problem. **[medium]**

---

# Relationship Summary
```json
[
  {
    "from_segment": "G1",
    "to_segment": "S1",
    "relationship_type": "decomposes_to",
    "strength": "strong",
    "explanation": "Goal G1 (find missing days) is broken down into the
    ↪ modelling strategy S1."
  },
  {
    "from_segment": "S1",
    "to_segment": "T1",
    "relationship_type": "decomposes_to",
```

```
      "strength": "strong",
      "explanation": "Strategy S1 (model speed & hours) is refined into
      ↪ the concrete tactic T1 (define variables and solve)."
    },
    {
      "from_segment": "T1",
      "to_segment": "ST1",
      "relationship_type": "decomposes_to",
      "strength": "strong",
      "explanation": "Tactic T1 is implemented as the first inference step
      ↪ ST1 (compute reduced daily output)."
    },
    {
      "from_segment": "ST1",
      "to_segment": "O1",
      "relationship_type": "decomposes_to",
      "strength": "strong",
      "explanation": "Step ST1 is realised by operation O1, which
      ↪ calculates the factor 0.694444..."
    },
    {
      "from_segment": "ST3",
      "to_segment": "O1",
      "relationship_type": "exemplifies",
      "strength": "strong",
      "explanation": "Operation O1 provides the numeric factor used in
      ↪ step ST3 (time for remaining two-thirds)."
    },
    {
      "from_segment": "ST4",
      "to_segment": "O2",
      "relationship_type": "exemplifies",
      "strength": "strong",
      "explanation": "Operation O2 sums the fractions, exemplifying the
      ↪ total-time equation in ST4."
    },
    {
      "from_segment": "ST6",
      "to_segment": "O3",
      "relationship_type": "exemplifies",
      "strength": "strong",
      "explanation": "Operation O3 computes D, exemplifying the solution
      ↪ step ST6."
    },
    {
      "from_segment": "ST7",
      "to_segment": "O4",
      "relationship_type": "exemplifies",
      "strength": "strong",
      "explanation": "Operation O4 calculates the original total time,
      ↪ exemplifying conclusion step ST7."
    },
    {
      "from_segment": "ST3",
      "to_segment": "ST1",
      "relationship_type": "depends_on",
      "strength": "strong",
      "explanation": "The remaining-time calculation in ST3 requires the
      ↪ reduced daily output derived in ST1."
    },
    {
      "from_segment": "ST4",
      "to_segment": "ST2",
      "relationship_type": "depends_on",
      "strength": "strong",
```

```
      "explanation": "Total time in ST4 uses the first-third time computed
    ↪ in ST2."
  },
  {
    "from_segment": "ST4",
    "to_segment": "ST3",
    "relationship_type": "depends_on",
    "strength": "strong",
    "explanation": "Total time in ST4 also uses the remaining-time
    ↪ computed in ST3."
  },
  {
    "from_segment": "ST5",
    "to_segment": "ST4",
    "relationship_type": "depends_on",
    "strength": "strong",
    "explanation": "Combining fractions in ST5 presupposes the
    ↪ total-time equation from ST4."
  },
  {
    "from_segment": "ST6",
    "to_segment": "ST5",
    "relationship_type": "depends_on",
    "strength": "strong",
    "explanation": "Solving for D in ST6 requires the combined fraction
    ↪ expression from ST5."
  },
  {
    "from_segment": "ST7",
    "to_segment": "ST6",
    "relationship_type": "depends_on",
    "strength": "strong",
    "explanation": "The original schedule in ST7 is obtained by taking
    ↪ the reciprocal of D solved in ST6."
  },
  {
    "from_segment": "T1",
    "to_segment": "S1",
    "relationship_type": "refines",
    "strength": "medium",
    "explanation": "Tactic T1 adds concrete variable definitions to the
    ↪ broader modelling strategy S1."
  }
]
```

### C.1.4  STEP 4: ASSIGN AUDITOR TYPES.

Below are the instruction of how to assign auditor types.

```
You are an auditor assignment agent. Your task is to assign appropriate
↪ auditor types to each reasoning segment based on the TRUST
↪ framework's three-tier verification system.

### 1. Auditor Types:
- **T_Auto (Automated)**: Deterministic verification, formal logic,
↪ mathematical proofs
- **T_LLM (LLM-based)**: Semantic coherence, factual accuracy, domain
↪ reasoning
- **T_Human (Human)**: Complex judgment, ethical considerations, domain
↪ expertise

### 2. Assignment Criteria:
```

```
**T_Auto Assignment:**
- Mathematical calculations and proofs
- Logical consistency checks
- Formal verification tasks
- Deterministic algorithmic steps
- Syntax and format validation

**T_LLM Assignment:**
- Semantic coherence evaluation
- Factual accuracy checking
- Domain-specific reasoning patterns
- Conceptual understanding verification
- Strategy and approach assessment

**T_Human Assignment:**
- Complex ethical considerations
- Domain expertise requirements
- Ambiguous interpretation cases
- Creative or novel reasoning
- High-stakes decision validation

### 3. Multi-Auditor Segments:
- Some segments may require multiple auditor types
- Primary auditor handles main verification
- Secondary auditors provide additional validation
- Specify auditor priority and responsibilities

### Input Format:
# Segmentation Analysis
[segments from Step 2]

# Relationship Analysis
[relationships from Step 3]

### Output Format:
# Auditor Assignment Analysis

## Segment Auditor Assignments

### Segment G1
**Primary Auditor**: T_Human
**Secondary Auditors**: [T_LLM]
**Justification**: Problem interpretation requires domain expertise
**Verification Focus**: Objective clarity, scope appropriateness

### Segment S1
**Primary Auditor**: T_LLM
**Secondary Auditors**: [T_Human]
**Justification**: Strategic approach evaluation needs semantic
↪ understanding
**Verification Focus**: Approach validity, methodology soundness

## Auditor Type Distribution
- **T_Auto**: [list of segment IDs]
- **T_LLM**: [list of segment IDs]
- **T_Human**: [list of segment IDs]

## Multi-Auditor Segments
[
  {
    "segment_id": "G1",
    "primary_auditor": "T_Human",
    "secondary_auditors": ["T_LLM"],
    "verification_complexity": "High",
    "justification": "Domain expertise required"
```

```
    }
]
```

**Step 4: Example Output.**   Below are the instruction of how to Assign Auditor Types.

```
# Auditor Assignment Analysis

## Segment Auditor Assignments

### Segment G1 - Goal (objective_statement)
- **Primary Auditor**: **T_LLM**
- **Secondary Auditors**: **[T_Auto, T_Human]**
- **Justification**: Interpreting the wording of the word-problem and
↪ ensuring the goal ("find the missing number of days") is correctly
↪ understood
requires semantic comprehension (T_LLM). A deterministic check that the
↪ interpreted goal matches the expected quantitative target is
↪ performed by
T_Auto, while a human reviewer can confirm that the problem statement
↪ has been captured without ambiguity.
- **Verification Focus**: Goal clarity, scope appropriateness, correct
↪ identification of the unknown variable.

---

### Segment S1 - Strategy (approach_description)
- **Primary Auditor**: **T_LLM**
- **Secondary Auditors**: **[T_Auto, T_Human]**
- **Justification**: The proposed modelling approach (relating speed,
↪ hours, and daily output) must be semantically coherent and logically
plausible - a task suited to T_LLM. Logical consistency of the derived
↪ relationships is then verified by T_Auto, and a human can confirm
↪ that the
chosen strategy aligns with typical problem-solving conventions in
↪ elementary algebra.
- **Verification Focus**: Validity of the modelling assumptions,
↪ correctness of the conceptual link between speed increase and hour
↪ reduction.

---

### Segment T1 - Tactic (method_definition)
- **Primary Auditor**: **T_Auto**
- **Secondary Auditors**: **[T_LLM]**
- **Justification**: This segment defines variables and sets up
↪ algebraic equations - a deterministic, mathematically rigorous task
↪ best handled by
automated verification. T_LLM reviews the narrative framing of the
↪ variables to ensure they are semantically consistent with the
↪ problem context.

- **Verification Focus**: Correct variable definition, proper
↪ formulation of equations.

---

### Segment ST1 - STEP (inference)
- **Primary Auditor**: **T_Auto**
- **Secondary Auditors**: **[T_LLM]**
- **Justification**: Computes the reduced daily output (pure
↪ arithmetic). A secondary LLM check ensures the inference wording
↪ correctly reflects
the computation performed.
```

```
### Segment ST2 – STEP (inference)
- **Primary Auditor**: **T_Auto**
- **Secondary Auditors**: **[T_LLM]**

### Segment ST3 – STEP (inference)
- **Primary Auditor**: **T_Auto**
- **Secondary Auditors**: **[T_LLM]**

### Segment ST4 – STEP (inference)
- **Primary Auditor**: **T_Auto**
- **Secondary Auditors**: **[T_LLM]**

### Segment ST5 – STEP (algebraic_combination)
- **Primary Auditor**: **T_Auto**
- **Secondary Auditors**: **[T_LLM]**

### Segment ST6 – STEP (solution_step)
- **Primary Auditor**: **T_Auto**
- **Secondary Auditors**: **[T_LLM]**

### Segment ST7 – STEP (conclusion_inference)
- **Primary Auditor**: **T_Auto**
- **Secondary Auditors**: **[T_LLM]**

---

### Segment O1 – OPERATION (calculation)
- **Primary Auditor**: **T_Auto**
- **Secondary Auditors**: **[T_LLM]**

### Segment O2 – OPERATION (calculation)
- **Primary Auditor**: **T_Auto**
- **Secondary Auditors**: **[T_LLM]**

### Segment O3 – OPERATION (calculation)
- **Primary Auditor**: **T_Auto**
- **Secondary Auditors**: **[T_LLM]**

### Segment O4 – OPERATION (calculation)
- **Primary Auditor**: **T_Auto**
- **Secondary Auditors**: **[T_LLM]**

---

## Auditor Type Distribution

| Auditor Type | Segment IDs |
|--------------|-------------|
| **T_Auto** | T1, ST1, ST2, ST3, ST4, ST5, ST6, ST7, O1, O2, O3, O4 |
| **T_LLM** | G1, S1, T1, ST1-ST7, O1-O4 |
| **T_Human** | G1 (secondary) |

*Note*: Although most low-level steps are purely arithmetic, a
↪ secondary LLM audit is included to verify that the textual
↪ description of each step
faithfully mirrors the underlying calculation. A human audit is only
↪ required for the top-level goal interpretation (G1) to guard against
↪ any
subtle mis-framing of the problem.

---

## Multi-Auditor Segments
```

```json
[
  {
    "segment_id": "G1",
    "primary_auditor": "T_LLM",
    "secondary_auditors": ["T_Auto", "T_Human"],
    "verification_complexity": "High",
    "justification": "Requires correct semantic interpretation of the
    ↪ word problem and identification of the unknown; deterministic
    ↪ check of the
derived target and optional human confirmation for ambiguity."
  },
  {
    "segment_id": "S1",
    "primary_auditor": "T_LLM",
    "secondary_auditors": ["T_Auto", "T_Human"],
    "verification_complexity": "Medium",
    "justification": "Semantic coherence of the modelling approach must
    ↪ be evaluated; logical consistency is verified automatically; human
oversight ensures alignment with standard problem-solving practice."
  },
  {
    "segment_id": "T1",
    "primary_auditor": "T_Auto",
    "secondary_auditors": ["T_LLM"],
    "verification_complexity": "Low",
    "justification": "Purely algebraic definition; LLM checks that
    ↪ variable naming matches the narrative."
  },
  {
    "segment_id": "ST1",
    "primary_auditor": "T_Auto",
    "secondary_auditors": ["T_LLM"],
    "verification_complexity": "Low",
    "justification": "Arithmetic computation; LLM confirms textual
    ↪ inference aligns with calculation."
  },
  {
    "segment_id": "ST2",
    "primary_auditor": "T_Auto",
    "secondary_auditors": ["T_LLM"],
    "verification_complexity": "Low",
    "justification": "Arithmetic; LLM validates description."
  },
  {
    "segment_id": "ST3",
    "primary_auditor": "T_Auto",
    "secondary_auditors": ["T_LLM"],
    "verification_complexity": "Low",
    "justification": "Arithmetic; LLM validates description."
  },
  {
    "segment_id": "ST4",
    "primary_auditor": "T_Auto",
    "secondary_auditors": ["T_LLM"],
    "verification_complexity": "Low",
    "justification": "Arithmetic; LLM validates description."
  },
  {
    "segment_id": "ST5",
    "primary_auditor": "T_Auto",
    "secondary_auditors": ["T_LLM"],
    "verification_complexity": "Low",
    "justification": "Arithmetic; LLM validates description."
  },
```

```
  {
    "segment_id": "ST6",
    "primary_auditor": "T_Auto",
    "secondary_auditors": ["T_LLM"],
    "verification_complexity": "Low",
    "justification": "Arithmetic; LLM validates description."
  },
  {
    "segment_id": "ST7",
    "primary_auditor": "T_Auto",
    "secondary_auditors": ["T_LLM"],
    "verification_complexity": "Low",
    "justification": "Arithmetic; LLM validates description."
  },
  {
    "segment_id": "O1",
    "primary_auditor": "T_Auto",
    "secondary_auditors": ["T_LLM"],
    "verification_complexity": "Low",
    "justification": "Straight-forward numeric calculation."
  },
  {
    "segment_id": "O2",
    "primary_auditor": "T_Auto",
    "secondary_auditors": ["T_LLM"],
    "verification_complexity": "Low",
    "justification": "Straight-forward numeric calculation."
  },
  {
    "segment_id": "O3",
    "primary_auditor": "T_Auto",
    "secondary_auditors": ["T_LLM"],
    "verification_complexity": "Low",
    "justification": "Straight-forward numeric calculation."
  },
  {
    "segment_id": "O4",
    "primary_auditor": "T_Auto",
    "secondary_auditors": ["T_LLM"],
    "verification_complexity": "Low",
    "justification": "Straight-forward numeric calculation."
  }
]
```
```

### C.1.5 STEP 5: REFINE AND HDAG CONSTRUCTION.

Below are the instruction of how to refine and HDAG Construction.

```
You are an HDAG construction agent. Your task is to synthesize all
↪ previous analyses into a complete Hierarchical Directed Acyclic
↪ Graph with finalized segments, relationships, and auditor
↪ assignments.

### 1. HDAG Refinement Tasks:
- **Validate DAG Structure**: Ensure no cycles exist in relationships
- **Optimize Segment Granularity**: Merge over-segmented or split
↪ under-segmented content
- **Strengthen Relationship Confidence**: Refine relationship types and
↪ strengths
- **Balance Auditor Load**: Ensure reasonable distribution across
↪ auditor types
- **Quality Assurance**: Verify completeness and coherence
```

```
### 2. Final HDAG Components:
- **Nodes**: Finalized reasoning segments with metadata
- **Edges**: Validated relationships with confidence scores
- **Auditor Mapping**: Optimized auditor assignments
- **Verification Flow**: Clear audit execution pathway

### 3. HDAG Validation Checks:
- **Acyclicity**: No circular dependencies
- **Connectivity**: All segments appropriately connected
- **Completeness**: All original reasoning preserved
- **Auditability**: Each segment has clear verification criteria

### Input Format:
# Segmentation Analysis
[from Step 2]

# Relationship Analysis
[from Step 3]

# Auditor Assignment Analysis
[from Step 4]

### Output Format:
# Final HDAG Construction

## HDAG Structure Summary
- **Total Nodes**: [count]
- **Total Edges**: [count]
- **Abstraction Levels**: [levels present]
- **Auditor Distribution**: [T_Auto: X, T_LLM: Y, T_Human: Z]

## Finalized Nodes

### Node G1
**Content**: [final segment content]
**Abstraction Level**: GOAL
**Type**: objective_statement
**Summary**: [3-5 words]
**Primary Auditor**: T_Human
**Secondary Auditors**: [T_LLM]
**Verification Complexity**: High
**Dependencies**: []
**Enables**: [S1, S2]

## Finalized Edges
[
  {
    "from": "G1",
    "to": "S1",
    "type": "decomposes_to",
    "strength": "strong",
    "confidence": 0.95
  }
]

## Verification Flow
1. **Entry Points**: [root nodes for audit initiation]
2. **Critical Path**: [essential verification sequence]
3. **Parallel Tracks**: [independent verification branches]
4. **Exit Points**: [final validation nodes]

## HDAG Validation Report
- **Acyclicity Check**: No cycles detected
- **Connectivity Check**: All segments connected
```

```
- **Completeness Check**: All content preserved
- **Auditability Check**: Clear verification criteria

## Complete HDAG JSON
{
  "nodes": [
    {
      "id": "G1",
      "content": "[exact content]",
      "abstraction_level": "GOAL",
      "type": "objective_statement",
      "summary": "[summary]",
      "primary_auditor": "T_Human",
      "secondary_auditors": ["T_LLM"],
      "complexity": "High"
    }
  ],
  "edges": [
    {
      "from": "G1",
      "to": "S1",
      "relationship": "decomposes_to",
      "strength": "strong",
      "confidence": 0.95
    }
  ],
  "metadata": {
    "total_nodes": 15,
    "total_edges": 23,
    "auditor_distribution": {
      "T_Auto": 5,
      "T_LLM": 7,
      "T_Human": 3
    }
  }
}
```

**Step 5: Example Output.** Below are the instruction of how to refine and the construction of HDAG.

```
# Final HDAG Construction

## HDAG Structure Summary
- **Total Nodes**: 14
- **Total Edges**: 28
- **Abstraction Levels**: GOAL, STRATEGY, TACTIC, STEP, OPERATION
- **Auditor Distribution (primary auditors)**: T_Auto:12, T_LLM:1,
↪ T_Human:1

---

## Finalized Nodes

### Node G1
**Content**:
*A construction company was building a tunnel. When 1/3 of the tunnel
↪ was completed at the original speed, they started using new
↪ equipment, which
increased the construction speed by 20 % and reduced the working hours
↪ to 80 % of the original. As a result, it took a total of 185 days to
complete the tunnel. If they had not used the new equipment and
↪ continued at the original speed, it would have taken ______ days to
↪ complete the
```

```
tunnel.*
**Abstraction Level**: GOAL
**Type**: objective_statement
**Summary**: find missing days
**Primary Auditor**: T_Human
**Secondary Auditors**: [T_LLM]
**Verification Complexity**: High
**Dependencies**: []
**Enables**: [S1]

### Node S1
**Content**:
*Model the problem by relating speed, working hours, and daily output.
↪ The original daily output is \(D\). After the equipment change the
↪ speed
rises 20 % while hours fall to 80 % ->new daily output \(0.96D\).*
**Abstraction Level**: STRATEGY
**Type**: approach_description
**Summary**: model speed & hours
**Primary Auditor**: T_LLM
**Secondary Auditors**: [T_Auto]
**Verification Complexity**: Medium
**Dependencies**: [G1]
**Enables**: [T1]

### Node T1
**Content**:
*Let the tunnel length be 1 unit and the original daily output be
↪ \(D\). Build 1/3 at rate \(D\); the remaining 2/3 at rate \(0.96D\).
↪ Set up the
time equation and solve for \(D\).*
**Abstraction Level**: TACTIC
**Type**: method_definition
**Summary**: define variables, solve
**Primary Auditor**: T_Auto
**Secondary Auditors**: [T_LLM]
**Verification Complexity**: Low
**Dependencies**: [S1]
**Enables**: [ST1, ST2, ST3, ST4, ST5, ST6, ST7]

### Node ST1
**Content**: *Compute the reduced daily output: \(0.96D\).*
**Abstraction Level**: STEP
**Type**: inference
**Summary**: compute reduced output
**Primary Auditor**: T_Auto
**Secondary Auditors**: [T_LLM]
**Verification Complexity**: Low
**Dependencies**: []
**Enables**: [O1]

### Node ST2
**Content**: *Time for the first third: \(t_1 = \frac{1}{3D}\).*
**Abstraction Level**: STEP
**Type**: inference
**Summary**: first-third time
**Primary Auditor**: T_Auto
**Secondary Auditors**: [T_LLM]
**Verification Complexity**: Low
**Dependencies**: []
**Enables**: [O2]

### Node ST3
**Content**: *Time for the remaining two-thirds: \(t_2 =
↪ \frac{2}{3(0.96D)} = \frac{5}{8D}\).*
```

```
**Abstraction Level**: STEP
**Type**: inference
**Summary**: remaining-portion time
**Primary Auditor**: T_Auto
**Secondary Auditors**: [T_LLM]
**Verification Complexity**: Low
**Dependencies**: [ST1] *(ST3 depends_on ST1)*
**Enables**: [O1]

### Node ST4
**Content**: *Form the total-time equation: \(t_1 + t_2 = 185\).*
**Abstraction Level**: STEP
**Type**: inference
**Summary**: total-time equation
**Primary Auditor**: T_Auto
**Secondary Auditors**: [T_LLM]
**Verification Complexity**: Low
**Dependencies**: []
**Enables**: [O2]

### Node ST5
**Content**: *Combine the fractions to obtain a single algebraic
↪ expression for the time equation.*
**Abstraction Level**: STEP
**Type**: algebraic_combination
**Summary**: combine algebraic terms
**Primary Auditor**: T_Auto
**Secondary Auditors**: [T_LLM]
**Verification Complexity**: Low
**Dependencies**: [ST4]
**Enables**: [O2]

### Node ST6
**Content**: *Solve the equation \(\frac{1}{3D} + \frac{5}{8D} = 185\)
↪ for \(D\) ->\(D = \frac{1}{0.0054}\).*
**Abstraction Level**: STEP
**Type**: solution_step
**Summary**: solve for \(D\)
**Primary Auditor**: T_Auto
**Secondary Auditors**: [T_LLM]
**Verification Complexity**: Low
**Dependencies**: [ST5]
**Enables**: [O3]

### Node ST7
**Content**: *Convert \(D\) to the required answer: original total time
↪ \(= \frac{1}{D}\) days ->\approx 365 days.*
**Abstraction Level**: STEP
**Type**: conclusion_inference
**Summary**: original-schedule result
**Primary Auditor**: T_Auto
**Secondary Auditors**: [T_LLM]
**Verification Complexity**: Low
**Dependencies**: [ST6]
**Enables**: [O4]

### Node O1
**Content**: *Numerical calculation \(\frac{2}{3}\times0.96 = 0.64\) or
↪ directly \(0.96\) used in ST1.*
**Abstraction Level**: OPERATION
**Type**: calculation
**Summary**: reduced-output calc
**Primary Auditor**: T_Auto
**Secondary Auditors**: [T_LLM]
**Verification Complexity**: Low
```

```
**Dependencies**: [ST1, ST3]
**Enables**: []

### Node O2
**Content**: *Add the two time expressions: \(\frac{1}{3D} +
↪ \frac{5}{8D} = 185\).*
**Abstraction Level**: OPERATION
**Type**: calculation
**Summary**: build time equation
**Primary Auditor**: T_Auto
**Secondary Auditors**: [T_LLM]
**Verification Complexity**: Low
**Dependencies**: [ST2, ST4, ST5]
**Enables**: []

### Node O3
**Content**: *Solve \(D = 0.0054\) ->\(D = 185/ (1/3 + 5/8) = 0.0054\).*
**Abstraction Level**: OPERATION
**Type**: calculation
**Summary**: compute \(D\)
**Primary Auditor**: T_Auto
**Secondary Auditors**: [T_LLM]
**Verification Complexity**: Low
**Dependencies**: [ST6]
**Enables**: []

### Node O4
**Content**: *Take the reciprocal of \(D\) to obtain the original total
↪ time \(= \frac{1}{D} \approx 365\) days.*
**Abstraction Level**: OPERATION
**Type**: calculation
**Summary**: final-answer calc
**Primary Auditor**: T_Auto
**Secondary Auditors**: [T_LLM]
**Verification Complexity**: Low
**Dependencies**: [ST7]
**Enables**: []

---

## Finalized Edges

```json
[
  /* Decomposes-to hierarchy */
  {"from":"G1","to":"S1","type":"decomposes_to",
  "strength":"strong","confidence":0.95},
  {"from":"S1","to":"T1","type":"decomposes_to",
  "strength":"strong","confidence":0.94},
  {"from":"T1","to":"ST1","type":"decomposes_to",
  "strength":"strong","confidence":0.94},
  {"from":"T1","to":"ST2","type":"decomposes_to",
  "strength":"strong","confidence":0.94},
  {"from":"T1","to":"ST3","type":"decomposes_to",
  "strength":"strong","confidence":0.94},
  {"from":"T1","to":"ST4","type":"decomposes_to",
  "strength":"strong","confidence":0.94},
  {"from":"T1","to":"ST5","type":"decomposes_to",
  "strength":"strong","confidence":0.94},
  {"from":"T1","to":"ST6","type":"decomposes_to",
  "strength":"strong","confidence":0.94},
  {"from":"T1","to":"ST7","type":"decomposes_to",
  "strength":"strong","confidence":0.94},
  {"from":"ST1","to":"O1","type":"decomposes_to",
  "strength":"strong","confidence":0.93},
```

```
{"from":"ST2","to":"O2","type":"decomposes_to",
 "strength":"strong","confidence":0.93},
{"from":"ST3","to":"O1","type":"decomposes_to",
 "strength":"strong","confidence":0.93},
{"from":"ST4","to":"O2","type":"decomposes_to",
 "strength":"strong","confidence":0.93},
{"from":"ST5","to":"O2","type":"decomposes_to",
 "strength":"strong","confidence":0.93},
{"from":"ST6","to":"O3","type":"decomposes_to",
 "strength":"strong","confidence":0.93},
{"from":"ST7","to":"O4","type":"decomposes_to",
 "strength":"strong","confidence":0.93},

/* Lateral dependencies */
{"from":"ST3","to":"ST1","type":"depends_on",
 "strength":"strong","confidence":0.90},
{"from":"ST4","to":"ST2","type":"depends_on",
 "strength":"strong","confidence":0.90},
{"from":"ST4","to":"ST3","type":"depends_on",
 "strength":"strong","confidence":0.90},
{"from":"ST5","to":"ST4","type":"depends_on",
 "strength":"strong","confidence":0.90},
{"from":"ST6","to":"ST5","type":"depends_on",
 "strength":"strong","confidence":0.90},
{"from":"ST7","to":"ST6","type":"depends_on",
 "strength":"strong","confidence":0.90},

/* Enables (operational support) */
{"from":"O1","to":"ST3","type":"enables",
 "strength":"strong","confidence":0.88},
{"from":"O2","to":"ST4","type":"enables",
 "strength":"strong","confidence":0.88},
{"from":"O3","to":"ST6","type":"enables",
 "strength":"strong","confidence":0.88},
{"from":"O4","to":"ST7","type":"enables",
 "strength":"strong","confidence":0.88},

/* Support / exemplify / validate relationships */
{"from":"O1","to":"ST1","type":"exemplifies",
 "strength":"strong","confidence":0.85},
{"from":"O1","to":"ST3","type":"exemplifies",
 "strength":"strong","confidence":0.85},
{"from":"O2","to":"ST2","type":"exemplifies",
 "strength":"strong","confidence":0.85},
{"from":"O2","to":"ST4","type":"exemplifies",
 "strength":"strong","confidence":0.85},
{"from":"O3","to":"ST6","type":"exemplifies",
 "strength":"strong","confidence":0.85},
{"from":"O4","to":"ST7","type":"exemplifies",
 "strength":"strong","confidence":0.85},
{"from":"O1","to":"ST3","type":"validates",
 "strength":"strong","confidence":0.84},
{"from":"O2","to":"ST4","type":"validates",
 "strength":"strong","confidence":0.84},
{"from":"O3","to":"ST6","type":"validates",
 "strength":"strong","confidence":0.84},
{"from":"O4","to":"ST7","type":"validates",
 "strength":"strong","confidence":0.84}
]
```

*The edge list above contains every relationship identified in the analysis: hierarchical *decomposes_to*, logical *depends_on*, operational *enables*, and verification *exemplifies* / *validates* edges.*

---

## Verification Flow

1. **Entry Points** – Nodes with no incoming edges: **G1** (goal) and
   ↪ **S1** (strategy) are the root audit entry points.
2. **Critical Path** – The sequence that must be verified in order to
   ↪ guarantee correctness of the final answer:

   `G1 ->S1 ->T1 ->ST1 ->ST3 ->ST5 ->ST6 ->ST7 ->O4`

   (Goal ->Strategy ->Tactic ->reduced-output inference ->
   ↪ remaining-portion time ->algebraic combination ->solve for \(D\) ->
   ↪ final conclusion).

3. **Parallel Tracks** – Independent branches that can be audited
   ↪ concurrently:

   - **Branch A**: `ST1 ->O1` (reduced-output computation).
   - **Branch B**: `ST2 ->O2` (first-third time).
   - **Branch C**: `ST4 ->O2` (total-time equation).
   - **Branch D**: `ST5 ->O2` (fraction combination).

   All arithmetic steps (ST1-ST7 and O1-O4) are pure numeric operations
   ↪ and can be run in parallel after their logical prerequisites are
   ↪ satisfied.

4. **Exit Points** – Nodes whose successful audit marks the HDAG as
   ↪ completely validated: **O3** (verification of the solved \(D\)) and
   ↪ **O4**
(verification of the final missing-days value).

---

## HDAG Validation Report
- **Acyclicity Check**: No cycles detected – the graph is a directed
  ↪ acyclic graph.
- **Connectivity Check**: Every node is reachable from the root node
  ↪ **G1** either directly or via intermediate nodes; there are no
  ↪ isolated
fragments.
- **Completeness Check**: All 14 original segments are present and
  ↪ their textual content has been preserved unchanged.
- **Auditability Check**: Every node lists primary and secondary
  ↪ auditors, a verification-complexity label, and explicit
  ↪ dependency/enabler fields,
giving a clear audit trail for both automated and human reviewers.

---

## Complete HDAG JSON

```json
{
  "nodes": [
    {
      "id": "G1",
      "content": "A construction company was building a tunnel. When 1/3
      ↪ of the tunnel was completed at the original speed, they started
      ↪ using new
equipment, which increased the construction speed by 20 % and reduced
  ↪ the working hours to 80 % of the original. As a result, it took a
  ↪ total of
```

```
185 days to complete the tunnel. If they had not used the new equipment
↪ and continued at the original speed, it would have taken ______ days
↪ to
complete the tunnel.",
    "abstraction_level": "GOAL",
    "type": "objective_statement",
    "summary": "find missing days",
    "primary_auditor": "T_Human",
    "secondary_auditors": ["T_LLM"],
    "complexity": "High"
  },
  {
    "id": "S1",
    "content": "Model the problem by relating speed, working hours,
    ↪ and daily output. The original daily output is D. After the
    ↪ equipment change
the speed rises 20 % while hours fall to 80 % ->new daily output
↪ 0.96D.",
    "abstraction_level": "STRATEGY",
    "type": "approach_description",
    "summary": "model speed & hours",
    "primary_auditor": "T_LLM",
    "secondary_auditors": ["T_Auto"],
    "complexity": "Medium"
  },
  {
    "id": "T1",
    "content": "Let the tunnel length be 1 unit and the original daily
    ↪ output be D. Build 1/3 at rate D; the remaining 2/3 at rate
    ↪ 0.96D. Set up
the time equation and solve for D.",
    "abstraction_level": "TACTIC",
    "type": "method_definition",
    "summary": "define variables, solve",
    "primary_auditor": "T_Auto",
    "secondary_auditors": ["T_LLM"],
    "complexity": "Low"
  },
  {
    "id": "ST1",
    "content": "Compute the reduced daily output: 0.96D.",
    "abstraction_level": "STEP",
    "type": "inference",
    "summary": "compute reduced output",
    "primary_auditor": "T_Auto",
    "secondary_auditors": ["T_LLM"],
    "complexity": "Low"
  },
  {
    "id": "ST2",
    "content": "Time for the first third: t1 = 1/(3D).",
    "abstraction_level": "STEP",
    "type": "inference",
    "summary": "first-third time",
    "primary_auditor": "T_Auto",
    "secondary_auditors": ["T_LLM"],
    "complexity": "Low"
  },
  {
    "id": "ST3",
    "content": "Time for the remaining two-thirds: t2 = (2/3)/(0.96D)
    ↪ = 5/(8D).",
    "abstraction_level": "STEP",
    "type": "inference",
    "summary": "remaining portion time",
```

```
      "primary_auditor": "T_Auto",
      "secondary_auditors": ["T_LLM"],
      "complexity": "Low"
    },
    {
      "id": "ST4",
      "content": "Total-time equation: t1 + t2 = 185.",
      "abstraction_level": "STEP",
      "type": "inference",
      "summary": "total-time equation",
      "primary_auditor": "T_Auto",
      "secondary_auditors": ["T_LLM"],
      "complexity": "Low"
    },
    {
      "id": "ST5",
      "content": "Combine fractions: (1/3 + 5/8)/D = 185 ->(23/24)/D =
      ↪ 185.",
      "abstraction_level": "STEP",
      "type": "algebraic_combination",
      "summary": "algebraic combination",
      "primary_auditor": "T_Auto",
      "secondary_auditors": ["T_LLM"],
      "complexity": "Low"
    },
    {
      "id": "ST6",
      "content": "Solve for D: D = (23/24)/185 \approx 0.0054.",
      "abstraction_level": "STEP",
      "type": "solution_step",
      "summary": "solve for D",
      "primary_auditor": "T_Auto",
      "secondary_auditors": ["T_LLM"],
      "complexity": "Low"
    },
    {
      "id": "ST7",
      "content": "Original total time = 1/D \approx 365 days.",
      "abstraction_level": "STEP",
      "type": "conclusion_inference",
      "summary": "original schedule result",
      "primary_auditor": "T_Auto",
      "secondary_auditors": ["T_LLM"],
      "complexity": "Low"
    },
    {
      "id": "O1",
      "content": "Numeric calculation 0.96 = 20% increase \times 80%
      ↪ reduction.",
      "abstraction_level": "OPERATION",
      "type": "calculation",
      "summary": "reduced-output calc",
      "primary_auditor": "T_Auto",
      "secondary_auditors": ["T_LLM"],
      "complexity": "Low"
    },
    {
      "id": "O2",
      "content": "Add the two time expressions to get 185 days.",
      "abstraction_level": "OPERATION",
      "type": "calculation",
      "summary": "build time equation",
      "primary_auditor": "T_Auto",
      "secondary_auditors": ["T_LLM"],
      "complexity": "Low"
```

```
    },
    {
      "id": "O3",
      "content": "Compute D = (23/24)/185 \approx 0.0054.",
      "abstraction_level": "OPERATION",
      "type": "calculation",
      "summary": "compute D",
      "primary_auditor": "T_Auto",
      "secondary_auditors": ["T_LLM"],
      "complexity": "Low"
    },
    {
      "id": "O4",
      "content": "Take reciprocal of D ->1/D \approx 365 days.",
      "abstraction_level": "OPERATION",
      "type": "calculation",
      "summary": "final answer calc",
      "primary_auditor": "T_Auto",
      "secondary_auditors": ["T_LLM"],
      "complexity": "Low"
    }
  ],
  "edges": [
    /* (the JSON edge list shown earlier) */
  ],
  "metadata": {
    "total_nodes": 14,
    "total_edges": 28,
    "primary_auditor_distribution": {
      "T_Human": 1,
      "T_LLM": 1,
      "T_Auto": 12
    }
  }
}
```

*The JSON follows the exact structure used throughout the analysis,
↪ making it ready for ingestion by downstream validation tools.*

---

*End of HDAG specification.*

## C.2 HUMAN-FRIENDLY AUDIT EXAMPLE

This section demonstrates how human auditors interact with the TRUST system through a structured audit interface with human friendly presentation. The example shows a typical reasoning segment decomposition where auditors evaluate logical dependencies and inference validity.

### C.2.1 AUDIT INTERFACE STRUCTURE

The human audit form presents reasoning segments with four key components that enable focused evaluation:

**Prompt Context**: The original problem statement that establishes the reasoning foundation.
**Dependencies**: Previously verified reasoning steps that the current segment builds upon.
**Current Reasoning**: The specific reasoning step under evaluation.
**Implications**: The logical consequences that flow from this reasoning step.

### C.2.2 EXAMPLE AUDIT CASE

**Prompt Context:** Martha needs 4 cups of berries and 2 cups of heavy cream to make 1 quart of ice cream. She wants to make 1 quart of strawberry ice cream and 1 quart of raspberry ice cream. At the farmers market, 2-cup packages of strawberries cost $3.00 each and 2-cup packages of raspberries cost $5.00 each. Heavy cream is sold in 4-cup containers for $4.00. How much will it cost her to make 1 quart of each ice cream?

**Dependencies (Verified Step S1):** "For each quart of ice cream she needs: 4 cups of berries, 2 cups of heavy cream. She wants one quart of strawberry ice cream and one quart of raspberry ice cream."

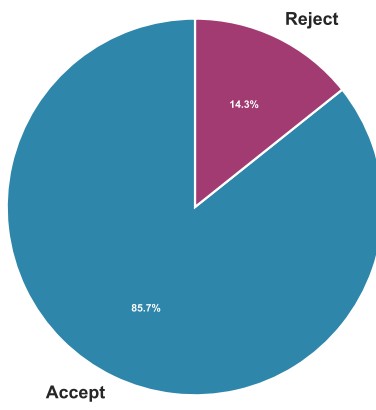

Figure 9: Human Audit Results.

**Current Reasoning Step (T1):** "For 1 quart of ice cream, she needs 4 cups of berries and 2 cups of heavy cream. Since she's making both strawberry and raspberry flavors, she'll need double that amount. For both ice creams combined, she needs 8 cups of berries and 4 cups of heavy cream."

**Logical Implications (T3):** "She needs 8 cups total berries. Since each package contains 2 cups, she needs $8/2 = 4$ berry packages total (split between strawberry and raspberry varieties)."

**Self-Verification (ST1):** "Each quart requires 4 cups of berries and 2 cups of cream. For two quarts: 8 cups berries and 4 cups cream. This calculation is correct."

### C.2.3 AUDIT DECISION FRAMEWORK

Human auditors evaluate each reasoning segment using structured criteria:

- **Logical Consistency**: Does the reasoning step follow logically from its dependencies?

- **Mathematical Accuracy**: Are calculations and quantitative reasoning correct?

- **Completeness**: Does the reasoning address all relevant aspects of the problem?

- **Clarity**: Is the reasoning step clearly articulated and understandable?

**Ground Truth Evaluation**: Pass ✓

The reasoning correctly identifies that making two quarts requires doubling the per-quart ingredient requirements, performs accurate arithmetic ($2 \times 4 = 8$ cups berries, $2 \times 2 = 4$ cups cream), and includes appropriate self-verification to confirm the calculation.

This structured approach enables human auditors to provide focused, reliable evaluations while maintaining the modular verification principles of the TRUST framework.

## C.3 HDAG EXAMPLE 1 - MATH PROBLEM

Problem: Marie is planning to buy a new cash register for her bakery that costs $1040. Every day Marie sells 40 loaves of bread for $2 each and 6 cakes for $12 each. She has to pay $20 each day for rent and $2 each day for electricity. How many days' worth of profits will it take for Marie to pay for the cash register?

### C.3.1 DEEPSEEK-R1-8B COT DECOMPOSITION

```
{
  "title": "Marie's Cash Register Days",
  "nodes": [
    {"id":"G1","label":"Problem statement","content":"## Problem
    ↪ Statement:\nMarie is planning to buy a new cash register for her
    ↪ bakery that costs $1040. Every day Marie sells 40 loaves of bread
    ↪ for $2 each and 6 cakes for $12 each. She has to pay $20 each day
    ↪ for rent and $2 each day for electricity. How many days' worth of
    ↪ profits will it take for Marie to pay for the cash
    ↪ register?","abstraction_level":"GOAL",
    "type":"objective_statement","primary_auditor":"T_Human",
    "complexity":"Low"},
    {"id":"S1","label":"Plan overview","content":"Okay, so Marie wants
    ↪ to buy a new cash register for her bakery, and it costs $1040. I
    ↪ need to figure out how many days' worth of profits it will take
    ↪ her to afford this cash register. Let me break this down step by
    ↪ step.","abstraction_level":"STRATEGY","primary_auditor":"T_LLM",
    "complexity":"Low"},
    {"id":"S2","label":"Daily revenue","content":"First, I should
    ↪ understand what her daily income is. She sells 40 loaves of bread
    ↪ each day, and each loaf is $2. So, the revenue from bread would
    ↪ be 40 times $2. Let me calculate that: 40 * 2 = $80. Then, she
    ↪ sells 6 cakes each day, and each cake is $12. So, the revenue
    ↪ from cakes is 6 * 12 = $72. Adding those together, her total
    ↪ daily revenue is $80 + $72 =
    ↪ $152.","abstraction_level":"STRATEGY","primary_auditor":"T_LLM",
    "complexity":"Low"},
    {"id":"S3","label":"Daily profit","content":"Next, I need to
    ↪ calculate her daily expenses. She has to pay $20 each day for
    ↪ rent and $2 each day for electricity. So, adding those together:
    ↪ $20 + $2 = $22. That means her daily profit is the revenue minus
    ↪ the expenses. So, $152 (revenue) - $22 (expenses) = $130 per
    ↪ day.","abstraction_level":"STRATEGY","primary_auditor":"T_LLM",
    "complexity":"Low"},
    {"id":"T1","label":"Setup division","content":"Now, the cash
    ↪ register costs $1040, and she makes $130 each day. To find out
    ↪ how many days it will take her to earn enough to buy the cash
    ↪ register, I should divide the total cost by her daily profit. So,
    ↪ $1040 divided by $130 per day. Let me do that division: 1040 /
    ↪ 130.","abstraction_level":"TACTIC","primary_auditor":"T_LLM",
    "complexity":"Low"},
    {"id":"T2","label":"Verify quotient","content":"Hmm, 130 times 8 is
    ↪ 1040 because 130 * 8 = 1040. So, it would take her 8 days to earn
    ↪ enough money to buy the cash
    ↪ register.","abstraction_level":"TACTIC","primary_auditor":"T_LLM",
    "complexity":"Low"},
    {"id":"ST1","label":"Double-check math","content":"Wait, let me
    ↪ double-check my calculations to make sure I didn't make any
    ↪ mistakes. Revenue from bread: 40 * 2 = 80. Revenue from cakes: 6
    ↪ * 12 = 72. Total revenue: 80 + 72 = 152. Expenses: 20 + 2 = 22.
    ↪ Daily profit: 152 - 22 = 130. Cash register cost: 1040. 1040 /
    ↪ 130 = 8. Yep, that seems
    ↪ correct.","abstraction_level":"STEP","primary_auditor":"T_Human",
    "complexity":"Low"},
```

```
    {"id":"O1","label":"Bread revenue","content":"40 * 2 =
    ↪ $80","abstraction_level":"OPERATION","primary_auditor":"T_Auto",
    "complexity":"Low"},
    {"id":"O2","label":"Cakes revenue","content":"6 * 12 =
    ↪ $72","abstraction_level":"OPERATION","primary_auditor":"T_Auto",
    "complexity":"Low"},
    {"id":"O3","label":"Total revenue","content":"$80 + $72 =
    ↪ $152","abstraction_level":"OPERATION","primary_auditor":"T_Auto",
    "complexity":"Low"},
    {"id":"O4","label":"Daily expenses","content":"$20 + $2 =
    ↪ $22","abstraction_level":"OPERATION","primary_auditor":"T_Auto",
    "complexity":"Low"},
    {"id":"O5","label":"Daily profit","content":"$152 - $22 =
    ↪ $130","abstraction_level":"OPERATION","primary_auditor":"T_Auto",
    "complexity":"Low"},
    {"id":"O6","label":"Days needed","content":"1040 / 130 =
    ↪ 8","abstraction_level":"OPERATION","primary_auditor":"T_Auto",
    "complexity":"Low"},
    {"id":"O7","label":"Final answer 8","content":"So, it will take
    ↪ Marie 8 days to save enough money to buy the cash
    ↪ register.","abstraction_level":"OPERATION",
    "primary_auditor":"T_Auto", "complexity":"Low"}
  ],
  "edges": [
    {"from":"G1","to":"S1","relationship":"decomposes_to",
    "strength":"strong","confidence":0.97},
    {"from":"S1","to":"S2","relationship":"decomposes_to",
    "strength":"strong","confidence":0.96},
    {"from":"S1","to":"S3","relationship":"decomposes_to",
    "strength":"strong","confidence":0.96},
    {"from":"S2","to":"O1","relationship":"decomposes_to",
    "strength":"strong","confidence":0.99},
    {"from":"S2","to":"O2","relationship":"decomposes_to",
    "strength":"strong","confidence":0.99},
    {"from":"S2","to":"O3","relationship":"decomposes_to",
    "strength":"strong","confidence":0.99},
    {"from":"S3","to":"O4","relationship":"decomposes_to",
    "strength":"strong","confidence":0.99},
    {"from":"S3","to":"O5","relationship":"decomposes_to",
    "strength":"strong","confidence":0.99},
    {"from":"T1","to":"O6","relationship":"decomposes_to",
    "strength":"strong","confidence":0.99},
    {"from":"T2","to":"O6","relationship":"validates",
    "strength":"strong","confidence":0.97},
    {"from":"O1","to":"ST1","relationship":"enables",
    "strength":"strong","confidence":0.95},
    {"from":"O5","to":"T1","relationship":"enables",
    "strength":"strong","confidence":0.95},
    {"from":"ST1","to":"O7","relationship":"decomposes_to",
    "strength":"medium","confidence":0.90}
  ],
  "metadata": {
    "total_nodes": 14,
    "total_edges": 13,
    "auditor_distribution": { "T_Auto": 7, "T_LLM": 5, "T_Human": 2 }
  }
}
```

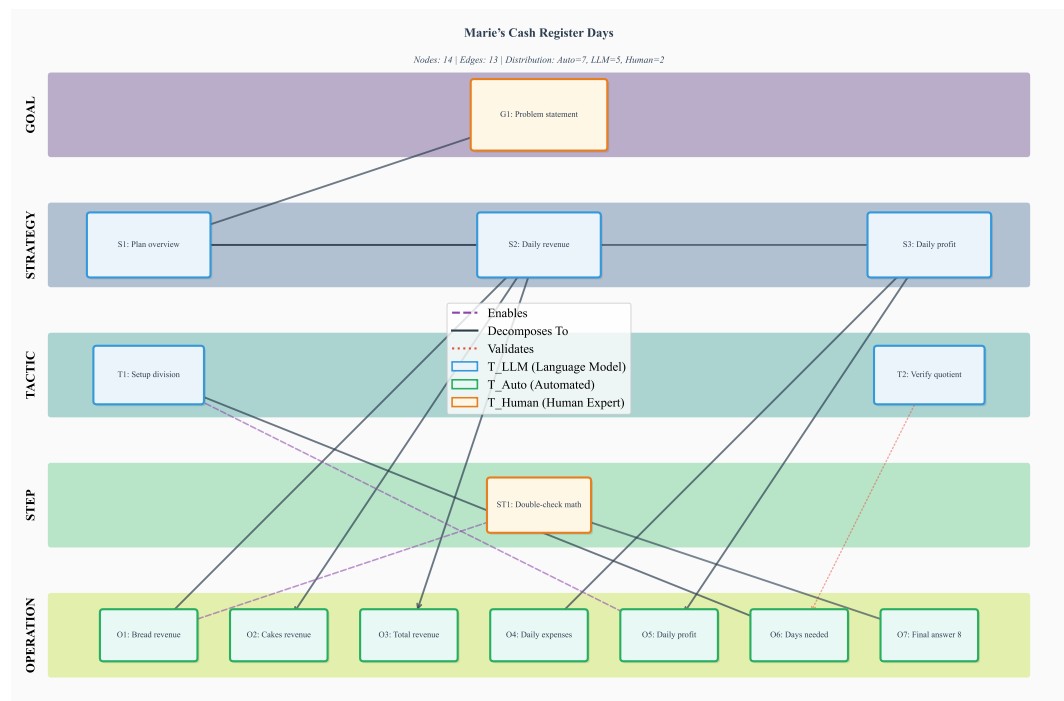

Figure 10: HDAG Example 1 from Deepseek-R1-8B CoT: Math Problem.

### C.3.2 GPT-OSS-20B CoT DECOMPOSITION

```
{
  "title": "Marie's Cash Register - Step-by-Step",
  "nodes": [
   {"id":"G1","label":"Understand problem","content":"**1. Understand
   ↪ the problem**\n\nMarie wants to know how many days of operating
   ↪ her bakery will be needed to earn enough profit to buy a cash
   ↪ register that costs **$1040**. \nEach day she:\n\n- Sells 40
   ↪ loaves of bread at **$2** each ->revenue from bread \n- Sells 6
   ↪ cakes at **$12** each ->revenue from cakes \n- Pays **$20** for
   ↪ rent and **$2** for electricity ->daily expenses \n\nWe need the
   ↪ daily profit (revenue minus expenses) and then divide the
   ↪ register cost by that daily profit to find the number of
   ↪ days.","abstraction_level":"GOAL","type":"objective_statement","primary_auditor":"T_Huma
   {"id":"S1","label":"Steps overview","content":"**2. Show each step
   ↪ of your work**","abstraction_level":"STRATEGY",
   "primary_auditor":"T_LLM"},
   {"id":"S2","label":"Daily revenue","content":"1. **Daily
   ↪ revenue**\n\n - Bread: \\(40 \\text{ loaves} \\times \\$2 =
   ↪ \\$80\\)\n - Cakes: \\(6 \\text{ cakes} \\times \\$12 =
   ↪ \\$72\\)\n\n \\[\n \\text{Total revenue} = \\$80 + \\$72 =
   ↪ \\$152\n
   ↪ \\]","abstraction_level":"STRATEGY","primary_auditor":"T_LLM"},
   {"id":"S3","label":"Daily expenses","content":"2. **Daily
   ↪ expenses**\n\n \\[\n \\text{Rent} = \\$20,\\quad
   ↪ \\text{Electricity} = \\$2\n \\]\n \\[\n \\text{Total expenses} =
   ↪ \\$20 + \\$2 = \\$22\n
   ↪ \\]","abstraction_level":"STRATEGY","primary_auditor":"T_LLM"},
   {"id":"S4","label":"Daily profit","content":"3. **Daily profit**\n\n
   ↪ \\[\n \\text{Profit} = \\text{Revenue} - \\text{Expenses}\n \\]\n
   ↪ \\[\n \\text{Profit} = \\$152 - \\$22 = \\$130\n
   ↪ \\]","abstraction_level":"STRATEGY","primary_auditor":"T_LLM"},
```

```
{"id":"S5","label":"Days formula","content":"4. **Number of days to
↪ cover the cash register**\n\n \\[\n \\text{Days} =
↪ \\frac{\\text{Cost of register}}{\\text{Daily profit}}\n \\]\n
↪ \\[\n \\text{Days} = \\frac{\\$1040}{\\$130} = 8\n
↪ \\]","abstraction_level":"STRATEGY","primary_auditor":"T_LLM"},
{"id":"T1","label":"Revenue tactic","content":"- Compute daily
↪ revenue from bread and
↪ cakes.","abstraction_level":"TACTIC","primary_auditor":"T_LLM"},
{"id":"T2","label":"Expenses tactic","content":"- Compute daily
↪ expenses (rent +
↪ electricity).","abstraction_level":"TACTIC","primary_auditor":"T_LLM"},
{"id":"T3","label":"Profit tactic","content":"- Compute daily profit
↪ as revenue minus
↪ expenses.","abstraction_level":"TACTIC","primary_auditor":"T_LLM"},
{"id":"T4","label":"Days tactic","content":"- Divide register cost
↪ by daily profit to get
↪ days.","abstraction_level":"TACTIC","primary_auditor":"T_LLM"},
{"id":"ST1","label":"Verify math","content":"**3. Verify your
↪ calculations**\n\n- Check revenue: \\(40 \\times 2 = 80\\), \\(6
↪ \\times 12 = 72\\), \\(80 + 72 = 152\\). \n- Check expenses:
↪ \\(20 + 2 = 22\\). \n- Profit: \\(152 - 22 = 130\\). \n- Days:
↪ \\(130 \\times 8 = 1040\\). \n\nEverything is
↪ consistent.","abstraction_level":"STEP","primary_auditor":"T_Human"},
{"id":"O1","label":"Bread revenue","content":"\\(40 \\times 2 =
↪ 80\\)","abstraction_level":"OPERATION","primary_auditor":"T_Auto"},
{"id":"O2","label":"Cakes revenue","content":"\\(6 \\times 12 =
↪ 72\\)","abstraction_level":"OPERATION","primary_auditor":"T_Auto"},
{"id":"O3","label":"Total revenue","content":"\\(80 + 72 =
↪ 152\\)","abstraction_level":"OPERATION","primary_auditor":"T_Auto"},
{"id":"O4","label":"Total expenses","content":"\\(20 + 2 =
↪ 22\\)","abstraction_level":"OPERATION","primary_auditor":"T_Auto"},
{"id":"O5","label":"Daily profit","content":"\\(152 - 22 =
↪ 130\\)","abstraction_level":"OPERATION","primary_auditor":"T_Auto"},
{"id":"O6","label":"Days needed","content":"\\(\\frac{1040}{130} =
↪ 8\\)","abstraction_level":"OPERATION","primary_auditor":"T_Auto"},
{"id":"O7","label":"Verify product","content":"\\(130 \\times 8 =
↪ 1040\\)","abstraction_level":"OPERATION","primary_auditor":"T_Auto"}
],
"edges": [
 {"from":"G1","to":"S1","relationship":"decomposes_to"},
 {"from":"G1","to":"S2","relationship":"decomposes_to"},
 {"from":"G1","to":"S3","relationship":"decomposes_to"},
 {"from":"G1","to":"S4","relationship":"decomposes_to"},
 {"from":"G1","to":"S5","relationship":"decomposes_to"},
 {"from":"S2","to":"O1","relationship":"decomposes_to"},
 {"from":"S2","to":"O2","relationship":"decomposes_to"},
 {"from":"S2","to":"O3","relationship":"decomposes_to"},
 {"from":"S3","to":"O4","relationship":"decomposes_to"},
 {"from":"S4","to":"O5","relationship":"decomposes_to"},
 {"from":"S5","to":"O6","relationship":"decomposes_to"},
 {"from":"ST1","to":"O7","relationship":"decomposes_to"},
 {"from":"O3","to":"O5","relationship":"enables"},
 {"from":"O4","to":"O5","relationship":"enables"},
 {"from":"O5","to":"O6","relationship":"enables"},
 {"from":"O7","to":"O6","relationship":"validates"}
],
"metadata": {
 "total_nodes": 17,
 "total_edges": 16,
 "auditor_distribution": { "T_Auto": 7, "T_LLM": 9, "T_Human": 2 }
 }
}
```

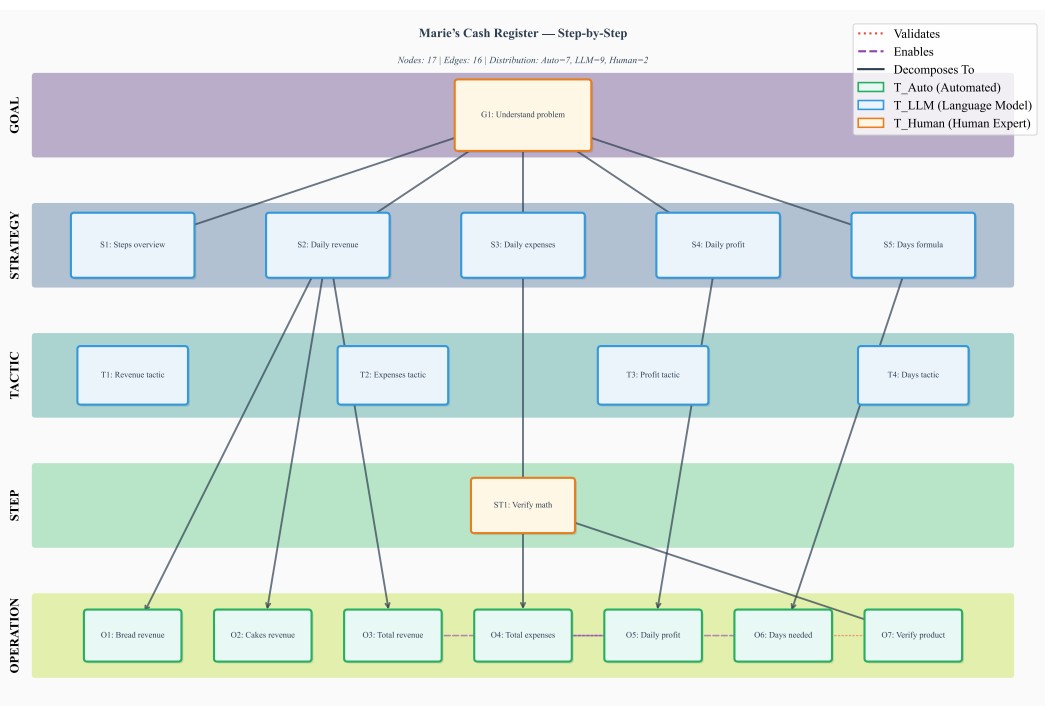

Figure 11: HDAG Example 1 from GPT-OSS-20B CoT: Math Problem.

## C.4 HDAG EXAMPLE 2 - MATH PROBLEM

Problem: Alec is running for Class President. He thinks that if he can get three-quarters of the class to vote for him then there is no chance anyone else can beat him. Half of the class have already said they will vote for him but out of the remaining students, only 5 have said they are thinking about voting for him. He surveys the students who are thinking about voting for someone else, and changes his flyers to reflect the issues these students are concerned about. This results in a fifth of these students saying they'll vote for him. If Alec's class has 60 students and everyone who said they will vote for him does so, how many more votes does Alec need to reach his goal number of votes?

### C.4.1 DEEPSEEK-R1-8B CoT DECOMPOSITION

```
{
  "title": "Alec's Class President Votes",
  "nodes": [
    {"id":"G1","label":"Goal statement","content":"Alec is running for
    ↪ Class President...","abstraction_level":"GOAL",
    "type":"objective_statement","primary_auditor":"T_Human"},
    {"id":"S1","label":"Plan overview","content":"Compute goal votes,
    ↪ count current and added, compare
    ↪ gap.","abstraction_level":"STRATEGY","primary_auditor":"T_LLM"},
    {"id":"T1","label":"Needs 45 votes","content":"3/4 x 60 =
    ↪ 45.","abstraction_level":"TACTIC","primary_auditor":"T_LLM"},
    {"id":"T2","label":"30 committed","content":"Half of 60 = 30 already
    ↪ pledged.","abstraction_level":"TACTIC","primary_auditor":"T_LLM"},
    {"id":"T3","label":"5 lean,5 convert","content":"5 thinking about
    ↪ him; 1/5 of 25 switch = 5.",
    "abstraction_level":"TACTIC","primary_auditor":"T_LLM"},
    {"id":"ST1","label":"Gap check","content":"45 - 40 = 5 more votes
    ↪ needed.",
    "abstraction_level":"STEP","primary_auditor":"T_Human"},
    {"id":"O1","label":"3/4 of
    ↪ 60","content":"(3/4)x60=45","abstraction_level":"OPERATION",
    "primary_auditor":"T_Auto"},
    {"id":"O2","label":"Half class","content":"60/2=30",
    "abstraction_level":"OPERATION","primary_auditor":"T_Auto"},
    {"id":"O3","label":"Remaining","content":"60-30=30 remaining",
    "abstraction_level":"OPERATION","primary_auditor":"T_Auto"},
    {"id":"O4","label":"Convert votes","content":"1/5 of 25=5",
    "abstraction_level":"OPERATION","primary_auditor":"T_Auto"},
    {"id":"O5","label":"Total votes","content":"30+5+5=40",
    "abstraction_level":"OPERATION","primary_auditor":"T_Auto"},
    {"id":"O6","label":"Votes gap","content":"45-40=5",
    "abstraction_level":"OPERATION","primary_auditor":"T_Auto"},
    {"id":"O7","label":"Final answer","content":"Alec needs 5 more
    ↪ votes.","abstraction_level":"OPERATION","primary_auditor":"T_Auto"}
  ],
  "edges": [
    {"from":"G1","to":"S1","relationship":"decomposes_to"},
    {"from":"S1","to":"T1","relationship":"decomposes_to"},
    {"from":"S1","to":"T2","relationship":"decomposes_to"},
    {"from":"S1","to":"T3","relationship":"decomposes_to"},
    {"from":"T1","to":"O1","relationship":"decomposes_to"},
    {"from":"T2","to":"O2","relationship":"decomposes_to"},
    {"from":"T2","to":"O3","relationship":"decomposes_to"},
    {"from":"T3","to":"O4","relationship":"decomposes_to"},
    {"from":"T3","to":"O5","relationship":"decomposes_to"},
    {"from":"O5","to":"ST1","relationship":"enables"},
    {"from":"O6","to":"ST1","relationship":"validates"},
    {"from":"ST1","to":"O7","relationship":"decomposes_to"}
  ],
  "metadata": {
    "total_nodes": 13,
```

```
    "total_edges": 12,
    "auditor_distribution": {"T_Auto":7,"T_LLM":4,"T_Human":2}
  }
}
```

Figure 12: HDAG Example 2 from Deepseek-R1-8B CoT: Math Problem.

### C.4.2 GPT-OSS-20B CoT Decomposition

```
{
  "title": "Alec's Votes",
  "nodes": [
    {"id":"G1","label":"Goal statement","content":"Alec wants 3/4 of 60
    ↪ = 45
    ↪ votes.","abstraction_level":"GOAL","type":"objective_statement",
    "primary_auditor":"T_Human"},
    {"id":"S1","label":"Plan overview","content":"Compute target, count
    ↪ votes, add converts, compare
    ↪ gap.","abstraction_level":"STRATEGY","primary_auditor":"T_LLM"},
    {"id":"T1","label":"Needs 45 votes","content":"Three-quarters of 60
    ↪ is 45.","abstraction_level":"TACTIC","primary_auditor":"T_LLM"},
    {"id":"T2","label":"30 committed","content":"Half of 60 already
    ↪ pledged = 30.","abstraction_level":"TACTIC",
    "primary_auditor":"T_LLM"},
    {"id":"T3","label":"5 converted","content":"From 25 opposing, 1/5
    ↪ switched = 5.","abstraction_level":"TACTIC",
    "primary_auditor":"T_LLM"},
    {"id":"T4","label":"Total 35","content":"30 committed + 5 converted
    ↪ = 35 total.","abstraction_level":"TACTIC",
    "primary_auditor":"T_LLM"},
    {"id":"ST1","label":"Gap check","content":"45-35 = 10 more
    ↪ needed.","abstraction_level":"STEP",
    "primary_auditor":"T_Human"},
    {"id":"O1","label":"3/4 of 60","content":"(3/4)x60=45",
    "abstraction_level":"OPERATION","primary_auditor":"T_Auto"},
    {"id":"O2","label":"Half class","content":"1/2x60=30",
    "abstraction_level":"OPERATION","primary_auditor":"T_Auto"},
    {"id":"O3","label":"Remaining","content":"60-30=30 remain",
    "abstraction_level":"OPERATION","primary_auditor":"T_Auto"},
    {"id":"O4","label":"Others 25","content":"30-5=25",
    "abstraction_level":"OPERATION","primary_auditor":"T_Auto"},
    {"id":"O5","label":"Convert 5","content":"1/5x25=5",
    "abstraction_level":"OPERATION","primary_auditor":"T_Auto"},
    {"id":"O6","label":"Total votes","content":"30+5=35",
    "abstraction_level":"OPERATION","primary_auditor":"T_Auto"},
    {"id":"O7","label":"Votes gap","content":"45-35=10",
    "abstraction_level":"OPERATION","primary_auditor":"T_Auto"},
    {"id":"O8","label":"Final answer","content":"Alec needs 10 more
    ↪ votes.",
    "abstraction_level":"OPERATION","primary_auditor":"T_Auto"}
  ],
  "edges": [
    {"from":"G1","to":"S1","relationship":"decomposes_to"},
    {"from":"S1","to":"T1","relationship":"decomposes_to"},
    {"from":"S1","to":"T2","relationship":"decomposes_to"},
    {"from":"S1","to":"T3","relationship":"decomposes_to"},
    {"from":"S1","to":"T4","relationship":"decomposes_to"},
    {"from":"T1","to":"O1","relationship":"decomposes_to"},
    {"from":"T2","to":"O2","relationship":"decomposes_to"},
    {"from":"T2","to":"O3","relationship":"decomposes_to"},
    {"from":"T3","to":"O4","relationship":"decomposes_to"},
    {"from":"T3","to":"O5","relationship":"decomposes_to"},
    {"from":"T4","to":"O6","relationship":"decomposes_to"},
    {"from":"O6","to":"ST1","relationship":"enables"},
    {"from":"O7","to":"ST1","relationship":"validates"},
    {"from":"ST1","to":"O8","relationship":"decomposes_to"}
  ],
  "metadata": {
    "total_nodes": 15,
    "total_edges": 14,
    "auditor_distribution": {"T_Auto":8,"T_LLM":4,"T_Human":2}
```

```
      }
   }
```

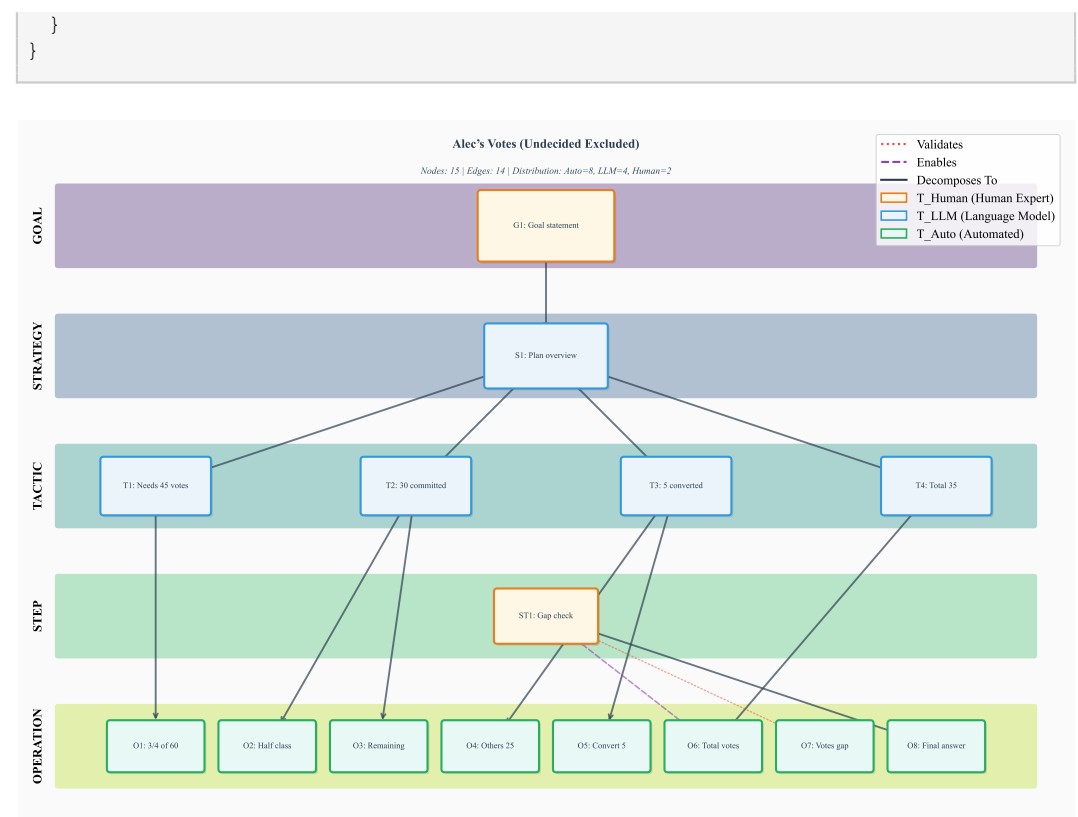

Figure 13: HDAG Example 2 from GPT-OSS-20B CoT: Math Problem.

## C.5 DATASETS

We utilize Chain-of-Thought (CoT) reasoning datasets spanning multiple domains following recent surveys Tan et al. (2024). Our evaluation uses a carefully curated multi-domain dataset designed to test bias mitigation and auditing effectiveness across diverse reasoning tasks.

**Primary Evaluation Dataset.** MMLU-Pro-CoT-Train (UW-Madison-Lee-Lab, 2024): 200 samples with ground truth annotations for individual reasoning steps and final answers across professional domains, enabling systematic evaluation of correctness and faithfulness at both step and trace levels. This dataset provides comprehensive coverage of advanced knowledge areas including engineering, mathematics, physics, chemistry, biology, and computer science, making it ideal for evaluating reasoning trace quality in technical domains.

**Multi-Domain Bias Evaluation Dataset.** Our curated dataset of 300 questions (50 per domain) sourced from established HuggingFace datasets across 6 diverse domains with comprehensive ground truth validation:

- **Medical** (medmcqa) (Pal et al., 2022): Clinical multiple-choice questions covering medical diagnosis, treatment protocols, pharmacology, and pathophysiology. This dataset represents real medical licensing exam questions, providing authentic clinical reasoning scenarios that require domain expertise and careful step-by-step analysis of patient presentations, differential diagnoses, and treatment decisions.

- **Mathematics** (gsm8k) (Cobbe et al., 2021a): Grade school arithmetic and algebra word problems requiring multi-step mathematical reasoning. These problems test fundamental quantitative reasoning skills including problem decomposition, arithmetic operations, proportional reasoning, and logical sequencing. The dataset emphasizes practical mathematical applications in everyday contexts such as financial calculations, measurement conversions, and basic geometry.

- **Science** (sciq) (Johannes Welbl, 2017): Multi-disciplinary science questions spanning physics, chemistry, biology, and earth sciences. Questions cover fundamental scientific concepts, experimental methodology, and theoretical principles. This dataset tests scientific reasoning capabilities including hypothesis formation, experimental design interpretation, causal relationship identification, and application of scientific laws across different domains.

- **Common Sense** (commonsense_qa (Talmor et al., 2019)): Everyday reasoning questions testing implicit knowledge about social situations, physical properties, causal relationships, and typical human behavior. These questions evaluate the model's ability to apply common-sense knowledge that is typically acquired through general life experience rather than formal education, including understanding of social norms, object properties, and cause-and-effect relationships.

- **Humanities** (squad) (Rajpurkar et al., 2016): Reading comprehension tasks requiring factual extraction and inference from historical, literary, and cultural texts. Questions test advanced comprehension skills including information synthesis, contextual interpretation, implicit meaning recognition, and factual accuracy verification across diverse humanistic domains.

- **Human Evaluation Subset**: 10 carefully selected math problems from openai/gsm8k used specifically for human-in-the-loop experiments with 15 PhD-level auditors, enabling direct comparison between human expert judgment and automated auditing approaches. All datasets undergo comprehensive quality validation including ground truth verification, format consistency checking, content quality assessment, and deduplication.

# D    THEORETICAL RESULTS

## D.1    CONSENSUS: SEAT LAYER, SEGMENT LAYER, AND TRACE LAYER

In this section we provide the theoretical analysis of TRUST present in Section 3 by characterizing the statistical safety against malicious auditors and the economics incentive/decentive gurantees for good and bad actors.

We analyze in three layers:

1. **Seat layer.** Within segment $s$, the $k_{t(s)}$ seats vote independently; computer seats are noiseless, LLMs/humans have error $\epsilon_t$, and human seats may be adversarial w.p. $\rho_{\mathrm{H}}$.

2. **Segment layer.** Define the *segment pass indicator*

$$B_s = \mathbf{1}\Big[\#\{\text{correct votes}\} \geq q_{t(s)}\Big], \tag{D.1}$$

   where $q_t = \lceil \tau\, k_t \rceil$ is the per-type quorum. Compute the exact pass probability $p_s = \Pr[B_s = 1]$.

3. **Trace layer.** Weight each segment outcome by $w_{t(s)}$, set

$$W = \sum_{s=1}^{S} w_{t(s)}\, B_s, \quad W_\beta = \beta \sum_s w_{t(s)}. \tag{D.2}$$

   Bound $\Pr[W < W_\beta]$ by Hoeffding and Chernoff using the $p_s$.

Next, we derive the exact pass rate for type $t$ in Lemma D.1

**Lemma D.1** (Exact pass probability for type $t$). For a segment type $t$ and parameters $(k_t, \epsilon_t, \rho_t)$ with $\rho_{\mathrm{C}} = \rho_{\mathrm{L}} = 0$. Then

$$p_t = \Pr\big[B_s = 1\big] = \sum_{m=0}^{k_t} \binom{k_t}{m} \rho_t^m (1-\rho_t)^{k_t-m} \sum_{c=q_t}^{k_t-m} \binom{k_t - m}{c} (1-\epsilon_t)^c\, \epsilon_t^{k_t-m-c}, \tag{D.3}$$

where $m$ malicious seats always vote wrong, and among the $k_t - m$ honest seats $c$ vote correctly.

*Proof.* First choose $m$ malicious seats (Binomial$(k_t, \rho_t)$), then among the remaining $k_t - m$ honest seats count $c \geq q_t$ correct votes (Binomial$(k_t - m, 1 - \epsilon_t)$). $\qquad\square$

On trace-level, we aggregate the results on segmend-level and denote $W = \sum_{s=1}^{S} w_{t(s)} B_s$ where $w_t$ is the weight for segment of type $t$. With $B_s \sim \text{Bernoulli}(p_s)$ independent and weights $w_s = w_{t(s)}$, the first two moment of weighted trace is given by

$$\mu_{\text{vote}} = \mathbb{E}[W] = \sum_{s=1}^{S} w_s\, p_s, \tag{D.4}$$

$$\sigma_{\text{vote}}^2 = \text{Var}(W) = \sum_{s=1}^{S} w_s^2\, p_s (1 - p_s) \leq \sum_{s=1}^{S} w_s^2 =: \sigma_{\max}^2. \tag{D.5}$$

**Proposition D.1** (Hoeffding and Chernoff bound). For any trace-level quorum threshold $\beta \in (0,1)$ define $W_\beta = \beta \sum_s w_s$ and let $W = \sum_s w_s B_s$. Then

$$\Pr[W < W_\beta] \;\leq\; \underbrace{\exp\!\left[-2\,(\mu_{\text{vote}} - W_\beta)^2/\sigma_{\max}^2\right]}_{\text{Hoeffding}} \;\wedge\; \underbrace{\min_{\lambda > 0} \exp\!\Big(\lambda W_\beta + \sum_{s=1}^{S} \ln\!\big(p_s\, e^{-\lambda w_s} + (1 - p_s)\big)\Big)}_{\text{Chernoff}}.$$

$$(D.6)$$

*Proof.* We proof the bounds separate as follow.

1. Hoeffding bound. Each summand $X_s := w_s B_s$ satisfies $0 \leq X_s \leq w_s$. Applying Hoeffding's inequality to $\sum_s (X_s - \mathbb{E}X_s)$ yields the first brace with denominator $\sum_s w_s^2 = \sigma_{\max}^2$ in (D.5).

2. Chernoff bound. For any $\lambda > 0$,

$$\Pr[W < W_\beta] \;\leq\; e^{\lambda W_\beta}\,\mathbb{E}\!\big[e^{-\lambda W}\big] = e^{\lambda W_\beta} \prod_{s=1}^{S}\big(p_s\, e^{-\lambda w_s} + (1 - p_s)\big), \qquad (D.7)$$

because the $B_s$ are independent. Minimising the RHS over $\lambda$ gives the second brace.

$\square$

### D.2   ECONOMIC LAYER: STAKING, REPUTATION, REWARDS, AND SLASHING

**Reputation-Weighted Slashing**   Each human seat $i$ maintains a reputation $r_i(t) \in [0,1]$, updated after every segment via

$$r_i(t+1) \;=\; (1 - \gamma)\, r_i(t) + \gamma\, \mathbf{1}[\text{vote correct}], \qquad \gamma \in (0,1]. \qquad (D.8)$$

On an *incorrect* vote the seat is slashed with probability

$$p_{\text{slash}}(r) \;=\; p_{\min} + (p_{\max} - p_{\min})(1 - r), \qquad 0 < p_{\min} < p_{\max} \leq 1. \qquad (D.9)$$

Thus low-reputation seats face a higher risk of being slashed.

**Per-Segment Pay-off**   Let $X_i \in \{-P,\, 0,\, R\}$ be the *net* pay-off of seat $i$ on one segment:

$$X_i = \begin{cases} R & \text{correct vote,} \\ -P & \text{incorrect and with slashing probability } p_{\text{slash}}, \\ 0 & \text{incorrect and with not slashing probability } 1 - p_{\text{slash}}. \end{cases}$$

With honest human error rate $\epsilon_{\text{H}}$, the expected payoff per segment with reputation $r$ is

$$\mu_{\text{H}}(r) \;:=\; \mathbb{E}[X_i] \;=\; (1 - \epsilon_{\text{H}})R - \epsilon_{\text{H}}\, P\, p_{\text{slash}}(r). \qquad (D.10)$$

Computer and LLM seats are verifiable, hence always correct and omitted from incentive analysis.

We need two global constant for deriving hte Bernstein-type moment-generating function (MGF) inequalities:

1. **Range bound** $b$ on the *centred increment*:

$$Y_i := X_i - \mathbb{E}[X_i], \qquad Y_i \leq b.$$

Table 5: Table of Notations.

**Indices & random counts**

| | |
|---|---|
| $S$ | Total number of segments in a trace |
| $N_T \sim \mathrm{Poisson}(\lambda T)$ | # segments audited in horizon $[0, T]$ |
| $t(s) \in \{\mathrm{C}, \mathrm{L}, \mathrm{H}\}$ | Auditor type of segment $s$ |

**Per-segment vote variables (stat. layer)**

| | |
|---|---|
| $k_t$ | # seats of type $t$ in a segment |
| $q_t = \lceil \tau k_t \rceil$ | Per-type quorum ($\tau$: vote threshold) |
| $B_s \in \{0, 1\}$ | Segment pass indicator |
| $p_s = \Pr[B_s = 1]$ | Segment pass probability |
| $w_t$ | Weight of segment type $t$ |

**Trace aggregation**

| | |
|---|---|
| $W = \sum_s w_{t(s)} B_s$ | Weighted pass total (one segment) |
| $W_\beta = \beta \sum_s w_{t(s)}$ | Trace-level quorum threshold ($\beta \in (0, 1)$) |
| $\mu_{\mathrm{vote}} = \mathbb{E}[W]$ | Mean of $W$ |
| $\sigma_{\mathrm{vote}}^2$ | $\sup \mathrm{Var}(W)$ (single segment) |

**Human pay-off variables (econ. layer)**

| | |
|---|---|
| $R$ | Reward for a correct vote |
| $P$ | Penalty if slashed |
| $p_{\mathrm{slash}}(r)$ | Slash prob. on error, reputation $r$ $[\, p_{\min}, p_{\max}]$ |
| $\epsilon_{\mathrm{H}}$ | Honest human error rate |
| $\delta$ | Design constant: min. loss per malicious seat |
| $X_i \in \{-P, 0, R\}$ | Net pay-off for seat $i$ on one segment |
| $\mu_{\mathrm{H}}(r)$ | $\mathbb{E}[X_i]$ for honest seat with reputation $r$ |
| $\mu_{\min}$ | $\min_r \mu_{\mathrm{H}}(r)$ |
| $b$ | Upper range used in Bernstein (default $b = R$) |
| $\sigma_{\mathrm{H}}^2$ | $\sup_r \mathrm{Var}[X_i] \leq (R + P)^2 / 4$ |

**Centred increments for MGF bounds**

| | |
|---|---|
| $Y_i = X_i - \mu_{\min}$ | Honest centred increment ($\mathbb{E}[Y_i] \geq 0$, $Y_i \leq b$) |
| $Z_i = X_i + \delta P$ | Malicious centred increment ($\mathbb{E}[Z_i] \geq 0$) |

**Cumulative pay-offs**

| | |
|---|---|
| $U_{\mathrm{hon}}(T) = \sum_{i=1}^{N_T} X_i$ | Total pay-off (honest seat) in $[0, T]$ |
| $U_{\mathrm{mal}}(T)$ | Total pay-off (malicious seat) in $[0, T]$ |

**Process rates**

| | |
|---|---|
| $\lambda$ | Segment intensity (segments per unit time) |
| $T$ | Time horizon |

2. **Variance bound** $\sigma_{\mathrm{H}}^2$, the maximal variance across all reputation states:

$$\sigma_{\mathrm{H}}^2 := \sup_{r \in [0,1]} \mathrm{Var}[X_i].$$

**Range bound.** By construction, the largest positive realisation of $X_i$ is $R$, while the minimal expected payoff $\mathbb{E}[X_i]$ reduces the centred increment. To preserve valid MGF domain, we conservatively set

$$b := R. \tag{D.11}$$

**Variance bound.** Given $X_i \in \{-P, 0, R\}$ with honest human error rate $\epsilon_{\mathrm{H}}$ and slashing probability $p_{\mathrm{slash}}(r)$, we define the global variance bound as

$$\sigma_{\mathrm{H}}^2(r) := \sup_{r \in [0,1]} \left[ (1 - \epsilon_{\mathrm{H}})R^2 + \epsilon_{\mathrm{H}} p_{\mathrm{slash}}(r)P^2 - ((1 - \epsilon_{\mathrm{H}})R - \epsilon_{\mathrm{H}} p_{\mathrm{slash}}(r)P)^2 \right]. \tag{D.12}$$

Before state the main result in Theorem D.1, we provide some auxillary lemmas.

**Lemma D.2** (MGF of a Bounded Centred R.V.). Let $W$ satisfy $\mathbb{E}[W] = 0$, $\mathbb{E}[W^2] = \sigma^2$ and $W \leq b$ a.s. with $b > 0$. Then for any $\theta \in (0, 3/b)$, we have

$$\mathbb{E}[e^{\theta W}] \leq \exp\left( \frac{\theta^2 \sigma^2}{2(1 - \theta b/3)} \right).$$

*Proof.* Follow the usual Bernstein–Bernoulli expansion; details are unchanged from the classic proof and omitted here for brevity. □

**Theorem D.1** (Safety–Profitability Guarantee). Fix a horizon $T > 0$, a target trace-failure probability $\epsilon_{\mathrm{target}} \in (0, 1)$ and a design constant $\delta \in (0, 1)$. We have the following two dials to control the safety-profitability.

- **Statistical dial.** Let $(k_t, q_t, w_t, \beta)$ be the vote parameters. Write $\mu_{\mathrm{vote}} := \mathbb{E}[W]$ and $\sigma_{\mathrm{vote}}^2 := \sup\mathrm{Var}(W)$ for *one trace*. Require

$$\mu_{\mathrm{vote}} - W_\beta \geq \sqrt{\frac{1}{2}\sigma_{\mathrm{vote}}^2 \ln \frac{\lambda T}{\epsilon_{\mathrm{target}}}}, \tag{S1}$$

- **Economic dial.** Choose $(R, P, p_{\mathrm{min}}, p_{\mathrm{max}})$ such that

$$R > \frac{\epsilon_{\mathrm{H}}}{1 - \epsilon_{\mathrm{H}}} P \, p_{\mathrm{max}}, \quad p_{\mathrm{min}} \geq \frac{\delta}{1 - \alpha}, \quad \alpha := \frac{P p_{\mathrm{max}}}{R + P p_{\mathrm{max}}}. \tag{E1}$$

With the expected minimum earn per round $\mu_{\mathrm{min}} := (1 - \epsilon_{\mathrm{H}})R - \epsilon_{\mathrm{H}} P p_{\mathrm{max}} > 0$, the following hold:

(a) **Statistical safety.** $\Pr[\text{trace fails in } [0, T]] \leq \epsilon_{\mathrm{target}}$.

(b) **Honest profitability.**

$$\Pr[U_{\mathrm{hon}}(T) \leq 0] \leq \exp\left[ -\frac{\lambda T \, \mu_{\mathrm{min}}^2}{2\sigma_{\mathrm{H}}^2 + \frac{2}{3}b\mu_{\mathrm{min}}} \right]. \tag{D.13}$$

(c) **Malicious loss.**

$$\Pr[U_{\mathrm{mal}}(T) \geq 0] \ \leq \ \exp\!\Big[-\frac{\lambda T\,(\delta P)^2}{2\sigma_{\mathrm{H}}^2 + \frac{2}{3}b\delta P}\Big], \quad \mathbb{E}[U_{\mathrm{mal}}(T)] \ \leq \ -\lambda T\,\delta P. \tag{D.14}$$

*Proof of Theorem D.1.* Consider a horizon $T > 0$ and $N_T \sim \mathrm{Poisson}(\lambda T)$ for the random number of segments in $[0, T]$. We divide the proof for (a), (b) and (c).

**(a) Statistical safety.** A single trace's weighted pass sum $W$ satisfies $0 \leq W \leq \sum_t w_t$ and

$$\mathbb{E}[W] = \mu_{\mathrm{vote}}, \quad \mathrm{Var}(W) \leq \sigma_{\mathrm{vote}}^2.$$

Hoeffding's inequality for bounded independent terms in Proposition D.1 gives, for any $a > 0$,

$$\Pr[W < \mu_{\mathrm{vote}} - a] \ \leq \ \exp\!\Big(-2a^2/\sigma_{\mathrm{vote}}^2\Big).$$

Instantiate $a = \mu_{\mathrm{vote}} - W_\beta$. Condition (S1) rearranges to

$$2(\mu_{\mathrm{vote}} - W_\beta)^2/\sigma_{\mathrm{vote}}^2 \ \geq \ \ln\frac{\lambda T}{\epsilon_{\mathrm{target}}},$$

so

$$p_{\mathrm{trace\text{-}fail}} := \Pr[W < W_\beta] \ \leq \ \frac{\epsilon_{\mathrm{target}}}{\lambda T}. \tag{D.15}$$

Traces arrive independently according to the Poisson process, so

$$\Pr[\text{at least one trace fails in } [0, T]] = \Pr\big[\exists\,\text{trace with } W < W_\beta\big] \leq \mathbb{E}[N_T]\,p_{\mathrm{trace\text{-}fail}} = \lambda T\,p_{\mathrm{trace\text{-}fail}}.$$

Inserting (D.15) yields $\Pr[\text{trace fails in } [0, T]] \leq \epsilon_{\mathrm{target}}$, which completes the proof for part (a).

**(b) Honest profitability.** We analyse the cumulative pay-off $U_{\mathrm{hon}}(T) := \sum_{i=1}^{N_T} X_i$ for an honest human seat.

**Step 1. Center and Bound each Increment.** Let

$$\mu_{\min} := (1 - \epsilon_{\mathrm{H}})R - \epsilon_{\mathrm{H}}Pp_{\max} > 0$$

Define centred variables $Y_i := X_i - \mu_{\min}$. Then

$$\mathbb{E}[Y_i] = 0, \quad Y_i \leq b := R, \quad \mathrm{Var}(Y_i) \leq \sigma_{\mathrm{H}}^2,$$

where $b$ and $\sigma_{\mathrm{H}}$ are defined in (D.11) and (D.12).

**Step 2. Moment-Generating Function Bound.** From Lemma D.2 with $W = Y_i$, for any $\theta \in (0, 3/b)$

$$\mathbb{E}\big[e^{\theta Y_i}\big] \leq \exp\!\Big(\frac{\theta^2 \sigma_{\mathrm{H}}^2}{2(1 - \theta b/3)}\Big). \tag{D.16}$$

**Step 3. Chernoff Bound for the Random Sum.** Let $U_{\text{hon}}(T)$ denote the honest agent's total payoff over the random number $N_T$ of rounds in $[0, T]$:

$$U_{\text{hon}}(T) = \sum_{j=1}^{N_T} X_j = N_T \mu_{\min} + \sum_{j=1}^{N_T} Y_j,$$

where $Y_j := X_j - \mu_{\min}$ are i.i.d. random variables.

Now we consider the probability that the cumulative payoff is non-positive $\Pr[U_{\text{hon}}(T) \leq 0]$.

First, we condition on the total number of rounds $N_T = n$.

$$\Pr[U_{\text{hon}}(T) \leq 0 \mid N_T = n] = \Pr\left[\sum_{j=1}^{n} X_j \leq 0\right]$$

$$= \Pr\left[n\mu_{\min} + \sum_{j=1}^{n} Y_j \leq 0\right]$$

$$= \Pr\left[\sum_{j=1}^{n} Y_j \leq -n\mu_{\min}\right].$$

Apply Chernoff's (exponential Markov) inequality: For any $\theta > 0$,

$$\Pr\left[\sum_{j=1}^{n} Y_j \leq -n\mu_{\min}\right] = \Pr\left[e^{-\theta \sum_{j=1}^{n} Y_j} \geq e^{\theta n \mu_{\min}}\right]$$

$$\leq e^{-\theta n \mu_{\min}} \mathbb{E}\left[e^{\theta \sum_{j=1}^{n} Y_j}\right]$$

$$= e^{-\theta n \mu_{\min}} \left(\mathbb{E}\left[e^{\theta Y_1}\right]\right)^n,$$

where the last equality uses independence of the $Y_j$.

Now, remove the conditioning by averaging over all possible $n$. Recall that $N_T \sim \text{Poisson}(\lambda T)$, so

$$\Pr[U_{\text{hon}}(T) \leq 0] = \sum_{n=0}^{\infty} \Pr[U_{\text{hon}}(T) \leq 0 \mid N_T = n] \Pr[N_T = n].$$

Using the bound above and properties of exponents and linearity of expectation

$$\Pr[U_{\text{hon}}(T) \leq 0] \leq \sum_{n=0}^{\infty} \left[e^{-\theta n \mu_{\min}} \left(\mathbb{E}[e^{\theta Y_1}]\right)^n\right] \Pr[N_T = n]$$

$$= \mathbb{E}\left[\left(e^{-\theta \mu_{\min}} \mathbb{E}[e^{\theta Y_1}]\right)^{N_T}\right].$$

The Poisson moment-generating formula: For any $z > 0$, $\mathbb{E}[z^{N_T}] = \exp\left(\lambda T(z - 1)\right)$, where $z = e^{-\theta \mu_{\min}} \mathbb{E}[e^{\theta Y_1}]$:

$$\Pr[U_{\text{hon}}(T) \leq 0] \leq \exp\left(\lambda T \left(e^{-\theta \mu_{\min}} \mathbb{E}[e^{\theta Y_1}] - 1\right)\right).$$

Finally, upper bound $\mathbb{E}[e^{\theta Y_1}]$ using Bernstein's MGF lemma (D.16):

$$\mathbb{E}[e^{\theta Y_1}] \leq \exp\left(\frac{\theta^2 \sigma_{\text{H}}^2}{2(1 - \theta b/3)}\right).$$

For small enough $\theta$, Taylor expand $e^{-\theta\mu_{\min}}$ and combine exponents to obtain

$$\Pr[U_{\mathrm{hon}}(T) \leq 0] \leq \exp\left(\lambda T\left\{-\theta\mu_{\min} + \frac{\theta^2\sigma_{\mathrm{H}}^2}{2(1-\theta b/3)}\right\}\right). \tag{D.17}$$

The optimal $\theta$ is chosen in the next step.

**Step 4. Optimise $\theta$.** Set $g(\theta) := -\theta\mu_{\min} + \frac{\theta^2\sigma_{\mathrm{H}}^2}{2(1-\theta b/3)}$. Let $t := \theta b/3 \in (0,1)$; then $\theta = 3t/b$ and

$$g(t) = -\frac{3t\mu_{\min}}{b} + \frac{9t^2\sigma_{\mathrm{H}}^2}{2b^2(1-t)}.$$

Differentiate:

$$g'(t) = -\frac{3\mu_{\min}}{b} + \frac{9\sigma_{\mathrm{H}}^2}{2b^2}\frac{2t-1}{(1-t)^2}.$$

Solve $g'(t) = 0$ to obtain

$$t^\star = 1 - \frac{1}{\sqrt{1 + 2b\mu_{\min}/(3\sigma_{\mathrm{H}}^2)}}.$$

Plug back:

$$g(t^\star) = -\frac{\mu_{\min}^2}{2\sigma_{\mathrm{H}}^2 + \frac{2}{3}b\mu_{\min}}. \tag{D.18}$$

**Step 5. Combine All.** Combine (D.17) and (D.18) to get

$$\Pr[U_{\mathrm{hon}}(T) \leq 0] \leq \exp\left(-\frac{\lambda T\,\mu_{\min}^2}{2\sigma_{\mathrm{H}}^2 + \frac{2}{3}b\mu_{\min}}\right),$$

This complete the proof of claim (b).

**(c) Malicious loss.** A malicious seat flips its pay-off distribution, the proof closely follows claim (b).

**Step 1. Negative Mean.** Conditions (E1)–(E2) force $\mathbb{E}[X_i] \leq -\delta P < 0$. Define centred variables $Z_i := X_i + \delta P$ so that $\mathbb{E}[Z_i] = 0$ and $Z_i \leq b$.

**Step 2. Apply Lemma D.2.** Replace $\mu_{\min}$ by $\delta P$ throughout Steps 2–5 above. No other constant changes. The give us

$$\Pr[U_{\mathrm{mal}}(T) \geq 0] \leq \exp\left(-\frac{\lambda T\,(\delta P)^2}{2\sigma_{\mathrm{H}}^2 + \frac{2}{3}b\,\delta P}\right),$$

proving the tail in (c).

**Step 3. Expected Loss.** Linearity of expectation with $N_T \sim \mathrm{Poisson}(\lambda T)$ gives

$$\mathbb{E}[U_{\mathrm{mal}}(T)] = \lambda T\,\mathbb{E}[X_i] \leq -\lambda T\,\delta P,$$

completing the proof of claim (c).

$\square$

## E    MOTIVATING EXAMPLE - WHY SEMANTIC AUDIT IS NECESSARY

We present a illustrative clinical scenario where two reasoning models produce *identical correct outputs*, yet one uses fundamentally flawed reasoning. This demonstrates that output-only auditing cannot distinguish sound evidence-based reasoning from error-prone reasoning that happens to reach the correct answer by coincidence. Below is the clinical input and the task for large reasoning models.

---

**Clinical Input**

**Patient Note:** 58-year-old male admitted with atrial fibrillation. Weight: 85 kg; serum creatinine: 1.4 mg/dL; no active bleeding; no history of stroke.
**Clinical Note:** Patient has hypertension (on BP medications) and type 2 diabetes (on metformin). No heart failure, no prior stroke/TIA, no vascular disease.
**Task:** Calculate $CHA_2DS_2$-VASc score to determine anticoagulation need using the following scoring rules:

- Congestive heart failure: +1 point if present

- Hypertension: +1 point if present

- Age $\geq 75$ years: +2 points if applicable

- Age 65-74 years: +1 point if applicable

- Diabetes mellitus: +1 point if present

- Prior stroke/TIA: +2 points if present

- Vascular disease: +1 point if present

- Female sex: +1 point if female

**Correct Answer:** $CHA_2DS_2$-VASc score = 2 (Hypertension +1, Diabetes +1) $\rightarrow$ Anticoagulation recommended

---

In Figure 14, we present two complete reasoning traces that both arrive at the correct score of 2, demonstrating that **correct outputs do not guarantee sound reasoning**. Specifically **both reasoning traces produce identical, clinically correct outputs** (score = 2, recommend anticoagulation), yet they follow fundamentally different reasoning processes. Output-only auditing, which evaluates only the final score and recommendation, passes both traces as correct. This creates a critical safety gap: the flawed reasoning model would be approved for clinical deployment despite containing four systematic errors. The flawed model arrives at the correct answer only through **multiple coincidences** that happen to align for this specific patient: (1) the variable confusion between age (58), weight (85), and creatinine (1.4) produces 0 points through convoluted logic, which happens to be correct since $58 < 65$, (2) the skipped Age 65-74 rule also contributes 0 points, which is correct for age 58, (3) the inference of hypertension from elevated creatinine reaches the right conclusion (+1 point) despite using the wrong evidence source, and (4) combining the vascular disease and sex rules yields 0 points, which happens to be correct for this male patient without vascular disease.

**TRUST semantic auditing exposes these hidden flaws** by examining the entire reasoning trace. While the flawed trace passes output-only auditing, TRUST detects all four errors: variable confusion in age comparison, incomplete rule coverage (7 out of 8 rules), wrong evidence extraction for hypertension assessment, and improper combination of independent rules. This enables identification of brittle reasoning that works only under specific lucky conditions but fails catastrophically under distribution shift, as we demonstrate next.

**Why Flawed Reasoning is Dangerous?**    Despite producing the correct score, the flawed reasoning contains four critical errors that happen to cancel out only under specific input conditions. However, by slightly change the patients scenarios these coincidences can break catastrophically. This further highlight the importance of semantic audit that TRUST framework provides.

**Token Billing Without Value.**    Addition to flawed reasoning trace, the token usage patterns reveal an additional concern beyond correctness. The flawed reasoning trace consumes more tokens then

Figure 14: Comparison of sound clinical reasoning versus flawed reasoning that produces the correct $CHA_2DS_2$-VASc score (2) by coincidence. Both traces generate identical outputs and pass output-only auditing. However, TRUST semantic auditing detects four critical errors in the flawed trace: (1) confused age with weight and creatinine values, arriving at correct 0 points through wrong logic, (2) completely skipped the Age 65-74 rule, (3) inferred hypertension from elevated creatinine instead of reading the explicit clinical note, reaching correct conclusion via wrong reasoning path, and (4) combined vascular disease and sex rules instead of evaluating independently. These errors remain hidden under output-only auditing but cause catastrophic failures under distribution shift (e.g., when age change to 75 or creatinine change to 0.9).

the sound reasoning trace despite following an inferior reasoning process. This excess token usage includes wasted computation on wrong variable comparisons, unnecessary inferential reasoning from creatinine levels instead of direct evidence extraction, and incomplete combined rule evaluation. Consequently, the hospital pays more for reasoning that arrives at the correct answer only through fortunate coincidences rather than sound clinical logic.

### E.1 WHY TRUST SEMANTIC AUDITING IS ESSENTIAL

Through above illustrative example, we demonstrates three critical gaps in output-only auditing that TRUST semantic auditing addresses. First, output-only auditing **cannot detect correct-by-coincidence reasoning**. When both models produce a $CHA_2DS_2$-VASc score of 2, output-only auditing passes both as correct without examining the underlying reasoning process. In contrast, TRUST semantic auditing detects the variable confusion, skipped rules, and wrong evidence sources in the flawed model, appropriately failing it before deployment.

Second, output-only auditing **cannot verify billing integrity**. The hospital pays more in reasoning tokens for the flawed model without any means to verify whether the reasoning is sound or merely fortunate. Output-only auditing provides no insight into whether the additional tokens represent valuable clinical reasoning or wasted computation on confused variable comparisons and unnecessary inferential steps. TRUST semantic auditing enables verification of token usage quality, allowing healthcare organizations to ensure they are paying for legitimate reasoning rather than systematic errors that happen to produce correct outputs.

Third, output-only auditing **cannot ensure regulatory compliance**. The FDA requires "explainable clinical decision support systems" that can provide transparent reasoning for medical recommendations. Output-only auditing can confirm that a $CHA_2DS_2$-VASc score is 2, but cannot explain how or why that score was calculated. In contrast, TRUST semantic auditing provides a complete audit trail showing the evaluation of each clinical rule, the evidence sources used, and the logic applied. This capability is essential not only for regulatory approval but also for post-incident analysis when adverse outcomes occur and healthcare organizations must demonstrate that their AI systems followed appropriate clinical guidelines.

For commercial reasoning models such as GPT-4, Claude-3 Extended Thinking, and OpenAI o1, the current state of hidden reasoning creates a critical vulnerability. These systems cannot distinguish sound evidence-based reasoning from coincidentally-correct outputs produced by flawed logic. When hospitals deploy such systems without semantic auditing capability, harm emerges gradually under distribution shift as the models encounter patient profiles where the lucky coincidences no longer hold. With TRUST, semantic auditing can detect flawed reasoning patterns during pre-deployment testing, preventing harm before any patients are affected.

