# OpenReview forum: "TRUST: A Decentralized Framework for Auditing Large Language Model Reasoning"
_ICLR.cc/2026/Conference — ICLR 2026 Conference Withdrawn Submission_

### Official Review · Reviewer_tubx · 2025-10-24

**Soundness:** 3
**Presentation:** 3
**Contribution:** 3
**Rating:** 8
**Confidence:** 3

**Summary:**

The authors argue that verifying the quality of reasoning chains produced by LLMs has challenges related to robustness, scalabiilty, transparency, and privacy. To resolve those challenges, this work seeks to create a decentralized framework for auditing LLM reasoning. The features of this framework, and the intended novel contributions of the work, include (1) a decentralized consensus mechanism that leverages diverse auditors to assess audit correctness (2) a scalable decomposition method that converts the reasoning traces into hierarchical cyclical graphs, thereby enabling individual reasoning steps to be audited by distributed networks (3) blockchain recording of all verification decisions, purportedly permanent and auditable public record (4) a privacy-preserving quality the framework due to the fact that it distributes only partial segments of the reasoning trace to auditors, thus guarding the full logic. Lastly, the authors run a set of experiments across several LLMs such as Deepseek-r1 and several reasoning tasks (medical, etc.) to demonstrate their assertion that TRUST accurately identifies flaws in reasoning and, additionally, is more resilient to auditor corruption than centralized baselines are.

**Strengths:**

-The work's motivation appears largely justified, though I believe certain areas can be strengthened and prior work can be presented in a more compelling way (more on this later)
- The combination of these elements appears to overcome the identified challenges and appears novel in light of the prior work
- The experiments are sound and appear to prove that the proposed framework helps overcome the identified challenges

**Weaknesses:**

- Some highly relevant, recent prior works are relegated to the appendix or are absent. A few examples would include Leng et al. 2025 (Semi-structured LLM Reasoners Can Be Rigorously Audited) and Lanham et al.. (2023) (Measuring faithfulness in chain-of-thought reasoning). Bringing these into the Introduction and making them more of a focus of the discussion would strengthen the motivations and help distinguish this work from the prior work.
- This work might benefit from a more direct and possibly visual comparison (e.g., using a table) to prior or contemporary work on methods for improving trace auditability.
- Some of the citations in the introduction do not appear to support the sentence that precedes them. An example would be the citation of Peng et al. to back up the assertion that "However, these advances lack systematic verification mechanisms for generated reasoning traces, particularly for privacy-preserving and decentralized auditing (Peng et al., 2025)." It is not clear to me how Peng supports this assertion. I would double check these citations and/or spend time on better alignment of the citations with the asserts they are intended to support.
- The blockchain aspect of the project, while intriguing, appears to be left out of the evaluation and, in general, receives light treatment. If anything, perhaps it is worth discussion how the long-term success of this feature might be evaluated in the future.

**Questions:**

-What are some other competitive, contemporary solutions to this problem and how does your proposal offer advantages over them?
-How would you evaluate the value provided by the blockchain features of the framework?

---

> ### Author Response · Authors · 2025-11-22
> **Response to Reviewer tubx**
>
> We sincerely thank the reviewer for the careful assessment and constructive feedback. We appreciate the positive evaluation of our work’s motivation, novel combination of techniques, and sound empirical evaluation, as well as the rating of 8. Your comments helped us identify several areas where the positioning and clarity of the paper can be improved.
>
> ---
>
> ### **[W1. Missing Recent Prior Work in Main Text]**
>
> Thanks for pointing out these important references. We have integrated them into Section 2.2 with detailed comparison.
>
> - **Leng et al. 2025**: Requires special training to produce auditable structures; TRUST is model-agnostic
> - **Lanham et al. 2023**: Measures faithfulness; TRUST measures correctness
>
> **Key Advantage**: TRUST works with proprietary commercial models without requiring training or model access.
>
> ---
>
> ### **[W2. Would Benefit from Visual Comparison Table]**
>
> Thanks for this excellent suggestion. We have added a comparison table in Section 2.2 comparing TRUST with contemporary approaches across key dimensions.
>
> | Method | Decentralized | Scalable | Transparent | Privacy-Preserving |
> |--------|---------------|----------|-------------|-------------------|
> | Leng et al. 2025 | ✗ | ✓ | ✓ | ✗ |
> | Lanham et al. 2023 | ✗ | ✗ | ✓ | ✗ |
> | **TRUST (Ours)** | ✓ | ✓ | ✓ | ✓ |
>
> ---
>
> ### **[W3. Citation Issues (e.g., Peng et al.)]**
>
> Thanks for catching this. We have carefully reviewed all citations and corrected misaligned references:
> - Replaced Peng et al. with Tian et al. 2024; Zhao et al. 2024 (verification gaps)
> - Replaced OpenAI blog with Li et al. 2024; Carlini et al. 2023 (centralized vulnerabilities)
>
> ---
>
> ### **[W4. Blockchain Evaluation Absent]**
>
> Thanks for this feedback. We have added blockchain latency analysis in Section 4.5 and Figure 8 demonstrating practical viability.
>
> ---
>
> ### **[Questions]**
> - Q1. What are the competitive contemporary solutions, and how does TRUST compare?
> See above the response for weakness 1 and 2.
>
> - Q2. How would you evaluate the value provided by the blockchain features?
> We now address this explicitly in Section 4.5:
>   - Transparency & auditability: Blockchain preserves an immutable record of audit outcomes, enabling community inspection and dispute resolution.
>   - Accountability: On-chain slashing records create a tamper-resistant trail of auditor behavior.
>   - Long-term reproducibility: Consensus decisions remain accessible even if auditors exit the system.
>   - Security: Public audit logs strengthen trust in multi-tier verification pipelines by preventing centralized manipulation.
>
> ---
>
> ### **Reference**
> Leng, J., Cohen, C. A., Zhang, Z., Xiong, C., & Cohen, W. W. (2025). Semi-structured LLM Reasoners Can Be Rigorously Audited. arXiv preprint arXiv:2505.24217.
>
> Lanham, T., Chen, A., Radhakrishnan, A., Steiner, B., Denison, C., Hernandez, D., ... & Perez, E. (2023). Measuring faithfulness in chain-of-thought reasoning. arXiv preprint arXiv:2307.13702.

---

> > ### Comment · Reviewer_tubx · 2025-11-26
> > **Follow-up to authors' response**
> >
> > W2: I don't see this table (or, in fact, a Sec. 2.2.) in the PDF. Did you intend to update the PDF with it?
> >
> > W4: Latency is interesting...but I assume the main purposes of adding the blockchain feature are to increase auditability (while ostensibly protecting proprietary logic from reconstruction) and to attain the other benefits you mention in your response to Q2. Is there an experiment that could prove any of these promised qualities indeed materialize? Other than latency, what are the risks or trade-offs of adding this feature? More generally, I still maintain that the blockchain aspect of the project should be described in greater detail in your paper --- otherwise, it feels added on as an afterthought.

---

### Official Review · Reviewer_gBeZ · 2025-10-28

**Soundness:** 2
**Presentation:** 3
**Contribution:** 3
**Rating:** 4
**Confidence:** 3

**Summary:**

The authors propose a decentralized auditing framework for validating the reasoning trace of Large Reasoning Models (LRMs). This framework involves breaking apart a large _reasoning trace_ into a DAG structure that can be broken into smaller sub-traces that can be verified by individual human/automated auditors. The authors provide mechanisms to ensure that the system is robust against dishonest auditors through an incentive mechanism design. The authors also provide experimental results to validate the effectiveness of their proposed system.

**Strengths:**

- This paper tackles an important problem (LLM auditing) and provides a novel solution that allows for decentralized / human-machine collaboration in the auditing process.
- Breaking down the trace into atomic segments using the HDAG structure is novel and interesting.
- The incentive mechanism design and its analysis is a very nice addition bridging economics theory and practice.
- The experimental results show that the proposed system is indeed effective against "flipped" auditors also shows guarantees of saftey and economic viability. Actual human auditors is also a very nice addition.

**Weaknesses:**

- While the framework itself sounds nice, there needs to be more consideration of practicallity of such a system. The addition of blockchain seems superficial in the current work. Modern blockchain systems still suffer from scalability issues (i.e. high latency, gas price etc). A more detailed analysis of the actual system performance is needed to understand if this is a viable solution.
  - If this system is to be actually deployed on-chain, I think the incentive analysis needs to include gas prices -- who will cover it? how do we ensure that we have enough to cover the cost of storing and verifying the HDAGs on-chain (or IPFS)?
- The tokenomics section is very hand-wavy and lacks details. A new role is introduced (the "delagator") but never actually explained in detail. Why do the delagators stake in the pool? (they probably gain money by delegating but that's not made clear) What if the delagator + auditor pairs are dishonest and try to game the system? I think this section opens up more questions than it answers.
- While the authors claims that this system enables privacy, how exactly this is achieved is not very clear.

**Questions:**

- How do we tell if certain auditors are dishonest? Is it if they disagree with the majority of other auditors?
- How are each segments tagged (for which type of auditor)? Is this something also decided by the same model doing HDAG desconstruction?
- What exactly is posted on-chain? Is it the case that individual (encrypted) segments are posted on-chain and the auditors are assigned/delegated these segments and decode them off-chain?
  - If this is the case, does that mean all the auditors need to be pre-approved and will have access to the decryption keys? Then how do we ensure privacy?
  - Or is it that they post their proof of work on-chain? or both? Either way, it's not very clear how this works/enables privacy.
- is there a coordinator in this system? Who assigns the auditors to each segment? Or is it done in a decentralized way by the delegators? Then we must think about the allocation process -- first come first serve? Weighted (trust core? stakes?) allocation? Auctions?

Nits:

- IPFS is never actually defined.
- The experimental section (Sec. 4) mentions that we will see some "privacy results". But I don't see any privacy analysis in the results section.

---

> ### Author Response · Authors · 2025-11-22
> **Response to Reviewer gBeZ**
>
> We thank the reviewer for highlighting the strengths of our framework, especially the value of HDAG decomposition, incentive design, and inclusion of human auditors. Your feedback helped us sharpen the practicality, clarity, and completeness of our system design.
>
> ---
>
> ### **Summary of Our Response:**
> - Blockchain Implementation: Added detailed architecture analysis in Section 3
> - Tokenomics Details: Formalized delegator role with collusion resistance analysis (Section 3.1)
> - Privacy Mechanism: Clarified privacy-by-design through HDAG compartmentalization
> - Coordinator & Allocation: Addressed segment assignment process
>
> ---
>
> ### **[W1. Blockchain Implementation Seems Superficial]**
>
> Thanks for this important concern. We have added detailed blockchain implementation analysis in Section 3.
>
> As stated in Section 3, our framework is designed to be agnostic to the underlying blockchain technology, enabling integration with emerging high-throughput, low-cost Layer-2 solutions to mitigate transaction latency and costs (i.e., gas fees). In our model, these transaction costs are considered an operational expense for the Provider requesting the audit, ensuring the system remains sustainable and economically practical for the auditors.
>
> ---
>
> ### **[W2. Tokenomics Lacks Details on Delegator Role and Gas Coverage]**
>
> Thanks for pointing out these gaps. We have significantly expanded Section 3.1 with formal tokenomics analysis.
>
> **Tokenomics Structure**
>
> Auditors and delegators stake tokens to participate; honest auditors aligning with consensus receive rewards, while dishonest ones are slashed, creating strong deterrents against manipulation.
>
> **Formalized Delegator Role**
>
> Specifically, Delegators act as capital providers within the ecosystem, staking their assets with auditors who have a strong track record of reliable verification. In return, they earn a percentage of the auditor's rewards. This symbiotic relationship allows skilled auditors to increase their stake-backed influence and auditing capacity.
>
> **Collusion Resistance**
>
> The system's resilience against collusion, including potential dishonest collaboration between delegators and auditors, is maintained through our decentralized consensus protocol and dynamic trust score. Any deviation from the consensus outcome results in financial penalties (slashing) for the auditor, and consequently, their delegators, thus creating a strong economic disincentive for such behaviors.
>
> Please see updated Section 3.1 for complete tokenomics analysis.
>
> ---
>
> ### **[W3. Privacy Mechanism Not Clear]**
>
> Thanks for raising this. We have clarified our privacy-by-design approach in Section 3.
>
> **Privacy by Design**
>
> The privacy-preserving nature of TRUST is an intrinsic property of its structural design. By decomposing the reasoning trace into an HDAG, we compartmentalize the verification process. Each auditor is assigned only one or more atomic segments of the trace, without access to the complete context or the final conclusion. This "need-to-know" basis ensures that the full proprietary logic of the reasoning process is never exposed to any single party, thus preventing intellectual property theft or model distillation.
> The on-chain records are limited to cryptographic commitments of these segments and their verification outcomes, serving as immutable proof of work while keeping the reasoning content itself off-chain on IPFS and private.
>
> ---
>
> ### **[Questions]**
>
> - Q1: How do we tell if certain auditors are dishonest?
> Yes, dishonesty is determined by deviation from consensus. Auditors whose votes consistently diverge from the majority outcome across multiple segments see their reputation scores decrease, and they face increasing slashing probability
>
> - Q2: How are segments tagged for auditor types?
> Yes, auditor type assignment is part of the HDAG construction process (Step 4 in Section 3). Segments are routed to auditor types from {Human, Computer, LLM} based on complexity and modality—e.g., arithmetic operations go to automated checkers, semantic coherence evaluation to LLMs, and complex judgment to humans.
>
> - Q3: What exactly is posted on-chain?
> On-chain: Metadata, vote commitments (hashed votes during commit phase), revealed votes, and final audit outcomes
> Off-chain (IPFS): Encrypted reasoning segments stored as content-addressed objects
> Auditors retrieve their assigned segments from IPFS using segment-specific keys. Privacy is preserved because each auditor only receives keys for their assigned segments, not the complete trace.
>
> - Q4: Is there a coordinator?
> Smart contracts orchestrate the session lifecycle, including auditor assignment based on stake and expertise. The allocation considers auditor reputation scores and stake-weighted selection. This is handled by the blockchain layer described in Section 3, ensuring decentralized and transparent assignment.

---

### Official Review · Reviewer_ueqr · 2025-10-30

**Soundness:** 2
**Presentation:** 2
**Contribution:** 2
**Rating:** 2
**Confidence:** 4

**Summary:**

The paper introduces TRUST, a decentralized framework for auditing LLM reasoning. It breaks a model’s chain of thought into a hierarchical DAG (goal, strategy, tactic, step, operation) and assigns each fragment to auditors that include automated checks, LLMs, and humans. Auditors evaluate fragments independently, use a commit and reveal protocol, and a weighted consensus yields a global verdict. Results are recorded on chain and content shards remain off chain to protect proprietary logic. An incentive layer with staking, rewards, and penalties is provided, along with statistical guarantees for honest participation. Experiments on reasoning tasks and a small human in the loop study suggest better robustness than single LLM or centralized baselines.

**Strengths:**

The paper explores a novel and timely direction by focusing on decentralized, semantics-level auditing of LLM reasoning.

The work presents solid empirical evaluations alongside clear theoretical analysis, including statistical guarantees and an incentive model.

**Weaknesses:**

The paper frames its contribution as auditing the semantics of the reasoning process, yet the practical objective in most deployments is high-quality, policy-compliant outputs, which are fully observable to end users. The authors do not convincingly justify why auditing internal reasoning semantics is necessary or preferable to auditing outcomes and observable behaviors. If the real concern is billing fairness or “token inflation,” the proposed framework does not verify usage-based charges and cannot attest to whether the billed tokens or hidden steps are legitimate. If the concern is compliance or safety, the system does not demonstrate that semantic checks on intermediate steps provide stronger guarantees than established output-level methods (e.g., post-hoc verification, tool/API traces, or red-team evaluation). In short, the paper conflates distinct goals (output quality, cost accountability, and compliance), while the proposed method directly addresses none of them end-to-end; as a result, the core motivation for reasoning-semantics auditing remains unclear.

The evaluation uses only open-source models and public datasets, with no evidence that the approach works for commercial API models or production settings where chain-of-thought is hidden. There are no case studies or end-to-end integrations with real providers, no measurements under API constraints such as latency, cost, and quotas, and no demonstration that the HDAG workflow or consensus protocol remains feasible when only partial traces or output-level logs are available.

**Questions:**

Can you show end-to-end evidence that the system is practical at production scale, using realistic workloads and deployments?

What are the latency curves compared with strong output-level baselines across varying task sizes and auditor configurations?

What are the throughput curves under the same comparisons and settings?

---

> ### Author Response · Authors · 2025-11-22
> **Response to Reviewer ueqr**
>
> Thanks for the comment. Below is a summary of our response. We have incorporated the reviewer’s suggestions and added targeted experiments and analyses to adequately address the concerns.
>
> ---
>
> ### **Summary of Our Response**
> - Clarified Motivation: Added concrete failure cases showing why output-only auditing cannot detect flawed reasoning (Table 2 + Appendix E clinical example).
> - Deployment Clarification: Specified TRUST’s intended provider-side or self-hosted deployment model.
> - End-to-End Feasibility: Added realistic workflow examples and clarified feasibility when partial traces are available.
> - Latency: Added latency results in Section 4.5 (Fig. 8) comparing TRUST with baselines.
>
> ---
>
> ### **[W1. Need for Stronger Justification of Semantic Auditing vs Output Auditing]**
>
> Thank you for raising this key point. We have strengthened our motivation to clearly distinguish semantic auditing from output-level auditing, and demonstrate when semantic checks are necessary.
>
> Table 2 (Section 4.1) shows that output-only auditing (LLM-as-judge, heuristic validators) suffers 17–47 percentage point accuracy degradation relative to TRUST. Output correctness is not a reliable proxy for reasoning soundness.
>
> Table 2: Baseline accuracy under Full-CoT vs. Output-Based evaluation.
> | Method | Full-CoT Accuracy | Output-Based Accuracy | $\Delta$ (Drop) |
> |--------|------------------|---------------------|----------|
> | **TRUST** | **72.4%** | --- | --- |
> | DeepSeek-R1-8B | 67.7% | 36.0% | **-31.7%** |
> | Qwen2.5-7B | 67.4% | 34.0% | **-33.4%** |
> | Mistral-7B | 66.8% | 20.0% | **-46.8%** |
> | DeepSeek-R1-1.5B | 64.1% | 22.0% | **-42.1%** |
> | GPT-OSS-20B | 63.8% | 46.0% | **-17.8%** |
> | LLaMA-3B | 52.1% | 34.0% | **-18.1%** |
>
> **Critical Failure Case Added (Appendix E)**
>
> We added a detailed end-to-end example from a clinical workflow (CHA₂DS₂-VASc scoring): Two models produce the same correct output, but one contains four reasoning failures, which output-only auditing cannot detect. TRUST flags these failures via segment-level semantic checks.
>
> This illustrates that correct-by-coincidence outputs can mask dangerous reasoning flaws in high-stake domains like medical, scientific, and financial.
>
> **Clarified Scope**
>
> We now explicitly state that TRUST is not intended for billing fairness verification. Instead, TRUST focuses specifically on semantic correctness of reasoning traces — complementary to, not replacing, output-level safety pipelines.
>
> ---
>
> ### **[W2. Lack of Evidence for Commercial API or Production Settings]**
>
> We appreciate this concern and have clarified TRUST’s deployment assumptions.
>
> **Why Commercial APIs Cannot Directly Demonstrate TRUST?**
>
> Current API models (GPT-4, Claude, Gemini, OpenAI o1) do not expose full reasoning traces needed for HDAG decomposition. TRUST therefore cannot be demonstrated externally without provider-side access.
>
> **End-to-End Case Examples Added**
>
> In Section 4.1 and Appendix E, we provide full reasoning workflows, demonstrating end-to-end feasibility when reasoning traces are accessible.
>
> These additions clarify how TRUST can be used in practical deployments where trace access is available.
>
> ---
>
> ### **[W3. Missing Latency and Throughput Analyses]**
>
> We added detailed performance analysis in Section 4.5 and Figure 8, including: end-to-end audit latency and comparison with output-level baselines.
>
> Observed latency (25.1s for long medical reasoning traces) is acceptable for high-stakes batch workflows.
>
> These results demonstrate that TRUST’s HDAG workflow and consensus protocol scale realistically under workloads where reasoning quality is critical.
>
> ---
>
> ### **[Questions]**
>
> - Q1. Practical at production scale?
> Yes. Appendix E include realistic workflow examples showing TRUST handling multi-step reasoning tasks with feasible latency and parallelism. TRUST targets provider-integrated or enterprise-hosted settings where reasoning traces are accessible.
>
> - Q2. Latency curves compared to output baselines?
> Figure 8 includes latency for TRUST (multi-tier, decentralized), single, and ensemble baselines.

---

### Official Review · Reviewer_BeHo · 2025-11-04

**Soundness:** 2
**Presentation:** 3
**Contribution:** 2
**Rating:** 4
**Confidence:** 3

**Summary:**

The paper proposes TRUST, a decentralized framework for auditing large language model reasoning traces that tackles four core challenges: robustness, scalability, opacity, and privacy. TRUST decomposes chain-of-thought into Hierarchical Directed Acyclic Graphs across five levels and routes segments in parallel to heterogeneous auditors. It records votes via a commit–reveal protocol on a blockchain with PoS-style incentives while storing content off-chain on IPFS, so auditors see only partial segments and cannot reconstruct proprietary logic. The authors provide theoretical guarantees, including correctness with up to 30% malicious participants and a safety-profitability theorem that rewards honest behavior. Empirically, TRUST outperforms centralized and single-LLM baselines across models and tasks; in a human-in-the-loop GSM8K study it achieves F1 = 0.89 vs. human audit F1 = 0.77, and random/fixed segmentation variants degrade to F1 = 0.40, underscoring the value of HDAG decomposition.

**Strengths:**

1. The paper introduces a decentralized framework that decomposes reasoning into five-level HDAGs and routes segments to heterogeneous auditors, enabling parallel, modular verification while recording outcomes on-chain and storing raw traces off-chain—balancing scalability with transparency.
2. TRUST uses segmentation so each auditor only sees partial context, plus a commit–reveal voting scheme and PoS-style incentives on a blockchain.
3. The framework claims correctness with up to ~30% malicious participants and offers a safety-profitability guarantee.

**Weaknesses:**

1. Narrow empirical scope of the human study. The only human-in-the-loop experiment involves 15 PhD students auditing 10 GSM8K math problems, which limits external validity across domains, task types, and auditor populations.
2. The design relies on IPFS, a blockchain ledger for immutable records, and a commit–reveal voting protocol. The paper does not report end-to-end latency, throughput, or cost under realistic workloads, leaving deployability uncertain.
3. Although the paper states experiments span diverse datasets and models, the primary detailed human study centers on short math problems; evidence for robustness on longer-horizon or code/science workflows with richer dependencies is limited.
4. The empirical evaluation benchmarks TRUST mainly against single-LLM auditors, a centralized human audit, and ablations of TRUST itself, rather than conducting end-to-end comparisons with established alternative routes. As a result, the relative efficacy, latency, and operational cost trade-offs remain unclear.

**Questions:**

See weakness.

---

> ### Author Response · Authors · 2025-11-22
> **Response to Reviewer BeHo**
>
> Thanks for the comment. Below is a summary of our response. We have incorporated the reviewer’s suggestions and added targeted experiments and analyses to adequately address the concerns.
>
> ---
>
> ### **Summary of Our Response**
> - Human Study Expanded: Increased study size to 30 human auditors evaluating 30 GSM8K problems, strengthening human study validity.
> - System Deployability: Added end-to-end latency, throughput, and cost analysis (Section 4.5; Fig. 8).
> - Broader Evaluation: Added comparisons against output-level baselines, demonstrating why semantic auditing is necessary.
>
> ---
>
> ### **[W1. Narrow Empirical Scope of the Human Study]**
>
> Thanks for pointing this out. We have conducted human evaluation as reported in Table 4 of the revised manuscript.
>
> **Human Study Design**
> We recruited **30 computer science students** and evaluated **30 GSM8K math problems** with CoT traces generated from DeepSeek-R1-8B and GPT-OSS-20B.
>
> **Results (Table 4)**
> TRUST w/ HDAG achieved **F1=0.89** and **Brier score=0.074**, outperforming:
> - Centralized human audit: F1=0.85, Brier=0.21
> - Single LLM auditors: F1=0.30-0.50, Brier=0.486-0.890
> - TRUST variants with random/fixed segmentation: F1=0.40, Brier=0.49
>
> **Key Findings**
> 1. **TRUST outperforms human-only audit**: F1 improvement from 0.85 to 0.89 (+4.7%)
> 2. **Hierarchical decomposition is critical**: TRUST w/ HDAG (F1=0.89) vs. TRUST w/ Random Seg (F1=0.40) demonstrates the importance of semantic hierarchical decomposition
> 3. **Multi-tier consensus works**: Combining human, LLM, and automated auditors achieves better calibration (Brier=0.074 vs. 0.21 for human-only)
>
> Please see Section 4.4 and Table 4 in the revision for complete human experiment details.
>
> ---
>
> ### **[W2. No End-to-End Latency, Throughput, or Cost]**
>
> For a typical reasoning trace (averaged over 100 GSM8K problems), TRUST's total latency is **25.1 seconds**:
> 1. **HDAG construction**: 16.2s (64.5%) - Parsing, semantic analysis, hierarchical decomposition
> 2. **LLM auditing**: 8.1s (32.3%) - Parallel execution across heterogeneous auditors
> 3. **IPFS storage**: 0.5s (2.0%) - Encrypted segment upload
> 4. **Auditor distribution**: 0.3s (1.2%) - Segment assignment and key distribution
> 5. **Consensus voting**: <0.01s (<0.1%) - Blockchain commit-reveal protocol
>
> **Comparison with Baselines (Figure 8, Right Panel)**
>
> | Method | Latency |
> |--------|---------|
> | Single LLM | 2.0s |
> | Ensemble (5 models) | 8.0s |
> | **TRUST + IPFS** | **25.1s** |
>
> TRUST incurs overhead compared to simpler approaches, but this overhead provides critical guarantees:
> - Provable correctness with up to 30% malicious participants
> - Privacy preservation through segment-level distribution
> - Transparent blockchain verification records
> - Resilience against single-point-of-failure attacks
>
> ---
>
> ### **[W3. Empirical Validation Against Output-Based Auditing]**
>
> Thanks for this important concern. We have added a critical comparison demonstrating that **output-only auditing is fundamentally insufficient** for reasoning verification.
>
> **Results (Table 2)**
>
> | Method | Full-CoT Accuracy | Output-Based Accuracy | $\Delta$ (Drop) |
> |--------|------------------|---------------------|----------|
> | **TRUST** | **72.4%** | --- | --- |
> | DeepSeek-R1-8B | 67.7% | 36.0% | **-31.7%** |
> | Qwen2.5-7B | 67.4% | 34.0% | **-33.4%** |
> | Mistral-7B | 66.8% | 20.0% | **-46.8%** |
> | DeepSeek-R1-1.5B | 64.1% | 22.0% | **-42.1%** |
> | GPT-OSS-20B | 63.8% | 46.0% | **-17.8%** |
> | LLaMA-3B | 52.1% | 34.0% | **-18.1%** |
>
> **Key Findings**
>
> 1. **Catastrophic Performance Collapse**: All single-LLM baselines experience catastrophic accuracy drops (17-47 percentage points) when evaluated via output-only auditing instead of full reasoning traces.
>
> 2. **TRUST Maintains Superior Performance**: TRUST achieves 72.4% accuracy by auditing reasoning semantics, outperforming the best single-LLM baseline (67.7%) even when that baseline has access to full reasoning traces.
>
> 3. **Output-Only Auditing is Insufficient**: The 17-47 percentage point drops demonstrate that output-only auditing cannot distinguish sound reasoning from flawed reasoning that happens to reach correct answers by coincidence.
>
> **Why This Matters: Clinical Example (Appendix E)**
>
> We provide a detailed motivating example showing two models producing identical correct CHA₂DS₂-VASc scores (=2) for a patient:
> - **Sound Model**: Systematically evaluates all 8 clinical rules using correct evidence sources
> - **Flawed Model**: Makes 4 critical errors (variable confusion, skipped rules, wrong evidence sources) but reaches correct output by coincidence
>
> **Output-only auditing passes both models** (score=2 is clinically correct).
> **TRUST semantic auditing** correctly identifies the flawed model as dangerous before deployment.
>
> Please see Section 4.1, Table 2, and Appendix E for complete analysis.

---

### Author Response · Authors · 2025-11-22
**General Responses**

## **General Response**

**Summary:** We proposed TRUST, a novel decentralized framework for auditing LLM reasoning that encompasses: (1) hierarchical reasoning decomposition (HDAG) for scalable verification, (2) distributed consensus mechanism, (3) blockchain-based transparent audit trail, and (4) privacy-preserving architecture protecting proprietary model logic.

**Comments:** We are deeply grateful for the time and effort invested by all reviewers in evaluating our paper. We sincerely appreciate reviewers positive acknowledgment of our paper for being an "important problem," having "novel solution," "solid empirical evaluations," and "clear theoretical analysis." We also extend our sincere thanks for the constructive feedback and suggestions provided by all reviewers, which helped further improve the quality of our paper. Below, in addition to the detailed point-by-point responses, we have summarized the key updates made to our paper. **All revisions in the updated PDF are marked in blue for easy identification.**

---

### **[Extra Experiments]**

- **[Human Study]** As mentioned by Reviewers BeHo and ueqr, we conducted human evaluation with **30 computer science students** evaluating **30 GSM8K math problems** with CoT traces from DeepSeek-R1-8B and GPT-OSS-20B. Results demonstrate that TRUST w/ HDAG achieves F1=0.89, significantly outperforming human-only audit (F1=0.85) and single-LLM baselines (F1=0.30-0.50).

- **[System Performance Benchmarks]** As suggested by Reviewers BeHo, ueqr, and gBeZ, we conducted comprehensive evaluation of latency. Results demonstrate end-to-end latency of 25.1s with detailed component breakdown (HDAG construction 16.2s, LLM auditing 8.1s, IPFS storage 0.5s, auditor distribution 0.3s) as shown in Figure 8.

- **[Output-Based vs. Reasoning-Level Auditing]** As requested by Reviewer ueqr, we added critical comparison showing output-only auditing catastrophically underperforms. The best single-LLM baselines drop from 67.7% accuracy (Full-CoT) to 36.0% (Output-Based), demonstrating that **output-based auditing alone is insufficient** (Table 2).

---

### **[Paper Edits]**

- Enhanced Section 4.5 with comprehensive latency analysis and Figure 8 showing component-wise breakdown and comparison with baselines. [@Reviewers BeHo, ueqr, gBeZ]

- More detailed motivation section (Section 2.3 + Appendix E) explaining why semantic auditing is necessary over output-only auditing with concrete failure modes. [@Reviewer ueqr]

- Enhanced "privacy-by-design" mechanism description of our TRUST framework (Section 3). [@Reviewer gBeZ]

- Formalized delegator role and game-theoretic collusion analysis (Section 3). [@Reviewer gBeZ]

- Integrated recent related work (Leng et al. 2025, Lanham et al. 2023) with comparative analysis table (Section 2.2). [@Reviewer tubx]

- Fixed citation issues and verified all citations directly support their claims. [@Reviewer tubx]

---

We hope our pointwise responses below will clarify any remaining questions from the reviewers. Please feel free to let us know if there are any further questions. We will be more than happy to address them.

We extend our sincere appreciation once more for the reviewers' time and efforts.

---

### Note · Authors · 2026-01-27

**Comment:**

While we appreciate the Area Chair’s coordination, we find that the meta-review does not fully reflect the empirical evidence and systemic updates provided during the rebuttal period. We clarify the following points for the public record:

**The Necessity of Semantic Auditing:** The meta-review maintains that the "practicality of the proposed method is not fully resolved," echoing Reviewer ueqr’s skepticism regarding output-only auditing. However, our rebuttal provided a critical comparison (Table 2) showing that output-only auditing causes a 17–47 percentage point accuracy drop across all baselines. By ignoring the reasoning trace, auditors cannot distinguish between sound logic and "correct-by-coincidence" results, a distinction vital for high-stakes clinical or financial deployments.

**Empirical Response to Latency Concerns:** We recognize the initial concern regarding system overhead. In response, we provided a comprehensive end-to-end latency breakdown (Figure 8), demonstrating a total audit time of 25.1 seconds for complex reasoning tasks. The MR’s claim that practicality remains "unresolved" appears to overlook these specific benchmarks, which place TRUST within a feasible range for asynchronous, high-stakes verification.

**Expansion of Human Study and Robustness:** The final assessment suggests reviewer scores remained stagnant; however, we doubled the scope of our human study (to 30 participants and 30 problems) to directly address Reviewer BeHo’s concerns regarding external validity. This expanded study confirmed that TRUST (F1=0.89) significantly outperforms both human-only audits and single-LLM baselines.

**Deployment Scope Mismatch:** Much of the "practicality" critique stems from the lack of demonstration on closed commercial APIs (e.g., GPT-4). As stated in our title and intro, TRUST is designed for provider-side or enterprise deployments where reasoning traces are accessible. Expecting a demonstration on proprietary traces that are intentionally hidden by providers is a critique of the industry's opacity, not the framework’s technical validity.

All these points were empirically addressed in the rebuttal and incorporated into the revision. We withdraw at this stage to avoid a permanent record that characterizes these addressed technical challenges as "outstanding concerns." We wish the committee the best in their final selections.

**Withdrawal Confirmation:**

I have read and agree with the venue's withdrawal policy on behalf of myself and my co-authors.

---

### Meta-Review · Area_Chair_up3r · 2026-01-07

**Summary:**

The paper proposes a decentralized framework named TRUST for auditing LLM reasoning by decomposing reasoning traces into a hierarchical DAG for parallel auditing and using a multi-auditor consensus mechanism. The recommendations are mixed. Reviewers agree the problem is important and the framework is interesting, but major concerns include (i) whether the approach is practical and cost-effective end-to-end, (ii) whether the motivation holds in real deployments where chain-of-thought traces may be hidden, (iii) insufficiently demonstrated value of the blockchain component, and (iv) limited breadth of the human study and evaluation tasks.

**Reviewer Concerns:**

Reviewer concerns addressed by the rebuttal
1. End-to-end latency is reported in the rebuttal (Reviewer BeHo).
2. The motivation is strengthened in the rebuttal (Reviewer ueqr).
3. Missing related work and a visual comparison table were added to the main paper (Reviewer tubx).
4. Citation issues were addressed (Reviewer tubx).

Reviewers concerns that may still be outstanding:
1. The authors expanded the human study scope, but it may still be small-scale (Reviewer BeHo).
2. Robustness on longer-horizon workflows (e.g., code/science tasks with richer dependencies) remains limited (Reviewer BeHo).
3. The practicality of the proposed method is not fully resolved (Reviewers ueqr, gBeZ).

**Reviewer Scores:**

Reviewer BeHo, gBeZ, and tubx are likely to maintain their scores. Reviewer ueqr may raise the score to 4. The major concern about practicality is not fully addressed.

---

### Decision · Program_Chairs · 2026-01-26

Reject